# Site-specific characterization of endogenous SUMOylation across species and organs

Ivo A. Hendriks [1], David Lyon [2], Dan Su[3], Niels H. Skotte[1], Jeremy A. Daniel[3], Lars J. Jensen[2] & Michael L. Nielsen [1]

Small ubiquitin-like modifiers (SUMOs) are post-translational modifications that play crucial roles in most cellular processes. While methods exist to study exogenous SUMOylation, large-scale characterization of endogenous SUMO2/3 has remained technically daunting. Here, we describe a proteomics approach facilitating system-wide and in vivo identification of lysines modified by endogenous and native SUMO2. Using a peptide-level immunoprecipitation enrichment strategy, we identify 14,869 endogenous SUMO2/3 sites in human cells during heat stress and proteasomal inhibition, and quantitatively map 1963 SUMO sites across eight mouse tissues. Characterization of the SUMO equilibrium highlights striking differences in SUMO metabolism between cultured cancer cells and normal tissues. Targeting preferences of SUMO2/3 vary across different organ types, coinciding with markedly differential SUMOylation states of all enzymes involved in the SUMO conjugation cascade. Collectively, our systemic investigation details the SUMOylation architecture across species and organs and provides a resource of endogenous SUMOylation sites on factors important in organ-specific functions.

[1] Proteomics Program, Novo Nordisk Foundation Center for Protein Research, Faculty of Health and Medical Sciences, University of Copenhagen, Blegdamsvej 3B, 2200 Copenhagen, Denmark. [2] Disease Systems Biology Program, Novo Nordisk Foundation Center for Protein Research, Faculty of Health and Medical Sciences, University of Copenhagen, Blegdamsvej 3B, 2200 Copenhagen, Denmark. [3] Protein Signaling Program, Novo Nordisk Foundation Center for Protein Research, Faculty of Health and Medical Sciences, University of Copenhagen, Blegdamsvej 3B, 2200 Copenhagen, Denmark. Correspondence and requests for materials should be addressed to M.L.N. (email: michael.lund.nielsen@cpr.ku.dk)

Small ubiquitin-like modifiers (SUMOs) are post-translational modifications (PTMs) that regulate many cellular processes[1], including virtually all nuclear functions[2]. SUMOylation is indispensable for eukaryotic life, and most vertebrates express three SUMO family members, SUMO1, SUMO2, and SUMO3. All SUMOs are conjugated by the same set of enzymatic machinery comprising one heterodimeric E1, a single E2, and a handful of E3 enzymes. SUMOylation is a reversible process, and several SUMO-specific proteases exist to facilitate removal of SUMO from lysines. SUMO2/3 are often referred to interchangeably owing to their high degree of homology[3], and overall the SUMO machinery is conserved across eukaryotes. In mouse, knock-out of SUMO2, which is the most abundantly expressed out of SUMO2/3, is embryonic lethal[4].

A central goal in biology is to understand the cellular mechanisms governing biological processes, and SUMO has emerged as an important factor in nuclear protein assemblies and a distinguished mark for functionally engaged processes[5]. Another mechanistic feature of SUMO is to spatially target functionally related proteins[6], emphasizing the importance of understanding the substrate specificity and global regulation of SUMOylation. Moreover, SUMO is essential for maintenance of genome integrity and regulation of intracellular signaling. Hence, SUMO has become an increasingly important subject of study in biological sciences and biomedicine, with the modification linked to various diseases including cancer, where SUMO or the SUMO enzymatic machinery are frequently misregulated[5,7], while SUMO-mediated transcriptional regulation is required to facilitate certain types of tumorigenesis[8]. Similarly, alterations in SUMO regulation have been associated with ischemia[9,10], Huntington's disease[11], diabetes[12], and heart failure[13,14]. As a result, inhibitors of SUMOylation are actively being developed[15], with clinical applications in mind.

To understand the functional behavior of SUMOylation in health and disease, it is pivotal to study SUMO at the endogenous level and within the confines of the relevant model systems. However, despite great biological and clinical interest, current knowledge of endogenous and systemic regulation of SUMOylation remains scarce. Although recent developments within mass spectrometry (MS)-based proteomics have facilitated insights into the extent of SUMOylation in cells[16], these studies were based upon introduction of exogenously expressed, epitope-tagged, and often mutated SUMO variants. As a result, these methods are unable to investigate endogenous SUMOylation, and infeasible for in vivo tissue analyses without relying on genetic engineering.

Although SUMO was discovered 20 years ago[17,18], the interest in studying SUMOylation in a biomedical context using proteomics approaches continues to grow. However, it has remained technically daunting to establish analytical strategies allowing site-specific characterization of endogenous SUMO, partly owing to the overall low stoichiometry of SUMO, along with highly active SUMO-specific proteases only deactivated under harsh buffer conditions[19]. Additionally, when using conventional proteomics approaches, digestion of SUMO2/3 with trypsin leaves a 32-residue mass tag on modified lysines, which complicates MS/MS analysis[16]. Still, contemporary approaches for studying endogenous SUMO have allowed detection of a limited number of natively SUMOylated targets at the protein level[20], or have employed genetic engineering to purify proteins modified by epitope-tagged SUMO from mice[21–23]. Further, a recent approach identified a small number of putative endogenous SUMO sites[24]. However, all current methods fail to facilitate systems-wide analysis of specific, endogenous, native, and in vivo SUMO2/3 sites.

Given the biomedical importance of understanding the cellular architecture of endogenous SUMO2/3, we set out to map

SUMOylation to great depth across cells and tissues. To this end, we developed an MS-based proteomics method that facilitates purification of peptides modified by endogenous, wild-type SUMO2/3, and therein pinpoint modified lysines. The method relies on a well-characterized antibody[25], and analysis of samples with standard MS approaches and freely available software. From triplicate analyses of Human embryonic kidney 293 (HEK) cells, we identified 14,869 unique SUMO2/3 sites mapping to 3870 endogenously SUMOylated proteins. However, immortalized cancer cell lines are selected for high proliferation rates and commonly do not represent the complex biological conditions in tissues. Thus, in order to delineate the physiological differences in SUMO architecture between tissues we mapped nearly 2000 SUMO2/3 sites from single-organ analyses of eight different mouse tissues.

Collectively, we describe the largest endogenous SUMO2/3 proteomics resource to date, provide insight into the baseline SUMOylation across cells and various organs, and highlight differences in SUMO architecture between these highly distinct model systems. Because our method facilitates the study of endogenous and in vivo SUMOylation patterns in most vertebrates, we present a final step in equalizing the proteomics playing field for the system-wide study of SUMO2/3.

## Results

**Enrichment of endogenously SUMOylated peptides**. To facilitate enrichment of SUMOylated peptides, we utilized the commercially available SUMO2/3 8A2 antibody[25], which recognizes the C-terminus of SUMO2/3 conserved across vertebrates, including human, mouse, chicken, Drosophila, Xenopus, and zebrafish. The antibody can also be acquired from the Developmental Studies Hybridoma Bank for in-house production[26], and has previously been used to enrich endogenously SUMOylated proteins[20,26]. However, despite identifying a few hundred SUMO substrates, no modification sites were reported.

To facilitate efficient proteomics analysis of SUMO2/3-modified lysines, we devised a peptide-level enrichment strategy utilizing the epitope of the 8A2 antibody, mapped to reside within the C-terminal part of SUMO2 (57-IRFRFDGQPI-66)[20]. We reasoned that protein digestion using trypsin would cleave within the epitope of SUMO2/3, thus prohibiting recognition by the 8A2 antibody, whereas digestion with endoproteinase Lys-C would leave the epitope intact. We confirmed this using dot blot analysis of tryptic and Lys-C digests of HeLa total lysate (Supplementary Fig. 1A–B). Moreover, as the endoproteinase Lys-C remains active at high molar concentrations of chaotropic buffers[27], this allows samples to be rapidly lysed under denaturing conditions, ensuring rapid inactivation of SUMO proteases and thereby preventing loss of SUMOylation during sample preparation.

Following Lys-C digestion, all samples were desalted on C8 resin to allow efficient capture of large hydrophobic peptides, bearing in mind that the SUMO2/3 mass remnant generated upon Lys-C digestion entails a total mass of 5.6 kDa (Supplementary Fig. 1C–D). Following desalting, samples were lyophilized and SUMOylated peptides were efficiently enriched by SUMO-IP using the 8A2 antibody (Supplementary Fig. 2A–B). Notably, the amount of antibody we used per milligram of starting protein material was 20 times lower than reported previously[20] (Supplementary Note 1).

Although efficient enrichment of peptides modified by endogenous SUMO is challenging, the single-largest complication of proteomics analysis of SUMO2/3 relates to the large mass remnant that remains after standard protein digestion. To alleviate this, we performed a series of pilot experiments to establish an effective strategy entailing a second digestion step to

complement the Lys-C digestion required for the SUMO-IP step. Comparison of several different proteinases revealed that Asp-N facilitated both consistent and reliable identification of endogenously SUMOylated peptides, yielding four-fold higher identification numbers compared to the other enzymes and generating a mass remnant that uniquely identified peptides as modified by SUMO-2/3 (Supplementary Fig. 2C–E and Supplementary Note 1). Although our strategy entails a digestion, purification, followed by another digestion; serial digestion methods are widely used in the proteomics field[16], and yield highly pure samples

when executed properly. Conclusively, we adopted a serial digestion strategy using Lys-C and Asp-N as the main workflow for the rest of our proteomics experiments (Fig. 1a).

**Mapping the endogenous SUMOylome in human cells.** Having established an optimized SUMO2/3 purification strategy, we benchmarked our method in HEK cells by profiling endogenous SUMOylation in triplicate under standard cell culture conditions, and in response to either heat shock or the proteasomal inhibitor

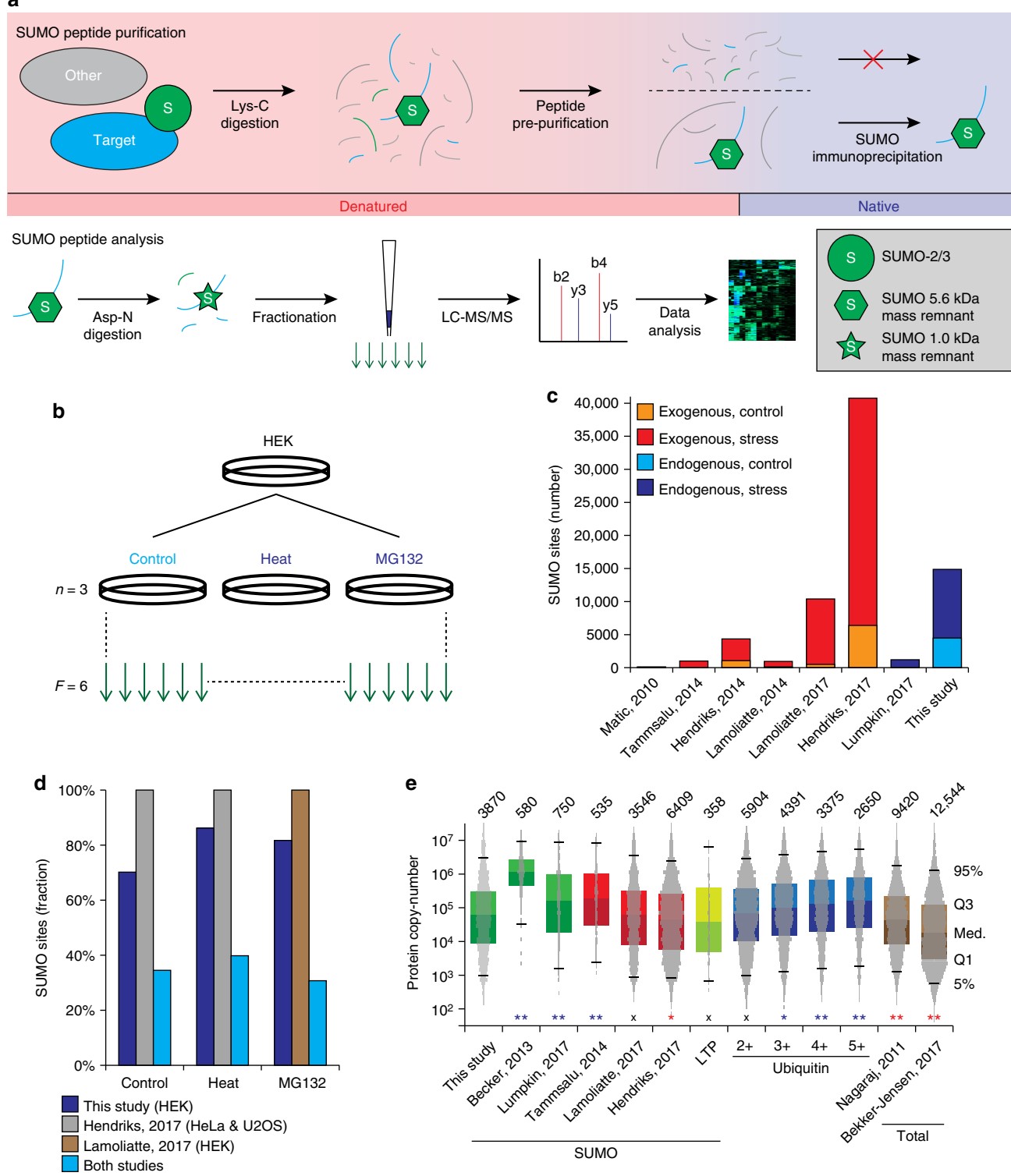

MG132 (Fig. 1b)[28]. All experiments were analyzed using a streamlined MS proteomics approach (Supplementary Note 2).

Collectively, we mapped 14,869 unique endogenous SUMO2/3 sites across the investigated cellular conditions in human cells, with 4476 sites identified under standard growth conditions (Fig. 1c, Supplementary Data 1 & 2). We observed an excellent degree of reproducibility with high Pearson correlations between same-condition replicates (Supplementary Fig. 3A), unsupervised hierarchical clustering of same-condition replicates (Supplementary Fig. 3B), and close grouping of same-condition replicates in principle component analysis (PCA) (Supplementary Fig. 3C). Although not reaching the same number of identified SUMOylation sites compared to our recent study using the exogenous K0-SUMO method[2], this study nonetheless represents the largest compendium of endogenous SUMO2/3 sites to date (Fig. 1c). Still, we identified similar numbers of SUMO2/3 sites as exogenous methods when performed at similar scale and cellular background (Fig. 1d, Supplementary Note 3).

To gain insight into the overall sensitivity of our endogenous approach, we compared the depth of sequencing achieved in our data against other SUMO2/3 proteomics studies. To this end, we compared the copy-numbers of identified SUMO2/3 target proteins, by utilizing protein-per-cell copy-numbers derived from IBAQ analysis of a deep HeLa proteome[29]. We found that endogenous SUMO2/3-targeted proteins entailed a median copy-number of 60,000, which was reassuringly similar to the largest exogenous SUMO studies[2,30], demonstrating that our strategy can reach great depth of sequencing despite the technical challenges associated with the study of endogenous SUMO2/3 (Fig. 1e). Compared to previous proteomics studies focused on endogenous SUMO[20,24], our approach achieved at least 3-fold greater sequencing depth. To emphasize that our method is not limited in sensitivity, we additionally compared our data to a similarly-sized group of 4391 ubiquitylated proteins identified by at least three lysine residues[31]. We observed a median protein copy-number of 100,000 for these ubiquitylated proteins, which further supported that our method achieves good sensitivity, and suggests that SUMOylation may systemically occur on less abundant proteins compared to ubiquitylation.

To demonstrate that our method does not entail a sequence-specific bias, we analyzed the adherence of identified SUMO2/3 sites to the KxE consensus motif, as this commonly is considered an important benchmark in SUMO proteomics data[32]. We found 24.9% of all SUMOylated lysines to reside in the KxE-type consensus motif, with 60.1% of total SUMOylation occurring on KxE motifs (Supplementary Data 1). When only considering sites identified under standard growth conditions, 36.4% of sites matched KxE, with 63.0% of total SUMO2/3 residing on KxE motifs. Overall, we validated our method by

reconfirming canonical SUMO phenomena including specific consensus sub-motifs and distribution of sites-per-protein (Fig. 2, Supplementary Fig. 4 and Supplementary Note 4).

Next we performed a full comparison of SUMO2/3 sites identified in this study, to those previously reported in proteomics screens[2,19,24,28,30,33–39], in addition to SUMO sites reported in the literature via low-throughput approaches (Supplementary Data 3). Out of all SUMO2/3 sites we identified, 67.0% were previously reported in screens primarily using exogenous SUMO, rising to 88.2% when looking at the top 10% highest-scoring sites. Differences in overlap could largely be attributed to variable protein digestion patterns, as most proteomics SUMO screens rely on tryptic digestion, whereas we employed a Lys-C/Asp-N approach.

We compared SUMO2/3 target proteins identified in our study to those identified by other SUMO proteomics studies[2,19,20,24,28,30,34–40]. Here, we observed an overlap of 87.8%, visualizing a higher overlap at the protein-level as compared to the site-level, suggesting that mainly sites within the same SUMO2/3 target proteins were identified by exogenous approaches (Supplementary Data 4). When considering the top 10% highest-scoring SUMOylated proteins, the overlap increased to 99.0%. Differences in the cellular SUMOylation response to stress were quantified in HEK cells, and overall found to be comparable to the global stress response observed in a previous study[2], with Pearson correlations of $R = 0.56$ and $R = 0.59$ between co-identified proteins and sites in response to MG132, respectively (Supplementary Fig. 5).

**A catalogue of in vivo SUMO in eight types of mouse organs.** As our method exhibited good sensitivity, we next sought to understand the physiological differences of SUMOylation across tissues. To this end, we performed SUMO-IP on eight types of mouse organs—brain, heart, kidneys, lungs, liver, skeletal muscle, spleen, and testes (Fig. 3a). Using immunoblot analysis, we validated that our SUMO-IP worked efficiently in mouse organs (Supplementary Fig. 6). To properly assess reproducibility of our method and gain quantitative insight, we performed single-organ analyses across five wild-type animals with all organs used in their entirety, excepting liver of which only half was analyzed.

Across all organs, we identified 1963 SUMOylation sites (Fig. 3b and Supplementary Data 5), mapping to 955 protein-coding genes (Supplementary Data 6). The number of SUMOylation sites per organ type appeared unrelated to the size or protein content of the organs. Overall, we observed the highest number of SUMOylation sites in liver, testis, kidney, and spleen, identifying >700 SUMO2/3 sites by direct MS/MS in these organs, with up to 1131 sites in liver (Fig. 3b). The lowest numbers of SUMOylation

**Fig. 1** A strategy for identifying endogenous SUMO2/3 sites. **a** Schematic overview of the purification strategy. Briefly, a denaturing lysate is prepared and digested with Lys-C. Peptides are pre-purified using C8 SepPak cartridges, after which peptides are lyophilized. The peptides are then dissolved in a mild buffer to facilitate immunoprecipitation using the 8A2 antibody. Purified SUMOylated peptides are subjected to a second round of digestion using Asp-N, after which they are fractionated on StageTip and analyzed by nanoscale LC-MS/MS. **b** Experimental design for the main cell culture data. All experiments were performed in cell culture triplicates, and analyzed as six fractions. **c** Overview of the total number of SUMO2/3 sites identified in this study, as compared to several other SUMO proteomics studies. **d** Overview of the relative number of SUMO2/3 sites identified per condition in this study, as compared to the largest currently published SUMO proteomics studies that used comparable cellular treatments[2,30]. **e** Estimation of depth of sequencing through comparison of the copy-number per cell of identified proteins. IBAQ-determined copy-numbers of proteins were derived from a recent deep total HeLa proteome study[29]. Protein copy-numbers were assumed to be comparable on average across different cultured human cell lines[29]. Top whisker: 95th percentile, top bound: 3rd quantile, center line: median, bottom bound: 1st quantile, bottom whisker: 5th percentile. Total numbers of proteins identified are displayed above the bars. The copy-number distribution histogram of detected proteins is overlaid for all datasets. For ubiquitin, the numbers indicate the tested number of ubiquitylated lysines per protein, as extracted from the PhosphoSitePlus ubiquitin site database. LTP low throughput. Asterisks denote significant differences between SUMO2/3 target proteins identified in this study and other datasets, with blue or red asterisks indicating our study achieved more or less depth, respectively. Determined by two-tailed Fisher's exact testing. * $P < 0.05$, ** $P < 0.001$, $^\times$ N.S

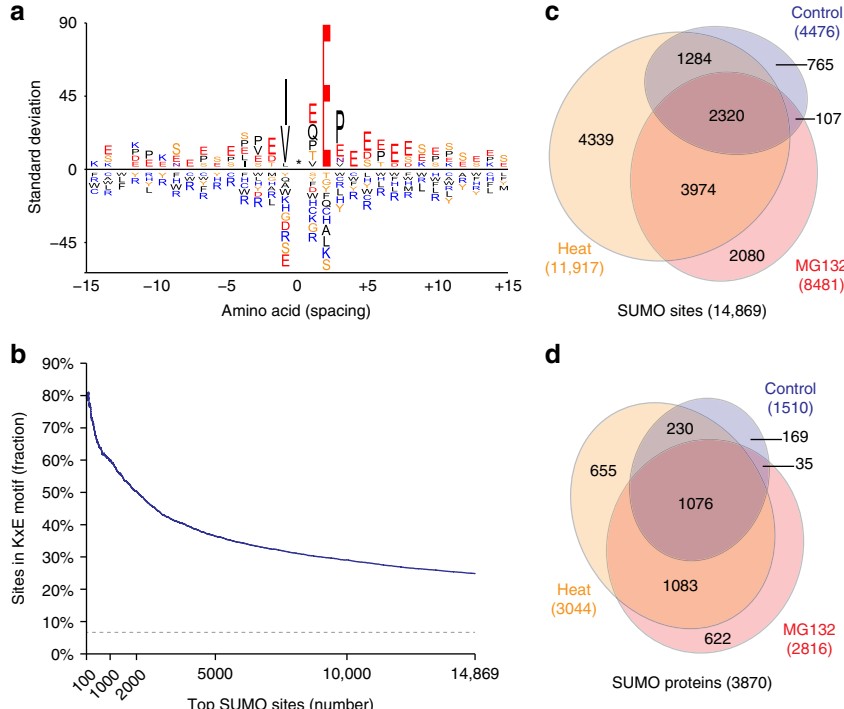

**Fig. 2** The endogenous SUMO2/3 consensus motif and SUMO dynamics. **a** IceLogo representation of amino acid residues enriched around the top 10% SUMOylated lysines identified in human cells, at position 0, indicated by the asterisk. Residues displayed above the line are enriched, and those displayed below the line are depleted, compared to a reference set of human nuclear proteins. **b** The adherence of all identified SUMO2/3 sites to the KxE motif, in relation to the average of top-scoring sites. **c** Venn diagram of SUMO2/3 sites identified in response to different treatments, drawn to scale. **d** Venn diagram of SUMO2/3 target proteins identified in response to different treatments, drawn to scale

sites were identified in heart and brain, both falling short of 200 MS/MS-identified sites. Using matching of MS1-level evidence between experiments and organs, we were able to increase identification density, hereby mapping >600 SUMO2/3 sites in all organs, >1000 sites in five organs, and up to 1309 unique sites in liver. Accordingly, the number of sites that could be identified across multiple organs increased considerably with the matching data (Fig. 3c).

Globally, we observed 38.1% of SUMOylated lysine residues in mouse organs to reside in the KxE motif, with 50.3% of all SUMOylation occurring on these motifs. The top 100 sites reached 67% KxE adherence, and the top 1000 sites matched 49.4% KxE (Fig. 3d), a number on par with proteomics studies reporting equal numbers of identified SUMO2/3 sites in cell lines under standard conditions[28,39]. Similar to the human SUMO2/3 data, we could confirm the existence of most known SUMOylation sequence motifs within the mouse data (Fig. 3e), although we did not observe enrichment for aspartic acid residues at −2 when using iceLogo analysis, while threonine residues appeared enriched from −5 to −1. To verify significantly enriched SUMO motifs, we additionally performed Motif-X analysis (Supplementary Data 5 and Supplementary Fig. 7), which confirmed a significant presence of KxE-type motifs. In addition, we observed a [ED]xKP motif, suggesting that the inverted SUMOylation consensus motif is significantly SUMOylated in mouse tissue when the modified lysine is followed by a proline residue. Overall, we found that the KxE adherence is variable between organ types, and did not correlate with site abundance or total number of sites (Fig. 3f). Collectively, our findings support that basic SUMO2/3 targeting preferences are comparable between cultured cells and tissues, with some variation in number of sites and KxE adherence between different organ types.

**Functions of SUMO in organs**. SUMOylation in mouse globally adhered to functions that are canonically associated with SUMO (Fig. 3g and Supplementary Data 7), as determined using term annotation enrichment analysis with multiple-hypothesis corrected two-tailed Fisher's P-values to ensure an FDR of <2%. The most enriched terms highlighted SUMO2/3 modification of transcription factors, chromatin regulators, numerous macromolecular protein complexes, and a predominantly nuclear localization. To gain insight into SUMO target proteins across organs, we performed label-free relative quantification of SUMO sites and proteins (Tables S5 and S6). Out of 1963 identified SUMO2/3 sites and 955 SUMO2/3 target proteins, we could quantify 46.3% and 77.8%, respectively. Reproducibility of the five replicates was assessed using scatter plot analysis, and we observed a high average Pearson correlation ($R = 0.69$) between all same-organ replicates, with the highest reproducibility ($R = 0.82$) in liver (Supplementary Fig. 8A). We moreover observed good reproducibility between the replicate animals with organs clustering together hierarchically (Fig. 4a), and in PCA (Fig. 4b). At the SUMO site level, hierarchical clustering and PCA analysis additionally supported data reproducibility (Supplementary Fig. 8B–C), and similarly outlined correlations between organ types, with the largest deviations in the PCA observed for liver and brain.

In order to assess whether any biological functions were enriched for SUMOylation across organ types, we considered SUMO2/3 target proteins inferred from MS/MS-identified SUMO sites only, and compared the SUMOylated proteins to the respective background proteomes for each organ type as documented in the TISSUES database[41]. Subsequently, we performed annotation enrichment analysis to extract qualitative differences between the SUMOylated proteins and the

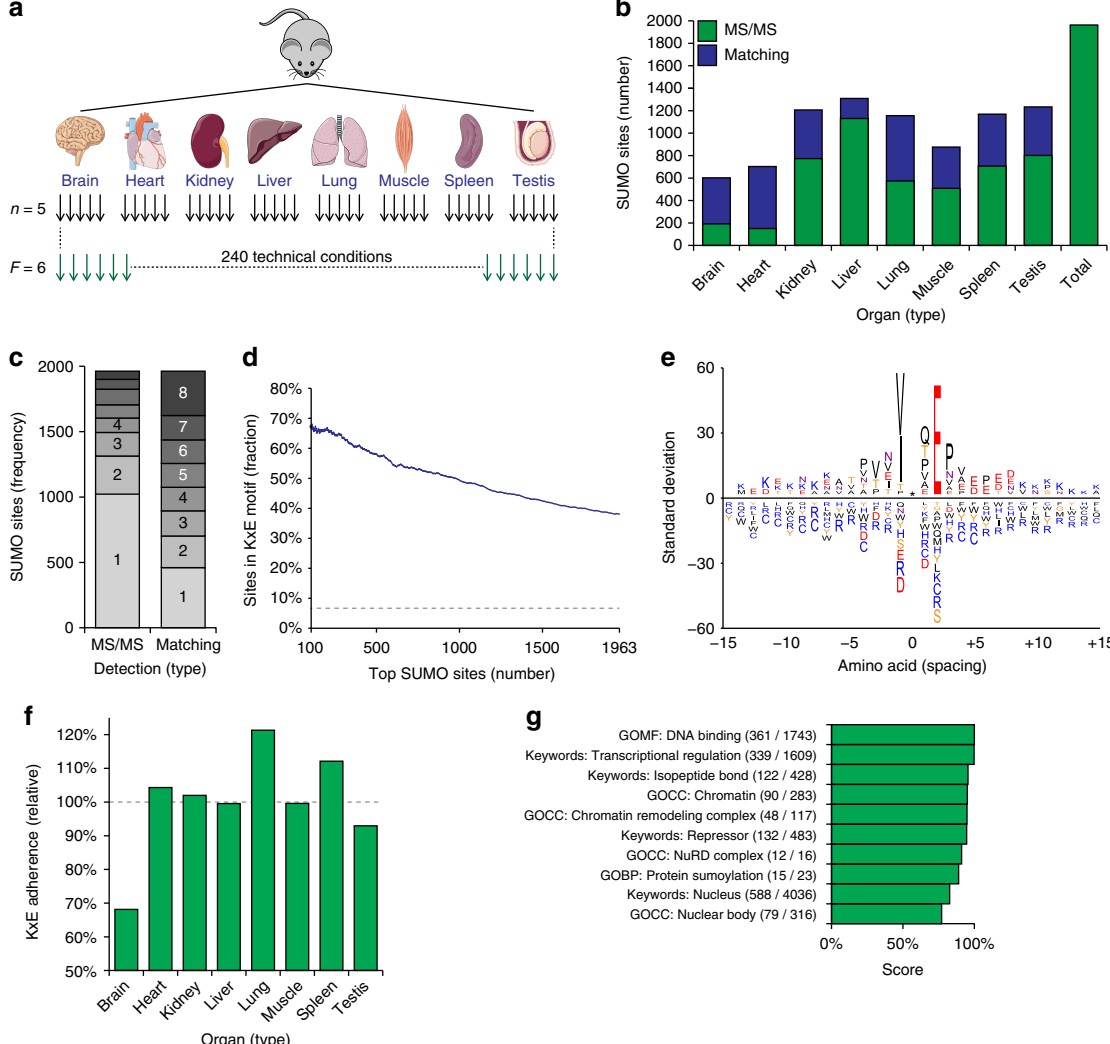

**Fig. 3** Mapping the mouse SUMOylome. **a** Experimental design of SUMO proteomics experiments carried out in mice. Eight types of organs were studied, in biological quintuplicate (five animals), and analyzed as six fractions. **b** Overview of the number of SUMO2/3 sites identified per organ, comparing direct MS/MS identifications, to identifications made when matching MS1-level data between organs. **c** Overview of the number of organs in which the same SUMO2/3 sites were detected. **d** The adherence of all identified SUMO2/3 sites to the KxE motif, in relation to the average of top-scoring sites. **e** IceLogo representation of amino acid residues enriched around all SUMOylated lysines identified in mouse, at position 0, indicated by the asterisk. Residues displayed above the line are enriched, and those displayed below the line are depleted, compared to a reference set of mouse nuclear proteins. **f** The adherence of MS/MS-identified SUMO2/3 sites to the KxE motif, per organ, relative to the normalized expected adherence based on (**d**). **g** Term enrichment analysis comparing identified mouse SUMO2/3 target proteins to mouse proteins known to be expressed in the eight organs analyzed. Numbers in parentheses indicate overlap between the numbers of proteins identified in this study, compared to those present in the reference. Score was derived from the logarithms of enrichment ratios and Benjamini–Hochberg FDR-corrected two-tailed Fisher's P-values of < 0.02. A full list of all enriched terms, and all relevant scores, is available in Supplementary Data 7

background proteomes, and moreover checked whether any enriched terms were also unique to the organ when compared to the other tissues we analyzed (Fig. 4c and Supplementary Data 8). All organs were enriched for many of the canonical SUMO functions when compared to the background proteomes (Supplementary Data 8), although we chose to focus on organ-unique terms that were significantly SUMO-enriched, in order to elucidate potential organ-specific biological pathways and functions significantly regulated by SUMOylation.

Strikingly, we did not observe many unique terms to be enriched in brain, and though core SUMO functions like nuclear localization were enriched, this occurred less so than in all other organs we investigated. In heart, pathways and functions related to heart function and development were enriched for

SUMOylation. Kidney was the only other organ where we did not observe organ-unique pathways to be enriched, likely owing to the considerable overlap we observed with liver in terms of SUMO2/3 sites. In liver, a range of metabolic functions were enriched, including amino acid metabolic pathways and the urea cycle. In lung, we observed enrichment for canonical SUMO functions, including pathways implying a more predominant chromatin-centric localization. We also observed SUMO enrichment in lung for terms associated with myeloid leukocyte and erythrocyte differentiation, suggesting that SUMOylation may play a role in immune system regulation. In muscle, SUMOylation was mainly enriched on muscle-specific proteins, and involved in glycolysis. In spleen, we observed an overall similarity to the SUMOylome in lung, and an enrichment for pathways

associated with chromatin remodelers. Finally, SUMOylation in testis was among the most distinct from all other types, with specific SUMO modification of proteins that are not necessarily expressed in a tissue-specific manner. For example, we observed strong SUMOylation enrichment for proteins associated with the DNA damage response, mitotic recombination, helicase activity, telomeres, and the global cellular stress response.

Although we corrected our term enrichment analysis for the proteomes of each organ type, SUMOylation could nonetheless be more prone to target to proteins expressed at higher levels within the distinct tissues. To assess the extent of this effect, we correlated elevated protein expression levels in each organ to increased levels of SUMOylation (Fig. 4d). Comparatively, SUMOylation in skeletal muscle and heart showed the largest dependence on protein abundance, with modification occurring on proteins predominantly expressed in these organs. Conversely, SUMOylation in brain, lung, spleen, and testis, only displayed a modest correlation with protein abundance, further suggesting

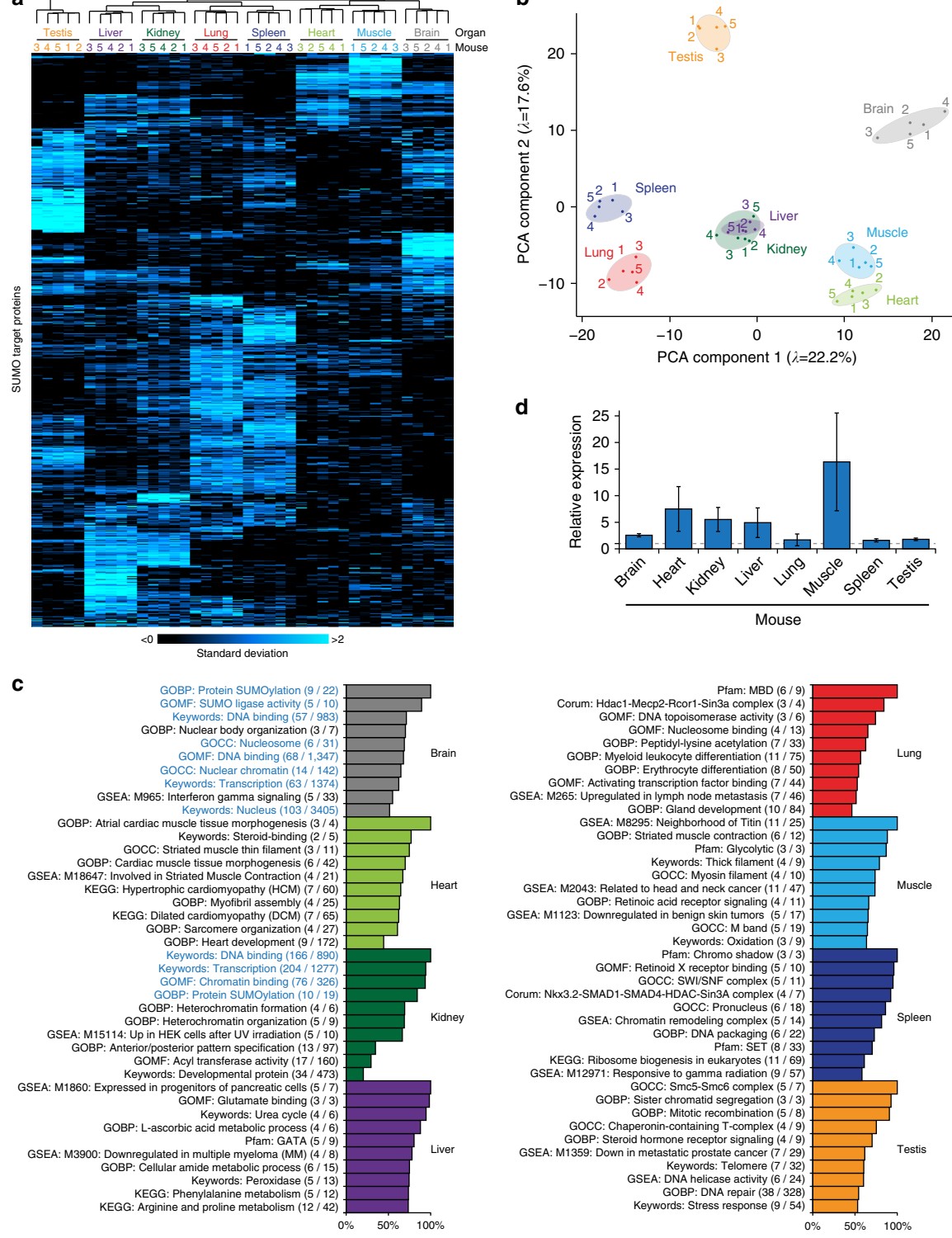

differences in the way SUMOylation is targeted to proteins in these organs.

**SUMOylation in human and mouse.** Although considerable differences exist between cultured human cells and mouse organs, we sought to elucidate whether the SUMOylome would similarly reflect this distinction. To this end, we performed a series of comparisons on modified proteins (Supplementary Data 9 and 10), including subcellular localization of SUMO target proteins (Fig. 5a, b), SUMO-phosphorylation co-modification (Fig. 5c, d, Supplementary Data 11 and 12), and structural targeting preference (Fig. 6a–c). All these analyses are further detailed in Supplementary Note 5. Overall, we observed globally constant properties of SUMOylation, including the predominant nuclear localization, preference for modifying disordered protein regions, and involvement of SUMO-phosphorylation co-modification.

Since we identified a considerable number of endogenous SUMO2/3 sites in human cells and mouse organs, we were interested in evaluating evolutionary conservation of SUMOylation across species. Generally, the SUMO enzymatic machinery is highly conserved across eukaryotes; however, conservation of the individual SUMOylation sites catalyzed by the SUMO enzymes has never been investigated on a larger scale.

We calculated Residue Conservation Scores (RCS) for all lysines in SUMOylated proteins identified in human and mouse. When considering RCS for all lysines, we found an average RCS of 74.3% for lysine residues residing in globular and buried regions, 70.9% for globular and exposed regions, and 65.4% for disordered regions (Fig. 6d). Thus, lysines in disordered regions are globally less conserved than lysines in ordered regions, in agreement with what has been described previously[42]. Strikingly, we found that SUMOylated lysines in ordered and exposed protein regions were significantly less evolutionarily conserved compared to non-modified lysines in the same structural context (Fig. 6e). Overall, SUMOylation preferentially targeted disordered regions, which are intrinsically poorly conserved, and otherwise SUMO2/3 had a tendency to modify less conserved lysines when targeted to globular regions. Thus, we highlight a high degree of SUMO evolution across species, in spite of the underlying structural and sequence-specific targeting preferences being similar between human cells and mouse tissue.

Finally, we assessed the density of SUMOylation in the investigated samples, by quantifying the total amount of SUMO2/3-modified peptides and proteins relative to the background signal (Supplementary Fig. 9 and Supplementary Note 6). We observed a SUMO-IP purity of over 93% from HEK cells, and 42–62% from mouse organs. Moreover, we found that the amount of SUMO2/3 peptides per milligram of starting material was up to 70 times larger when purifying SUMO2/3 from HEK cells as compared to liver, indicative of a large accumulation of SUMO2/3 in actively cycling HEK cells.

**Quantifying the native SUMO2/3 equilibrium.** Within eukaryotic cells, SUMO proteins are expressed as immature precursors which are processed into mature counterparts by SUMO proteases[43]. During the SUMO conjugation cycle, the E1-E2-E3 enzymatic machinery is responsible for the reversible conjugation of mature free SUMO to protein substrates[44]. However, despite the dynamic nature of SUMO conjugation, the exact equilibrium of SUMO pools remains relatively poorly understood, due to the lack of analytical tools to reliably quantify distinct pools of cellular SUMO. Because our method enriches the C-terminal part of SUMO2/3 regardless of whether it is conjugated to another protein, we were able to quantify the cellular pools of conjugated SUMO2/3, unconjugated SUMO2/3, and immature SUMO2 and SUMO3, in cultured cells and tissue (Fig. 7a), hereby providing insight into the regulation of these distinct pools of SUMO. Moreover, analysis of the conjugated pool of SUMO2/3 allows concomitant discrimination between SUMO-chain formation, modification of SUMO E1 (SAE1/UBA2), E2 (UBC9), and E3 enzymes, and conjugation to any other protein.

In HEK cells, we found 93% of SUMO2/3 conjugated to target substrates, while only 6% of SUMO2/3 existed as free SUMO (Fig. 7b). The fraction of conjugated SUMO2/3 increased further in response to cellular stress, up to 96% after MG132 and up to 98% after heat shock. However, considering the global pool of conjugated SUMO2/3 increased by ~50% in response to heat shock (Supplementary Fig. 3D), our data suggest that ongoing synthesis of new SUMO2/3 is likely essential for the SUMO stress response. Under control conditions, ~6% of SUMO2/3 was part of SUMO-chains, ~3% of SUMOylation was found modifying E3 SUMO ligases, ~1% modified the E2, and only a small fraction modified the E1. Interestingly, nearly 1% of the total SUMO pool corresponded to immature SUMO3, which is generally considered to be expressed at a lower level than SUMO2. In response to stress, we observed more E3 ligase modification, indicative of increased SUMOylation activity[44]. Additionally, in support of an increased demand for SUMO maturation, the small pool of immature SUMO2/3 that was detectable under control condition was abolished in response to stress treatments.

Intriguingly, when quantifying the distribution of SUMOylation within mouse organs, we observed a much lower percentage of conjugated SUMO2/3 (Fig. 7c). On average, only 52% of SUMO2/3 was conjugated across all organs, with 47% free SUMO2/3, ~1% immature SUMO2, and only a minor fraction of immature SUMO3. SUMO conjugation ranged from above-average rates of 72% and 70% in liver and lung, to below-average rates of 21% and 35% in brain and heart, respectively.

**Fig. 4** Analysis of SUMOylation across mouse organs. **a** Hierarchical clustering analysis of Z-scored label-free quantified (LFQ) expression values corresponding to mouse SUMO2/3 target proteins detected across replicates and organs. Blue coloring indicates relative presence in a sample as compared to others. **b** Principle component analysis (PCA) of all mouse experiments. The principle components represent the greatest degree of variability observed within the data, and grouped experiments are generally more similar than distant experiments. Eigenvalues are displayed on the axes. **c** Qualitative term enrichment analysis comparing SUMO2/3 target proteins between organs, with all SUMO target proteins inferred from MS/MS-identified SUMO site. SUMO2/3 target proteins were compared against corresponding organ background proteomes, and only terms that were significantly enriched are displayed. Black text indicates terms which additionally correspond to functions that were uniquely found to be enriched in one organ, whereas blue text indicates terms that were enriched in more than one organ. Numbers in parentheses indicate overlap between the numbers of SUMO target proteins identified in the organ, compared to those present in the reference. Score was derived from the logarithms of enrichment ratios and Benjamini-Hochberg FDR-corrected two-tailed Fisher's P-values of <0.02. A full list of all enriched terms, and all relevant scores, is available in Supplementary Data 8. **d** Overview of the relative expression levels of SUMO2/3 target proteins identified in each mouse organ, as compared to the expression levels of the same proteins in the other organs. The dotted line represents a value of 1, i.e. no difference compared to other organs. Expression levels (Mouse GeneAtlas V3) were derived from the TISSUES database[41]. Error bars represent SD, $n = 5$ animals

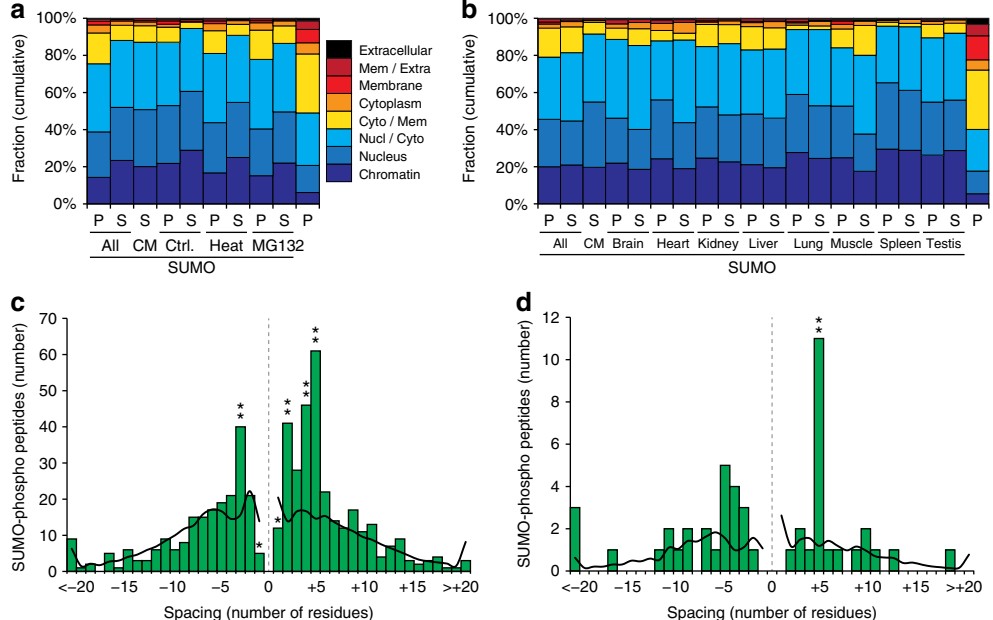

**Fig. 5** SUMO2/3 subcellular localization and SUMO-phospho co-modification across human and mouse. **a** Subcellular localization analysis, based on Gene Ontology Cellular Compartments (GOCC). All SUMO2/3 target proteins (P) and SUMO2/3 sites (S) were assigned a localization based on GOCC terms, ranging from chromatin-associated to extracellular. Subsets of SUMO2/3 target proteins and sites were compared to each other, and to a full human background proteome (right-most value)[29], which represents all proteins present in cells. CM consensus motif (KxE). **b** As (**a**), but for SUMO2/3 target proteins and sites identified in mouse organs. The mouse background proteome (right-most value) was derived from the TISSUES database[41], covers all eight organs, and represents all proteins present in the organs. **c** Schematic overview of SUMO-phospho co-modified peptides identified by MS/MS in human cell lines. SUMOylated lysines are at position 0, with spacing indicating the residue at which phosphorylation occurred. The black line corresponds to the average length profile of all detected SUMO2/3 site peptides. Asterisks denote significantly different values, as determined by two-tailed Fisher's exact testing. * $P < 0.05$, ** $P < 0.001$. **d** As (**c**), but for SUMO-phospho co-modified peptides identified by MS/MS in mouse organs

Correspondingly, in brain, 78% of SUMO2/3 was observed to be unconjugated. Strikingly, the small amount of conjugated SUMO2/3 in brain was found to largely modify the SUMO E1 and E2 enzymes, or integrated into SUMO-chains, with less than half of conjugated SUMO2/3 targeting other proteins. SUMOylation in testis displayed similar patterns, with 35% of SUMO2/3 residing on the E2 enzyme, 22% within SUMO-chains, and only 40% modifying other target proteins. The amount of SUMO-chain formation in testis was significantly higher than in all other organs (Fig. 7c). Outside of brain and testis, SUMOylation predominantly targeted a wider variety of substrates, similar to HEK cells. In lung, we noted a significantly larger SUMO2/3 fraction (4%) to modify SUMO E3 ligases compared to all other organs (Supplementary Fig. 10), and on par with HEK cells. Immature SUMO2 was more abundant in all mouse organs, forming 1.2% of the total pool of SUMO2/3, contrasting HEK cells where immature SUMO3 was more abundant. In heart, significantly more immature SUMO2 was measured in comparison to other organs.

Taken together, we observed considerable differences in the SUMO2/3 equilibrium, highlighting dynamic global requirements for SUMOylation that varied significantly between cultured cell lines and tissues. We validated these findings using immunoblot analyses and a distinct experimental approach (Supplementary Fig. 11 and Supplementary Note 7). Intriguingly, considering our tissue analyses were performed in wild-type mice, our findings support the growing number of observations that SUMO dynamics may be substantially enhanced in many cancers[45] and neurodegenerative diseases[46]. Moreover, our data indicate a requirement for enhanced SUMO2/3 equilibrium in rapidly proliferating cultured cells, which suggests that a high availability and degree of conjugation of SUMO2/3 may occur in, and in turn facilitate, proliferating cancer cells.

**Endogenous SUMO chain topology.** In contrast to the well-described functions of poly-ubiquitylation[47], comparatively little is known about the functional role of poly-SUMOylation in vertebrates[48]. Moreover, SUMO chains have typically been investigated in the context of mutated and overexpressed SUMO. Thus, we decided to investigate SUMO chain-topology under endogenous and in vivo conditions, by extracting abundance profiles for all individual SUMO linkages. In HEK cells, we observed ~35% of SUMOylation to reside on K11 in SUMO2 (Fig. 7d), which was previously reported as the major site for poly-SUMOylation and is the only KxE-type consensus motif in SUMO2. Upon heat or proteotoxic stress, modification of K11 increased further to ~50%. However, in contrast to previous chain-topology studies[2], we observed a notable degree of modification on K7, K21, and K33 in untreated cells. In response to stress, SUMOylation on K21 and K33 was reduced. Modification of SUMO3 generally followed the same trend as SUMO2 (Supplementary Fig. 12A), while for SUMO1, most poly-SUMOylation by endogenous SUMO2/3 occurred via K7, which resides in an inverted ExK-type consensus motif (Supplementary Fig. 12B). Modification on K7 decreased in response to cellular stress, whereas K17, K23, K37, and K45, displayed increased SUMOylation in response to stress. We also observed SUMO-2/3 modification of K78 in SUMO1, with ~10% of all SUMO1 chains occurring via this residue. K78 is the most C-terminal lysine residue in SUMO1, and with no further lysines or arginines towards the C-terminus it is not readily detectable using tryptic approaches[16]. Although our data do not provide direct clues to the functional role of the observed poly-SUMOylation, the chain-topology analysis demonstrates that a wide range of lysine residues within SUMO proteins are involved in endogenous poly-SUMO formation. Moreover, the overall abundance of SUMO residing in poly-chains compared to the overall conjugation of

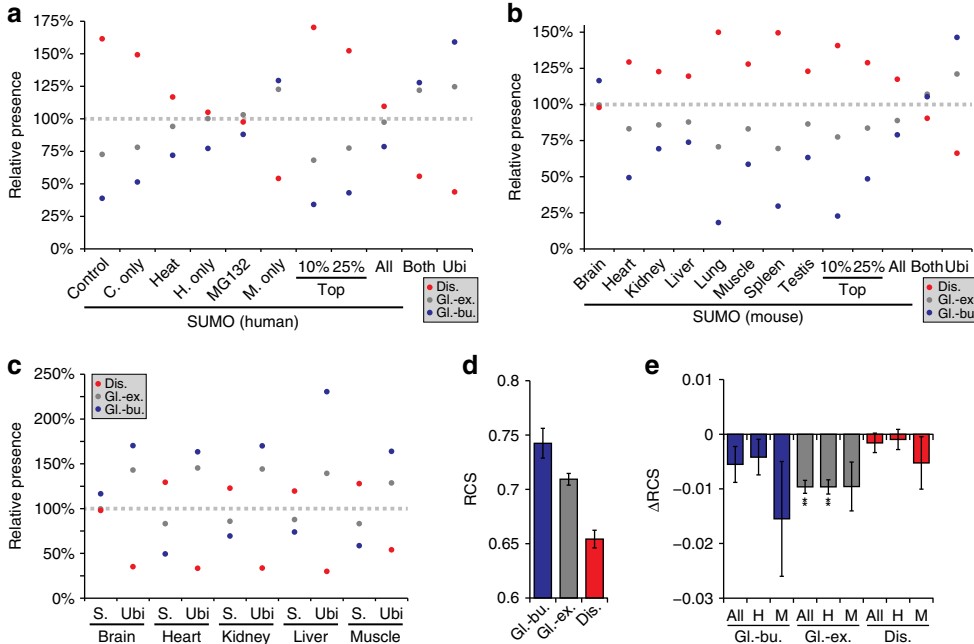

**Fig. 6** Evolutionary conservation of SUMO2/3 structural targeting preferences. **a** Schematic overview of structural properties of subsets of lysines in human SUMO2/3 target proteins, as compared to structural properties of all other lysines within the same proteins. A relative presence of 100% is indicative of no change as compared to the background. Ubiquitin (ubi) sites were derived from PhospoSitePlus (PSP), and only considered if detected in 3 + studies. "Both" corresponds to lysines detected as SUMOylated in this study, and ubiquitylated in PSP data. "C." control, "H." heat, "M." MG132, "Dis." disordered, "Gl.-ex." globular-exposed, "Gl.-bu." globular-buried. **b** As (**a**), but for lysines in mouse SUMO2/3 target proteins. Ubiquitin (ubi) sites were derived from PSP and only considered if detected in 2+ studies. **c** As (**b**), but comparing subsets of SUMOylated lysines to subsets of ubiquitylated lysines detected in the same organs in a ubiquitin proteomics study[59]. **d** Overview of the average Residue Conservation Scores (RCS) for lysine residues situated in all SUMO2/3 target proteins identified in this study. Error bars represent 10xSEM, $n = 16,957$ globular-buried, $n = 124,495$ globular-exposed, and $n = 77,828$ disordered lysine residues. "Dis." disordered, "Gl.-ex." globular-exposed, "Gl.-bu." globular-buried. **e** Average delta RCS values derived from pairwise comparisons of all SUMOylated lysine residues to non-SUMOylated lysine residues within the same proteins and structural context. Error bars represent SD, $n = 846$, $n = 754$, $n = 92$, $n = 8528$, $n = 7716$, $n = 812$, $n = 5679$, $n = 4875$, and $n = 804$ pair-wise comparisons for listed values, respectively. **P < 0.001, determined by two-tailed paired Student's t-testing. "H" human, "M" mouse, "Dis." disordered, "Gl.-ex." globular-exposed, "Gl.-bu." globular-buried

SUMO in cells (Fig. 7b), supported that mono-SUMOylation is the major SUMOylation event in cultured cells.

In mouse organs, we mainly observed SUMO chain formation on SUMO2. Strikingly, we observed a notably different SUMO2 chain topology in mouse organs, with much less modification on the KxE-type K11, while K21 and K33 were observed the most highly modified. With the sequence of SUMO2 fully conserved between human and mouse, this hints at differential regulation of SUMO chain topology in HEK cells compared to organs. Still, the overall relative abundance of poly-chain formation was comparable to HEK cells, suggesting that poly-SUMOylation may play a consistent role across cells and organs. SUMO1 was only found to be modified on K7 and K78, with insufficient data for statistical significance, whereas we could not quantify SUMO3 chain topology in mice owing to C-terminal sequence homology to SUMO2. Moreover, N-terminal quantification of SUMO3 was obfuscated by two isoforms, Q9Z172-2 and G3UZA7, which we found expressed and SUMO-modified in mouse tissues. Interestingly, both of these SUMO3 isoforms differ remarkably more in their sequence from SUMO3 than SUMO2 does, and provide direct evidence for the existence and expression of distinct SUMO family members in mouse organs. Overall, we observed a marked difference in SUMO chain topology between cell culture and mouse organs, with SUMO2 in mice being predominantly modified on lysines other than the canonical K11, and three distinct variants of SUMO3 being expressed and SUMO-modified in mouse organs.

## Discussion

Here, we describe a proteomics strategy facilitating the unbiased and site-specific study of endogenous SUMO2/3, which is compatible with most vertebrate model systems. As a proof of principle, we applied our method in cell culture, identifying in excess of 14,000 SUMOylated lysines, achieving cellular depth similar to previous studies[2], but without the requirement for exogenously expressed and mutated SUMO2/3 or other genetic engineering. Because of this, we could utilize our strategy to profile in vivo SUMOylation across eight types of mouse organs, identifying 1963 unique SUMO2/3 sites.

Technically, our developed approach exhibits a notably increased sequencing sensitivity compared to previous endogenous SUMO methodologies[20,24,49,50], and is the only method that can specifically identify lysine residues modified by SUMO-2/3 (Supplementary Note 8). To achieve this, we utilized the 8A2 antibody to purify SUMOylated peptides, contrary to another method that purified endogenously SUMOylated proteins[20]. We reason that the increased depth achieved by our method is largely owing to efficient purification of peptides instead of proteins, and through purification in very mild conditions after dissolution of lyophilized peptides, allowing the antibody to exert its full specificity.

Recently, a study was published that facilitates identification of endogenously SUMOylated lysines, through digestion of total lysate with the wild-type α-lytic protease (WALP) enzyme followed by di-glycine enrichment[24]. The WALP-only strategy has

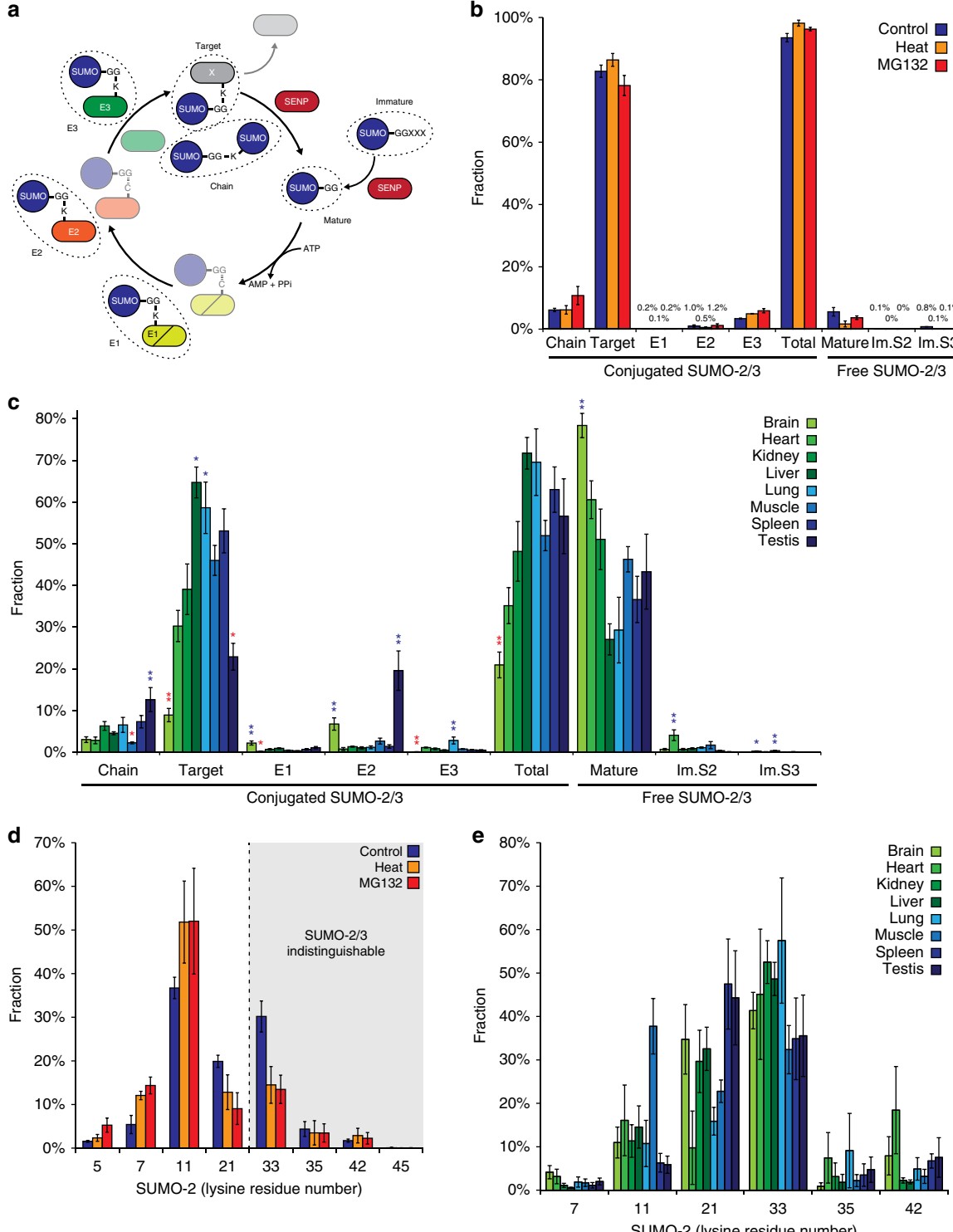

**Fig. 7** Insight into the SUMO equilibrium. **a** Overview of the SUMOylation cycle, highlighting different pools of SUMO2/3 that can be quantified using our purification strategy. Notably, conjugated SUMO2/3 and free SUMO2/3 can be quantified, along with immature free SUMO. Conjugated SUMO2/3 was further split into SUMO covalently conjugated to the E1, E2, or E3 enzymes, to other SUMO (chain), or to any other target protein. Note that the usual transfer of SUMO from E1 to E2 to target requires the cysteine and does not end up on lysines within the same enzymes, but transfer to lysines in SUMO enzymes can occur when the enzymes are active in close proximity to each other. **b** Quantification of the SUMO2/3 equilibrium in human cell lines in response to various treatments, visualizing the fraction of total SUMO existing as conjugated to certain target proteins, or as free SUMO. Error bars represent SD, $n = 3$ cell culture replicates. **c** As (**b**), but quantifying the SUMO2/3 equilibrium in different mouse organs. Error bars represent SEM, $n = 5$ animals. Asterisks indicate significant differences (blue, higher; red, lower) between the indicated organ and the six median organs within the same category, by two-tailed Student's $t$-test. $**P < 0.001$, $*P < 0.05$. **d** Quantification of endogenous SUMO2/3 chain architecture in HEK cells in response to various treatments, corresponding to endogenous SUMO-2 modified by SUMO-2/3. Error bars represent SD, $n = 3$ cell culture replicates. **e** As (**d**), but quantifying the SUMO-2 chain architecture in different mouse organs. Error bars represent SEM, $n = 5$ animals

various shortcomings, including the inability to distinguish SUMO family members and a high potential for false-positive identification of ubiquitylation sites as SUMO[24] (Supplementary Note 8). In contrast to the WALP approach, our methodology validates the presence of the SUMO2/3 mass remnant by its unique fragmentation pattern (Supplementary Data 13 and Supplementary Note 9). Furthermore, our strategy is at least four-fold more sensitive compared to all data reported in other endogenous studies[24,50], while using comparable equipment, a single fraction of a single sample, and only 60 min of MS time (Supplementary Fig. 13).

Biologically, we observed a striking difference between the SUMO2/3 equilibrium in cell culture and organs. In cell culture we found the large majority of SUMO2/3 to be conjugated, whereas considerable pools of free SUMO2/3 were observed in mouse organs. Similar observations have previously been made for ubiquitin, with the majority of ubiquitin conjugated in cell culture, but a large pool of free ubiquitin existing within the brain[51]. Additionally, we observed a much greater amount of total SUMO2/3 in cell culture, relative to the total protein content. Considering SUMO2/3 is known to be a dynamic modifier, and canonically linked to cell-cycle regulation, replication, and the DNA damage response[52], our observations suggest that rapidly and ever-dividing cancer cells rely on the SUMOylation pathway to maintain their genomic integrity. By contrast, cell division is much less prominent in organs, hence SUMO2/3 is not expected to be present or conjugated to a similarly high extent, but instead exists as a free pool to facilitate the response to cellular stress. The high density of SUMO2/3 in cell lines could also be a result of continuous selection pressure, with increased amounts of endogenous and exogenous stresses creating a dependence on SUMOylation to survive. Indeed, overexpression of SUMO-2/3 has previously been shown to increase the speed at which cells divide in culture[35]. Ironically, whereas SUMO2/3 canonically serves to protect faithful multiplication of the cell's genomic content while protecting the cell against stress, it may also ultimately potentiate the rapid and uncontrolled cellular division observed in cancer, and facilitate the dramatic changes observed within the genomic architecture of cancer cells.

Although exhibiting differences in SUMO dynamics and density, the basic targeting preferences of SUMO did not differ between human cells and mouse organs, with SUMOylation predominantly occurring on KxE-type motifs, a preferential occurrence in disordered regions of proteins, and enrichment in the nucleus and similar spatial assemblies (Fig. 8). Still, SUMOylation across organs targeted more specific, often organ-unique functions, although many canonical SUMO2/3 targets were modified across all organs. Similarly, SUMO2/3 predominantly exists as unconjugated in organs, revealing a notable difference in SUMO2/3 equilibrium and dynamics between cultured cells and organs.

Compared to other organs, SUMOylation was more prevalent and more likely to be conjugated instead of free, in liver, lung, spleen, and testis (Fig. 8). In liver, we identified the largest number of sites, although partly because liver was the largest organ analyzed. Nonetheless, SUMOylation appeared to actively play a role in liver, modifying many liver-specific proteins and likely influencing important metabolic pathways. In testis, we observed the highest amount of SUMO2/3 relative to total protein, and a predominant modification of proteins involved in the DNA damage response and replication, which would be supported by the ongoing cell division in the form of spermatogenesis. A high degree of E2 auto-modification and SUMO-chain formation was also found in testis, suggesting that SUMO modification of Ubc9 and SUMO-chain formation may play a critical role in regulation of proteins involved in spermatogenesis.

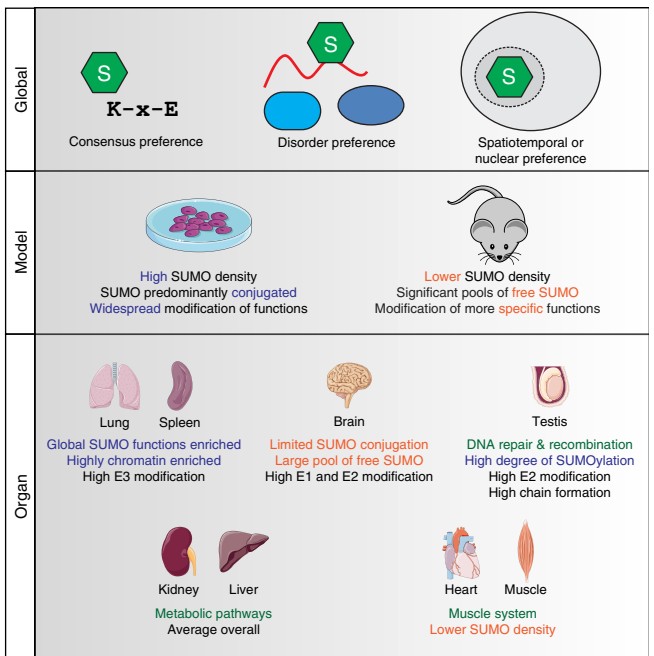

**Fig. 8** Defining the SUMO architecture. Schematic visualization of SUMOylation phenomena observed throughout this study, ranging from global SUMOylation properties, to model-specific and organ-specific preferences. Globally, SUMOylation is preferentially targeted to KxE-type consensus motifs, prefers modifying lysines residing in disordered protein regions, and predominantly modifies proteins localized in the nucleus or otherwise enriched at spatial cellular assemblies such as nuclear bodies, the nuclear pore complex, or at the chromatin. Depending on model system, large variations in the SUMO2/3 equilibrium may be observed, with high amounts of SUMO2/3 and high conjugation rates observed in rapidly dividing cells, and only moderate amounts of SUMO2/3 and significant pools of unconjugated SUMO2/3 observed in normal organs. Whereas SUMO2/3 targets virtually all nuclear cellular functions in cell culture, the SUMO system can be more specifically tuned towards organ-specific functions within specific tissue types. Notably enriched SUMOylated functions were spermatogenesis-related in testis, metabolic pathways in liver and kidney, and muscle system functions in skeletal muscle and the heart

Indeed, SUMO modification of Ubc9 itself has previously been shown to be important during meiosis[53]. Moreover, we found ZNF451, a SUMO E3 ligase that can efficiently extend SUMO chains[54], to be predominantly SUMOylated in testis, suggesting ZNF451 could be responsible for the higher degree of SUMO chains observed in testis. In lung, we observed the highest degree of SUMO E3 auto-modification, often indicative of their activity.

In brain, we detected relatively few SUMOylation events, especially in relation to the size of the organ. Moreover, nearly 80% of SUMO2/3 in brain appeared to exist in free mature state, and more than half of conjugated SUMO2/3 was found in SUMO-chains and on the E1 and E2 enzymes, leaving only 10% of the SUMO2/3 pool conjugated to various target proteins. As SUMOylation commonly is studied in the context of neurological disease, we found this exceptionally striking. A recent publication supports our findings and suggests that SUMO conjugation of synaptic proteins is not only rare, but often false-positively identified[55]. Since we investigated the brain SUMOylome in normal young adult mice, our findings could imply that the SUMOylation system potentially exists in a primed state. This is further supported by the high degree of SUMOylation on the enzymatic machinery, which could be indicative of high levels of SUMO-specific protease activity able to

rapidly reverse SUMOylation on other target proteins. In the context of neurological stress, this standby SUMOylation system would spring into action, potentially through temporary deactivation of SUMO-specific proteases, and thereby mitigate or avert crisis. In support of this hypothesis, the level of conjugated SUMO2/3 is known to increase in the brain during ischemia[22]. Moreover, deactivation of SUMO-specific protease SENP2 in the nervous system has been shown to cause neurodegeneration in mice[56]. However, further research using appropriate model systems will be necessary to consolidate insight into the exact regulatory function of SUMOylation in the context of neurological disease.

Interestingly, the tissue-specific distribution of SUMO2/3 sites in brain diverges from distribution of other PTMs. For example, phosphorylation, glycosylation and ubiquitylation, are most abundantly observed in brain[57–59], where SUMO2/3 exhibited the lowest modification abundance. Our data therefore signify unique differences in the PTM-based cellular physiology of organs as compared to cell culture analyses where most PTMs are observed to be highly expressed. Further discussion relating to SUMO-chain formation, evolutionary conservation of SUMO, and organ-specific SUMOylation patterns, is available in Supplementary Discussion.

Collectively, we present a practical and efficient strategy for mapping the attachment sites of endogenous SUMO2/3 that will help to pave the way for future endogenous and in vivo SUMO2/3 proteomics endeavors, with SUMO2 being the predominantly expressed and most dynamic of all SUMO family members. Additionally, our methodology provides quantitative insight into baseline SUMOylation levels across multiple organs, demonstrating marked differences in SUMO2/3 architecture compared to cultured human cells. Moreover, the unbiased compilation of SUMO2/3 sites identified across eight organs will serve as a valuable resource to the field. With enhanced SUMOylation dynamics emerging as a general hallmark of several cancer types, the ability of our analytical strategy to detail the SUMO architecture in disease-related tissue will ultimately enhance our understanding of the functional role of SUMOylation in health and disease.

## Methods

**Cell culture.** HEK, HeLa, and U2OS cells were obtained from ATCC and routinely tested for mycoplasma infection. The cell lines were not authenticated. Cells were incubated at 37 °C and 5% $CO_2$ in high Glucose GlutaMAX DMEM (Thermo Fisher). Heat shock was performed by incubating cells at 43 °C for 1 h. Proteasome inhibition was performed by treating cells with 10 µM of MG132 (Z-Leu-Leu-Leu-al; Sigma) for 8 h. For HeLa and U2OS cell culture replicates, ~200 million cells were cultured. For HEK cell culture replicates, ~300 million cells were cultured.

**Mice.** Mouse experiments were performed in accordance with the Danish Animal Experiments Inspectorate's guidelines (permit no. 2012-15-2934-00346) and the EC Directive 86/609/EEC for animal experiments. For MS experiments, five three-month old male C57BL/6 mice were euthanized by cervical dislocation, and the brain, heart, kidneys, lung, liver, skeletal muscle, spleen, and testes were immediately excised. Tissues were snap frozen in liquid nitrogen by dropping them into separate 50 mL tubes containing 20 mL of liquid nitrogen, after which the tubes containing the organs were transferred to −80 °C and stored until further processing. For follow-up immunoblot (IB) experiments, two three-month old male C57BL/6 mice were euthanized and processed in similar fashion.

**Lysis of cells.** After treatment, growth medium was decanted and cells were immediately washed twice with ice-cold PBS. Cells were collected by gentle scraping at 4 °C, and pelleted by centrifugation in a swing-out centrifuge at 400 × g. Cells were washed once with ice-cold PBS prior to pelleting them again. Subsequently, cells were lysed in 10 pellet volumes of room temperature Lysis Buffer (6 M guanidine, 50 mM TRIS, pH 8.5), alternating vigorous vortexing and shaking to ensure swift lysis. Cell lysates were immediately snap frozen using liquid nitrogen, and kept at −80 °C until further processing. Lysates were thawed at room temperature, and supplemented with 5 mM chloroacetamide (CAA) and 5 mM Tris(2-

carboxyethyl)phosphine (TCEP), just prior to sonication using 10 s pulses at 30 W. One pulse was used per 10 mL of lysate.

**Homogenization of organs for MS and immunoblot.** Frozen organs were kept on dry ice until seconds before lysis. For MS lysis, four 1.4 mm and two 2.8 mm zirconium beads were added per 2 mL homogenizer tube (Precellys), in addition to 10 µL of 0.5 M freshly dissolved chloroacetamide (CAA). For follow-up IB lysis, 100 µL of 0.2 M freshly dissolved N-ethylmaleimide (NEM) was added per tube instead of CAA. Six tubes were prepared for liver, three tubes for muscle, two tubes for brain, lungs and kidney, and one tube for testes, spleen, and heart. Larger organs were crushed using percussive force while keeping the organs cooled on dry ice, in order to facilitate distribution across multiple tubes. All tubes containing beads, organs or crushed organs, and SUMO protease inhibitors, were kept on dry ice (−80 °C) at all times. Only immediately prior to homogenization through bead-milling, 1 mL of lysis buffer was added to all tubes. For MS sample preparation, this buffer was identical to the one used for cultured cell lysis (6 M guanidine, 50 mM TRIS, pH 8.5). For follow-up IB sample preparation, the lysis buffer was composed of 2% SDS, 50 mM TRIS pH 8.5, 150 mM NaCl, freshly supplemented with broad-range protease inhibitor cocktail (cOmplete™ Mini, Roche) and 20 mM N-ethylmaleimide; essentially as described previously[26]. Bead-milling was performed using a Precellys 24 homogenizer system, at 5500 rpm for 20 s, with organs typically already homogenized in several seconds. After each grinding cycle, tubes were briefly centrifuged at 500 × g, and subjected to another round of grinding in case any large particulate matter remained. When lysing in guanidine (for MS), this was only observed while homogenizing skeletal muscle. When lysing in SDS (for IB), this was observed while homogenizing testis, spleen, heart, and skeletal muscle. After grinding, lysates were centrifuged at 500 × g for 30 s and transferred to 15 mL tubes containing one additional volume of lysis buffer, in order to separate homogenate from the zirconium beads. Protein concentration was determined using standard Bradford assays for MS samples, and lysates were diluted to 5 mg/mL using lysis buffer. For follow-up IB samples, standard BCA assays were used instead because Bradford is incompatible with high concentrations of detergent. Diluted lysates were subjected to 10 s pulses of sonication at 30 W, with one pulse used per 10 mL of total lysate volume. Following sonication, MS lysates were supplemented with 5 mM CAA and 5 mM TCEP, and incubated at 37 °C for 30 min. Reduction and alkylation was not performed for follow-up IB samples, although alkylation by NEM is likely to have occurred during bead-grinding in the presence of a high concentration of NEM. Next, lysates were centrifuged for 30 min at 3000 × g in a swing-out centrifuge with delayed deceleration. No discernable pellets were observed after centrifugation, but rather for some organs debris was observed to float at the top of the samples. Thus, centrifuged lysates were carefully decanted into new tubes, and subsequently passed through 0.45 µm disc filters in order to clarify them. For follow-up IB analysis (Supplementary Fig. 11), HEK cells were lysed similarly to mouse organs, by pelleting the HEK cells in homogenizer tubes and snap freezing them with liquid nitrogen, prior to subjecting them to the exact same lysis procedure as the mouse organs.

**MS lysate digestion and peptide purification.** From this point, the MS samples preparation protocol is the same for cell line and tissue samples. Homogenized lysates were digested by addition of Lys-C (Wako) to a 1:200 enzyme-to-protein ratio, overnight and at room temperature. Partially-digested lysates were diluted with 3 volumes of 50 mM ice-cold ammonium bicarbonate, gently mixed, and allowed to warm up to room temperature. Subsequently, another round of Lys-C digestion was performed with a 1:200 enzyme-to-protein ratio, overnight and at room temperature. Digested lysates were acidified by addition of trifluoroacetic acid (TFA) to a final concentration of 0.5%. Acidified digests were centrifuged at 3000 × g in a swing-out centrifuge, for 30 min at 4 °C. Next, the cleared digests were carefully decanted into new tubes. Peptides were purified using C8 Sep-Pak cartridges (Waters) according to the manufacturer's instructions, using one 500 mg sorbent Sep-Pak cartridge per 25 mg of digested protein. Prior to elution of peptides, Sep-Pak cartridges were pre-eluted using 5 mL of 20% acetonitrile (ACN) in 0.1% TFA, and 3 mL of 25% ACN in 0.1% TFA. Peptides were sequentially eluted using 1 mL 35% ACN in 0.1% TFA, 1 mL 40% ACN in 0.1% TFA, and 2 mL 45% ACN in 0.1% TFA, after which all elutions were pooled for each replicate. Elutions were transferred to 50 mL tubes with punctured caps, and frozen overnight at −80 °C. Deep-frozen eluted peptides were lyophilized until complete dryness, over a period of 96 h.

**Preparation of SUMO-IP beads.** SUMO-IP beads were prepared up to 1 week prior to performing the IP. A volume of 1 µL of SUMO-IP beads was used per 1 mg of starting protein material, for all IPs performed. To prepare one batch of SUMO-IP beads, 750 µL of Protein G Agarose beads (Roche) were washed 4× with PBS and transferred to 1.5 mL tubes, after which 500 µL SUMO-2/3 antibody (8A2, acquired from Abcam, ab81371; ~5–10 µg/µL antibody) was added. Tubes were filled fully by addition of PBS, and subsequently beads and antibody were incubated in a mixer at 4 °C for 1 h, after which the beads were washed 3× with ice-cold PBS. Next, antibody was crosslinked to the beads by addition of 0.2 M sodium borate, pH 9.0, freshly supplemented with 20 mM dimethyl pimelimidate (DMP), and

incubated in a mixer for 30 min at room temperature. The crosslinking step was repeated once, using freshly dissolved DMP. Subsequently, beads were washed twice with ice-cold PBS, twice with room temperature 0.1 M glycine pH 2.8, and twice with ice-cold PBS. Finally, SUMO-IP beads were stored until use, at 4 °C in PBS supplemented with 5 mM sodium azide.

**SUMOylated peptide purification**. Lyophilized peptides were dissolved in ice-cold SUMO-IP Buffer (50 mM MOPS, 10 mM Na2HPO4, 50 mM NaCl, pH 7.2), using 1 mL SUMO-IP Buffer per 5 mg of protein initially in the sample. Samples were cleared by centrifugation at $3000 \times g$ for 30 min at 4 °C, and transferred to 15 mL tubes, with no more than 10 mL of samples per tube. A volume of 50 µL of SUMO-IP beads were added per 10 mL of sample, after which the sample was mixed for 3 h at 4 °C. Beads were briefly pelleted, transferred to 1.5 mL tubes using ice-cold SUMO-IP buffer, and washed once more with SUMO-IP buffer. Next, beads were washed with ice-cold PBS, transferred to clean 1.5 mL LoBind tubes (Eppendorf), and washed once more with PBS. Subsequently, beads were washed with ice-cold MQ water, transferred to clean 1.5 mL LoBind tubes, and washed once more with MQ water. Peptides were eluted from the beads using two volumes of ice-cold 0.15% TFA in MQ water, by gently mixing the beads and allowing them to stand on ice for 30 min. The elution was repeated once, and both elutions were pooled. Elutions were cleared by centrifugation through 0.45 µm filters, and pH-neutralized by addition of 1/10th volume of 1 M Na2HPO4. Eluted SUMOylated peptides were frozen at −80 °C until further processing.

**Second-stage digestion of SUMOylated peptides**. The purified Lys-C-digested SUMO2/3 peptides were further digested with various enzymes. All mouse samples and the majority of all cell line samples were further digested with Asp-N (Roche), and some initial cell line samples were digested with either trypsin (Thermo), Glu-C (Promega), or wild-type alpha-lytic protease (WALP; Sigma). Asp-N, Glu-C, and WALP digestions were performed in the neutralized SUMO-IP elutions, overnight and at 30 °C, using 1 µg of enzyme per 50 mg of protein initially in the sample. For trypsin digestion, urea was added to a final concentration of 2 M. One microgram of trypsin per 50 mg of protein initially in the sample was used, digesting overnight and at room temperature.

**Desalting and on-StageTip high-pH fractionation**. C18 StageTips were prepared in-house[2,60]. For all experiments, four plugs of C18 material (Sigma-Aldrich, Empore™ SPE Disks, C18, 47 mm) were layered per StageTip. For single-shot samples desalted at low pH, StageTips were activated with 100 µL 100% methanol, equilibrated with 100 µL 80% ACN in 0.5% acetic acid, and equilibrated twice with 50 µL 0.5% acetic acid, prior to loading samples. After loading, samples were washed twice with 50 µL 0.5% acetic acid prior to elution. Samples from initial experiments with trypsin, Glu-C, and WALP were desalted at low pH (0.5% acetic acid) and eluted as a single sample with 50% ACN in 0.5% acetic acid. The majority of samples prepared with trypsin, Asp-N, and WALP, were high-pH fractionated on StageTip. For this, StageTips were conditioned with 100 µL methanol, equilibrated with 100 µL 80% ACN in 200 mM ammonium hydroxide, and washed twice with 75 µL 50 mM ammonium hydroxide. The pH of the digested samples was raised by addition of ammonium hydroxide to a final concentration of 20 mM. Samples were loaded on StageTips, and washed twice with 75 µL 50 mM ammonium hydroxide. Subsequently, six fractions (F1-6) were eluted by sequential elution with increasing amounts of ACN in 20 mM ammonium hydroxide. For Asp-N digests, elutions were performed with 4, 7, 10, 13, 17, and 25% ACN. For trypsin digests, elutions were performed with 5, 8, 12, 18, 27, and 50% ACN. For WALP digests, elutions were performed with 2, 4, 7, 10, 13, and 40% ACN. Optimization was performed on earlier experiments, and these fractions represent the finalized profiles used for the majority of the samples. All fractions were completely dried in a SpeedVac at 60 °C. Dried peptides were dissolved in 10 µL 0.1% formic acid, and frozen at −20 °C until analysis.

**Mass spectrometric analysis**. Samples were analyzed on 15 cm long 75 µm internal diameter columns, packed in-house with ReproSil-Pur 120 C18-AQ 1.9 µm beads (Dr. Maisch), connected to an EASY-nLC 1200 system (Thermo). Column elution was performed using a multitude of different gradients, owing to considerable differences in peptide properties depending on high-pH fraction and the enzymes used for digestion of the peptide mixtures. Settings were optimized during earlier experiments, and the following values represent the finalized settings used for the large majority of all samples, and for all samples that were quantified. Single-shot samples were analyzed with 140 min effective gradient time, and fractions with 70 min effective gradient time, with Buffer A (0.1% FA) as the initial buffer, and an increasing amount of Buffer B (80% ACN in 0.1% FA) over time. The following values represent the increasing value of Buffer B across the largest time-frame of the analytical gradients, and do not include initial ramp-up and washing blocks. Single-shot analyses; Glu-C: 5–30%, trypsin: 25–45%, WALP: 5–20%. Fraction analyses; Asp-N; F1: 13–24%, F2: 14–27%, F3–5: 15–30%, F6: 17–32%. Trypsin; F1–6: 26–42%. WALP; F1–2: 5–20%, F3–6: 5–30%. A column heater was used to heat the column to 40 °C, and ionization was performed using a Nanospray Flex Ion Source (Thermo). Analysis of the ion stream was performed using a Q-Exactive HF mass spectrometer (Thermo). Spray voltage was set to 2 kV,

with a capillary temperature of 275 °C, and an S-Lens RF level of 50%. Full scans were made at a resolution of 60,000, with an AGC target of 3,000,000 and a maximum injection time of 60 ms. The following scan ranges were used; Asp-N: 400–1600 *m/z*, Glu-C: 300–1750 *m/z*, trypsin: 600–1900 *m/z*, WALP: 300–1200 *m/z*. Precursor fragmentation was achieved through higher-energy collision dis-association (HCD) with a normalized collision energy of 25, an AGC target of 100,000, and an isolation width of 1.3 *m/z*. The following precursor charge states were considered for MS/MS; Asp-N: 2–6, Glu-C: 2–7, trypsin: 3–6, WALP: 2–5. A dynamic exclusion of 60 s and 45 s was used for single-shot samples and fractions, respectively. For collection and analysis of precursors by MS/MS, loop count was set to 7, MS2 resolution to 60,000, maximum injection time to 120 ms, and intensity threshold to 5000.

**Raw data analysis**. MS proteomics RAW data are available at the ProteomeXchange Consortium database via the Proteomics Identifications (PRIDE) partner repository[61], under dataset ID PXD008003. All RAW files were analyzed using MaxQuant software version 1.5.3.30[62,63]. For the human cell line data, data from different enzymes was analyzed in separate searches. All mouse data were processed together in a single search. Additional searches were performed for initial comparison between different enzymes; however, results from these were not included in the final processed databases. Default MaxQuant settings were used, with exceptions and important settings outlined below. For generation of the theoretical peptide library, the FASTA databases were downloaded from UniProt[64]. The human database was downloaded on 22 February 2017, and the mouse database on 10 May 2017. Label-free quantification (LFQ) was enabled. For all searches, protein N-terminal acetylation and methionine oxidation were included as potential variable modifications (default), in addition to phosphorylation on serine, threonine, and tyrosine. SUMO2/3 mass remnants were defined as follows. Asp-N; DVFQQQTGG, $H_{60}C_{41}N_{12}O_{15}$, monoisotopic mass 960.4301, neutral loss b7-DVFQQQT, diagnostic mass remnants [b2-DV, b3-DVF, b5-DVFQQ, b6-DVFQQQ, b7-DVFQQQT, b9-DVFQQQTGG, QQ, FQ, FQQ]. Glu-C, variant 1; VFQQQTGG, $H_{49}C_{33}N_9O_{10}$, monoisotopic mass 731.3602, neutral loss b6-VFQQQT, diagnostic mass remnants [b2-VF-CO, b2-VF, b3-VFQ, b4-VFQQ, b5-VFQQQ, b6-VFQQQT, b7-VFQQQTG, b8-VFQQQTGG]. Glu-C, variant 2; DTIDVFQQQTGG, $H_{77}C_{51}N_{13}O_{19}$, monoisotopic mass 1175.5459, neutral loss b10-DTIDVFQQQT, diagnostic mass remnants [b2-DT, b3-DTI, b4-DTID, b5-DTIDV-H2O, b5-DTIDV, b6-DTIDVF, b7-DTIDVFQ, b8-DTIDVFQQ, b9-DTIDVFQQQ, b10-DTIDVFQQQT]. Glu-C, variant 3; MEDEDTIDVFQQQTGG, $H_{105}C_{70}N_{17}O_{29}S_1$, monoisotopic mass 1679.6985, neutral loss b14-MEDEDTIDVFQQQT, diagnostic mass remnants [b2-ME, b3-MED, b4-MEDE, b5-MEDED, b6-MEDEDT, b6-MEDEDT-H2O, b7-MEDEDTI, b7-MEDEDTI-H2O, b8-MEDEDTID, b8-MEDEDTID-H2O]. Glu-C, variant 3, oxidized; same as above but with one additional oxygen atom. Trypsin, variant 1; FDGQPINETDTPAQLEMEDEDTIDVFQQQTGG, $H_{218}C_{146}N_{36}O_{58}S_1$, monoisotopic mass 3435.4936, neutral loss b30-FDGQPI-NETDTPAQLEMEDEDTIDVFQQQT, diagnostic mass remnants [b4-FDGQ, b6-FDGQPI, b7-FDGQPIN, b8-FDGQPINE, b10-FDGQPINETD, b11-FDGQPI-NETDT, b13-FDGQPINETDTPA, b14-FDGQPINETDTPAQ, b15-FDGQPI-NETDTPAQL, b16-FDGQPINETDTPAQLE]. Trypsin, variant 2; FRFDGQPINETDTPAQLEMEDEDTIDVFQQQTGG, $H_{239}C_{161}N_{41}O_{60}S_1$, mono-isotopic mass 3738.6632, neutral loss b32-FRFDGQPINETDTPAQLEME-DEDTIDVFQQQT, diagnostic mass remnants [b6-FRFDGQ, b12-FRFDGQPINETD, b13-FRFDGQPINETDT, b16-FRFDGQPINETDTPAQ, b17-FRFDGQPINETDTPAQL, b18-FRFDGQPINETDTPAQLE, b24-FRFDGQPI-NETDTPAQLEMEDEDT, b25-FRFDGQPINETDTPAQLEMEDEDTI, b26-FRFDGQPINETDTPAQLEMEDEDTID, b27-FRFDGQPINETDTPAQLEME-DEDTIDV]. The WALP mass remnant was searched as di-glycine, as it is identical to the tryptic ubiquitin mass remnant. Enzyme cleavage specificity was set as follows; Asp-N: C-term K and N-term D, E, up to 8 missed cleavages. Glu-C: C-term K, D, E, up to 8 missed cleavages. Trypsin: C-term K, R, up to 4 missed cleavages. WALP: C-term K, V, A, T, S, L, G, up to 10 missed cleavages. Minimum peptide length was set to 7 amino acids, and maximum peptide mass was set as follows; Asp-N: 6000 Da, Glu-C: 7000 Da, trypsin: 9000 Da, WALP: 3200 Da. Matching between runs was performed with a match time window of 2 min and an alignment time window of 40 min. We performed a global validation of the matching between runs performance, which is described in Supplementary Note 10. Data was filtered for peptide-spectrum-match and protein assignment by posterior error probability to achieve a false discovery rate of <1% (default), with a recalibrated mass tolerance of 4.5 ppm (default). Automatic filtering of modified peptides was performed through application of a site decoy fraction of 2%, an Andromeda score cut-off of 40 (default), and a stringent delta score cutoff of 20.

**Manual filtering of MaxQuant output tables**. The MaxQuant output tables were additionally manually filtered for the various score values reported by Max-Quant[62,63], requiring a delta score of >40 for multiply modified peptides, a localization delta score of >6 for all modified peptides, the absence of a reversed database hit, and the presence of diagnostic mass remnant fragments in the MS/MS spectra corresponding to Asp-N, Glu-C, and trypsin samples. Trypsin, Glu-C and WALP-derived SUMO sites were not allowed on peptide C-terminal lysines. Asp-N-derived SUMO sites were allowed on peptide C-terminal lysines if the next

residue was an aspartic acid or glutamic acid. Duplicate assignments of MS/MS scans to multiple SUMO sites, and duplicate ±25 amino acid sequence windows flanking SUMO-modified lysines, were manually discarded. Stringent manual filtering resulted in discarding 13% of sites from the human Asp-N data, and 18% of sites from the mouse Asp-N data. SUMOylated proteins were reconstructed based on the "proteinGroups.txt" file, and only sites that remained after manual filtering were mapped back to the proteins. SUMO target proteins were defined as those with at least one SUMOylated peptide, and other proteins were discarded from further analysis. Data pertaining to tissue-specific protein expression levels in mouse were extracted from the TISSUES database[41].

**Identification of SUMO-phospho co-modified peptides.** The "evidence.txt" output file from MaxQuant was used to filter for SUMO-phospho co-modified peptides, and only direct MS/MS evidence was considered. A localization delta score of >6 was demanded for both SUMOylation and phosphorylation. A combined score was calculated from the excess score over Andromeda, delta, and localization scores, primarily weighing the delta score, and co-modified peptides were filtered for a combined score of >20 (i.e. a delta score of >40), applying an additional layer of FDR over the MaxQuant default. The highest-scoring unique modified peptides were retained in the list, and 51 amino acid sequence windows were assigned to both SUMOylation and phosphorylation. Quantitative information was extracted from modificationSpecificPeptides.txt and aligned to the corresponding co-modified peptides.

**Quantification and scoring of SUMO sites and proteins.** The $n = 3$ HEK data (cell culture replicates) and $n = 5$ mouse data (five animals) were quantified, using MaxQuant LFQ intensities, and requiring at least 2 peptides for quantification of proteins. Mouse data were median-normalized to compensate for variations in overall protein abundance between different organs. For the HEK data, proteins were only quantified if detected in 3 out of 3 replicates in at least one condition. For the mouse data, proteins were only quantified if detected in at least 4 out of 5 replicates in at least one organ. After filtering, missing values were globally imputed using Perseus software[65]. Sites were processed and quantified analogously to the proteins, for both HEK and mouse data. To find significant differences in response to stress in the HEK data, 2-log transformed values were subjected to two-sample testing in Perseus, with a permutation based FDR cutoff of 5% and a p0 value of 1. To find significant differences between organs in the mouse data, 2-log transformed values were subjected to ANOVA testing in Perseus, with a permutation based FDR cutoff of 5% and a p0 value of 1. To visualize relative abundance of SUMO sites and proteins, Z-scoring was performed and averaged across same-condition replicates. For ranking of SUMO target proteins, a score was derived from the number of SUMO sites identified in the protein, the average score of the sites in the protein, the overall intensity of all SUMO sites, and the fraction of SUMO site intensity compared to total protein intensity (purity). For ranking of human SUMO sites, a score was derived from the number of MS/MS scans identifying the site, site intensity under standard growth conditions, Andromeda score, delta score, and the localization delta score. For mouse SUMO sites, the number of replicates identifying the site was additionally factored into the score.

**Statistical analysis and data visualization.** All IceLogos were generated using the IceLogo v1.2 tool[66]. For all IceLogos, reference data sets were generated based on all proteins annotated as nuclear-localized on Uniprot, for both human and mouse data separately. For Motif-X analyses, the online tool (http://motif-x.med.harvard.edu) was used[67], with pre-aligned foreground, lysine as the central character, a width of 11, and otherwise default settings. Heatmaps were generated using hierarchical clustering as integrated in Perseus software, using Z-scored values as input. Principle component analyses were generated using Perseus, using Z-scored values as input. All term enrichment analyses were performed using Perseus, by annotating reference lists or subsets of proteins with annotation terms, and performing two-tailed Fisher's testing. The categories used for annotation of proteins were Gene Ontology (GO) Biological Processes (GOBP), GO Cellular Compartments (GOCC), GO Molecular Functions (GOMF), Kyoto Encyclopedia of Genes and Genomes (KEGG), Gene Set Enrichment Analysis (GSEA), Comprehensive Resource of Mammalian protein complexes (CORUM), Protein families database (Pfam), and a set of general keywords. Observed differences were filtered for a Benjamini–Hochberg multiple-hypotheses corrected P-value of <0.02, with P-values acquired through two-tailed Fisher's testing. Relative scores, only used for the purpose of intuitive data visualization, were calculated based on the negative base 10 logarithm of the enrichment ratio between observed and expected adherence to terms, and on the base 2 logarithm of the FDR-corrected P-value. The exact formula used was $(Log_2(enrichment)^3 \times -Log_{10}(P\text{-value}))^{0.25}$, after which scores were normalized to 100% for the maximum value.

**Quantification of conjugated and free pools of SUMO.** The mature sequence of SUMO2 was inserted into the FASTA to allow detection of free mature SUMO2/3 in the MaxQuant searches. For quantification of pools of SUMO, the "evidence.txt" MaxQuant output file was used, and all peptides either modified by SUMO2/3, or peptides derived from SUMO2/3 itself, were considered. Only evidence obtained from Lys-C/Asp-N experiments was used. Modification of peptides by SUMO was

sub-classed into targeting SUMO itself (chain formation), the E1 enzyme subunits (SAE1 and UBA2), the E2 enzyme (UBC9), the E3 enzymes (any protein name containing "E3 SUMO ligase"), or otherwise conjugation to other targets. Peptides derived from SUMO2/3 were sub-classed as internal, mature free SUMO2/3, immature SUMO2, or immature SUMO3. Internal peptides were considered unknown, as they could originate from any state of SUMO, and not used for the quantification. Peptides ending in QQTGG (predominantly DVFQQQTGG) were considered as mature free SUMO2/3. Peptides containing but not ending with QQTGG were considered as immature SUMO2 (human/mouse: DVFQQQTGGVY), or immature SUMO3 (human: DVFQQQTGGVP and DVFQQQTGGVPESSLAGHSF, mouse: DVFQQQTGGSASRGSVPTPNRCP). Intensities for each group were summed separately for individual replicates, and fractions were calculated from the summed intensities and used for averages, standard deviations, and Student's two-tailed t-testing.

**Quantification of SUMO chains.** For SUMO chain quantification, intensity values were directly taken from the evidence.txt file, and only evidence from Lys-C/Asp-N experiments was considered. SUMO-modified peptides originating from SUMO1, SUMO2, and SUMO3, were isolated and binned based on modified lysines. In case multiple lysines were simultaneously modified, the peptide intensity was added to each site. Summed site intensities were converted to fractional values within each replicate, and fractional values were averaged across replicates and used for further statistical calculations.

**Structural prediction.** 3870 human and 955 mouse SUMO target proteins were subjected to structural predictive analysis. The IUPred workflow was used[68] to determine disordered (D-type) or globular (G-type) regions in the proteins. the ACCpro workflow was used[69] to determine solvent-exposed (E-type) or buried (B-type) regions in the proteins. All disordered lysines were considered exposed, resulting in three classifications for all lysines: D-type, globular/exposed (GE-type) and globular/buried (GB-type).

**Evolutionary conservation analysis.** Human and mouse Uniprot accession numbers were mapped to ENSP identifiers using "full_uniprot_2_string.04_2015.tsv" derived from STRING[70], in addition to the "Retrieve/ID mapping" functionality from Uniprot[64]. Multiple sequence alignments (MSA) of the mammalian orthologous group (OG), their corresponding ENSP identifiers, and fine-grained orthologs (FGO), were retrieved from eggNOG[71]. ENSP identifiers were mapped to eggNOG group names via "maNOG.members.tsv", and FGO pairs between human and mouse were retrieved via the eggNOG REST API, thereby mapping Uniprot to ENSP identifiers and determining fine-grained orthologous proteins between human and mouse. In order to obtain a metric for evolutionary conservation, Residue Conservation Scores (RCS) were calculated[72], using MSA derived from eggNOG and the following parameters; method was set to "trident", matrix to "blosum62", diversity to 1, chemistry to 0.5, and a gap penalty of 2 was used. This resulted in RCS values for all amino acid residues within the MSA. In order to evaluate differences between evolutionary conservation of SUMOylated and non-SUMOylated lysine residues, we performed a two-tailed paired Student's t-test on the aforementioned RCS values, while correcting for disordered, globular-exposed, and globular-buried regions. P-values <0.001 were deemed significant.

**Comparison of proteomic SUMO studies.** In order to align human SUMO target proteins, a scaffold was created based on all unique protein-coding genes as downloaded from UniProt on 16 July 2017. Protein identifier and protein names were imported, along with basic protein size and mass information. SUMO target proteins identified in this study were aligned to the scaffold, along with SUMO target proteins identified in other SUMO proteomics studies[2,19,20,24,28,30,34–40], and proteins identified in two total proteome studies[29,73]. Alignment was primarily performed using Uniprot identifier, and otherwise protein-coding gene. For comparison of human SUMO sites, a scaffold was generated from all 51 amino acid sequence windows derived from this study, as well as 51 amino acid sequence windows identified in other SUMO proteomics studies[2,19,24,28,30,33–39]. Duplicate 51 amino acid sequences were excluded, and afterwards all studies were individually mapped to the scaffold. Qualitative and quantitative information from this study, and from one previous SUMO proteomics study, was aligned to the database. Information on low-throughput-identified SUMO sites, ubiquitylation sites, acetylation sites, and methylation sites, was extracted from PhosphoSitePlus on 19 July 2017, and aligned to the database.

**Immunoblot and dot blot analyses.** Dot blot (slot blot) analyses were performed using a Slot Blot Manifold PR648 system (Thermo) connected to a vacuum pump. To assist in capture and visualization of peptides and small proteins, supported nitrocellulose membranes with 0.2 μm pore-size were used (Protran, Amersham) in the dot blot system. Samples were loaded and subsequently transferred onto the membranes by application of a mild vacuum for ~15 s. For 1D gel electrophoresis, samples were supplemented with 1/3rd volume of NuPAGE™ LDS Sample Buffer (4×), dithiothreitol was added to a final concentration of 100 mM, and samples were incubated at 70 °C for 10 min prior to loading. Peptides were separated on NuPAGE 4–12% Bis-Tris gradient gels (Thermo), using MES buffer to assist

separation of peptides and small proteins. Proteins were separated on the same gradient gels, but using MOPS buffer. Electrophoretic separation was achieved using XCell SureLock™ Mini-Cell Electrophoresis systems (Thermo), set to 80 V for 10 min and then 200 V until the loading dye front reached the bottom of the gel (~45 min). To assist in capture and visualization of peptides and small proteins, supported nitrocellulose membranes with 0.2 μm pore-size were used (Protran, Amersham) for immunoblot transfer. Subsequently, XCell SureLock™ Mini-Cell Electrophoresis systems were used to transfer peptides or proteins from the gels to the nitrocellulose membranes. For peptides, transfer buffer contained 25 mM TRIS, 192 mM glycine, and 20% methanol. For proteins, transfer buffer contained 50 mM TRIS, 384 mM glycine, and 10% methanol. For peptides, transfer was performed ice-cold for 1 h at 20 V and 2 h at 25 V. For proteins, transfer was performed ice-cold for 1 h at 25 V, 1 h at 30 V, and 1 h at 35 V. At this stage, membranes acquired either through dot blot (vacuum) transfer or electrophoretic transfer, were handled similarly. Membranes were rinsed with MQ water, and subsequently incubated with Ponceau-S solution (Sigma) for 1 min. After incubation, membranes were rinsed three times with MQ water and kept in 0.1% acetic acid to stabilize Ponceau-S stain during scanning of the membranes. Subsequently, Ponceau-S was rinsed off the membranes by incubation in PBS supplemented with 0.1% Tween-20 (Sigma); PBST. Membranes were blocked for 1 h at room temperature using 8% (w/v) skim milk powder in PBST (8% milk), after which incubation with primary antibody occurred overnight in 8% milk at 4 °C. Subsequently, membranes were washed three times 10 min with ice-cold PBST, re-blocked for 30 min with 10% milk at 4 ° C, incubated with secondary antibody in 10% milk at 4 °C, washed three times 20 min with ice-cold PBST, and washed three times 10 min with ice-cold PBS. Visualization was achieved using enhanced chemiluminescence kits (Novex) according to the manufacturer's instructions. Chemiluminescence was captured using Amersham Hyperfilm ECL (Sigma). Anti-SUMO2/3 8A2 antibody (Abcam, ab81371) was used as the primary antibody at a 1:2500 concentration in all cases. Goat-anti-mouse HRP conjugated secondary antibody (Jackson Immunoresearch, 115–035–003) was used at a concentration of 1:2500. All scans of immunoblots and Ponceau-S-stained membranes displayed in this manuscript are uncropped and display the full molecular weight range.

**Data availability**. Mass spectrometry RAW data are available at the ProteomeXchange Consortium database via the Proteomics Identifications (PRIDE) partner repository, under dataset ID PXD008003. Other data can be obtained from the corresponding author upon reasonable request.

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

## Acknowledgements

The work carried out in this study was in part supported by the Novo Nordisk Foundation Center for Protein Research; the Novo Nordisk Foundation (grant agreement number NNF14CC0001 and grant agreement number NNF13OC0006477); The Danish Council of Independent Research (grant agreement number DFF 4002-0005 and grant agreement number DFF 4183-00322A). I.A.H. is supported by the European Molecular Biology Organization (grant number ALTF 503-2016). We thank members of the NNF-CPR for fruitful discussions, J. Lukas for critical reading of the manuscript, S. Schopper, J. Madsen and M. Rykær at the NNF-CPR Mass Spectrometry Platform for instrument support and technical assistance, and E. Villanueva for assistance with mouse handling.

## Author contributions

I.A.H. and M.L.N conceived the project, and I.A.H optimized the endogenous SUMO-IP workflow, and prepared all cell culture samples. S.D., N.H.S., and J.A.D. provided and handled mice. I.A.H. performed all experiments, optimized MS methodology, measured all samples on MS, and processed all MS raw data. D.L. and L.J.J. performed structural predictions and evolutionary conservation analyses. I.A.H. and M.L.N. performed bioinformatics analyses. M.L.N. supervised the project. I.A.H. and M.L.N. wrote the manuscript with input from all authors.

## Additional information

**Competing interests:** The authors declare no competing interests.

