## [Peer Review File · Nature Communications]

Reviewers' comments:

Reviewer #1 (Remarks to the Author):

The current study by Hendriks and colleagues present a novel proteomic resource of SUMO2/3 conjugated proteins found in a human-derived cell line and mouse tissues. The study describes the combination of an anti-SUMO2/3 immunoprecipitation with a mass spectrometric approach that enables enrichment and identification of endogenous SUMO2/3 targeted lysines in proteins from a human cell line and various mouse tissues. While such large proteomics resources were already available for multiple cell lines, here the authors went further and established a novel method that allows for the identification of SUMO2/3-targeted lysines in various mouse tissues, aiming to deepen our current understanding of the roles of SUMO2/3 conjugation in vivo.

The amount of data generated in the Hendriks et al. study is enormous, and the impact for the field will be, without doubt, major. After careful characterization of their protocol using HEK cells, the authors applied their method to enrich and identify SUMO2/3 conjugates from 8 different mouse tissues and reached a resolution never achieved so far. They not only mapped endogenous SUMO2/3 acceptor sites in various mouse tissues, but they also made use of sophisticated bioinformatical tools to address major questions in the SUMO field. They deciphered species and organ-specific SUMO2/3 proteomes, the sub-cellular preferences of SUMO2/3 conjugates, addressed cross-talk with phosphorylation, proteins structural preferences of the SUMO2/3-targeted sites, and evolutionary conservation among the SUMO2/3 proteins substrates.

Interestingly, their approach also reveals the dynamics of endogenous SUMOylation by quantifying the equilibrium of free versus bound SUMO2/3, and by providing a description of the topology of SUMO2/3 chains in vivo and in vitro. In sum, this paper de-coded the SUMO2/3-ome in mice. As such, I highly recommend the study by Hendriks et al. to be published in Nature Communications. The present study is convincing and will become influential in the SUMO field and in other field of science.

The authors now face the challenges of making such an immense repertoire of information easily accessible to scientists of other disciplines. Further, the authors should highlight the usefulness of their method to other labs. In this context, one main point, and a few minor issues should be address by the authors.

Major Point

The authors established the catalogue of SUMO2/3 substrates in 8 different mice organs and attempt to describe an organ-specific function of SUMO2/3 targets. To do so, they compare the SUMO2/3 targets found in the organ in questions to the targets found (and not found) in other organs. In the figure legend from Figure 3C, the authors say that "SUMO target proteins were considered enriched in an organ if significantly different from other organs". Consequently, in the brain, SUMOylated synaptic proteins are enriched as compare to other organs, and so are SUMOylated enzymes involved in metabolic processes in the liver, and SUMOylated proteins involved in muscle morphogenesis pathways in the heart. Unfortunately, this type of approach used to highlight tissue-specific enriched processes is counter-intuitive and strikes as circular. Since each organ differs from the other, this list more likely reflects organ-specific proteome and not process-specific SUMOylated proteome.

For example, the brain is a very unique organ, with post-mitotic and highly specialized cells. The brain is indeed genuinely enriched in synapses as compared to all other organs of the body. Therefore synaptic proteins are (massively) enriched in brain-derived tissue, and all other organs are consequently "depleted" in synapses. It is therefore not clear what is the essence of a conclusion that SUMOylated synaptic proteins are enriched in the brain in comparison to other organs. The real question is what are the SUMOylated proteins enriched in neurons in comparison

to the neuron proteome?

Enrichment analysis of SUMO2/3 pathways should be done by comparing the list of SUMO2/3 substrates found in a given organ to the proteome of that organ, to reflect enrichment pathways related to that specific organ, and not as compared to the presence of, or absence within the pool of SUMO2/3 targets found in the other organs. Such global organ-specific proteome analysis is available, at least, for the brain (Sharma, Schmitt et al. 2015).

Further supporting the need for a re-analysis of the enrichment of SUMO2/3 targets in each organ, later in the manuscript the authors describe solely 21% of SUMO2/3 conjugation rate in the brain, with 78% of mature SUMO2/3 being unconjugated. Out of the 21% SUMO2/3 conjugates, more than half were to SUMO enzymes. In figure 4B, around 90% of the brain SUMOylated proteins are nuclear proteins and indeed, looking at Table S5 and S6, nuclear proteins are apparently the most abundant SUMO2/3 targets in the brain. These observations do not correlate with the claim that synaptic proteins are enriched SUMO2/3 targets in the brain, as apparent in Figure 6 and Figure 3C. The authors find by themselves these findings striking and stay precautious about possible false-discovery of SUMOylated synaptic proteins. Still, such a strong argument could lead to an over-interpretation of the role of SUMOylation at synapses. As such, Figures 4B, 3C and the organ-specific function labeling in Figure 6 should either be removed and the argument of organ specificity dropped, or Figure 4B, 3C and 6 should be up-dated according to the novel comparison method.

Minor Points

- The scrolling between the 13 different Excel tables would have been easier if they were named according to their numbering in the manuscript.
- Not many studies previously described proteomics analysis from mice tissues. Along with the study from Becker et al. 2013 (NSMB), the authors should also mention previous work done in mice (Tirard, Hsiao et al. 2012, Yang, Sheng et al. 2014). The later two were the first in vivo SUMO proteomics and should be cited despite the absence of SUMO sites in their analysis and the use of genetically engineered animals.
- The evidence linking SUMOylation and neurological disorders such as PD and AD are still weak. As such, ischemia is the neurological disorder with the strongest link to altered SUMOylation. The authors should rather cite (Krumova and Weishaupt 2012, Yang and Paschen 2015).
- The authors mention in their text that they used the antibody 8A2 from the hybridoma bank, but it says abcam in the materials and methods, from which company was the antibody bought?
- Over the entire manuscript, the authors compare humans versus mouse SUMO2/3-ome. However, HEK293 are perhaps human cells, but they are cancer cells with no evolutionary constrain and the distinction between human cancer cell line and human kind should be done more carefully in the manuscript. The differences found between the human and mouse SUMO2/3-ome might rather reflect differences between in vitro grown cancer cell lines and mouse organs in vivo. Similarly, the title and some part of data presentation are again over-selling as it is a SUMO2/3 resource, and no information about SUMO1 is included. As such, the study does not enclose all SUMO paralogues and it should be referred specifically to SUMO2/3. So as for the species comparison, it would be better to tone down some part of the text and remain more factual, the density of the data is fantastic enough and do not require further embellishment.
- Figure 3b is cited instead of figure 4b in the subcellular localization part.
- Reproducibility of the IP protocol from mouse tissues:
Previous protocols for the enrichment and identification of SUMO substrates from mouse organs

were described and are extremely useful resources (Barysch, Dittner et al. 2014, Tirard and Brose 2016). The authors should provide some more details about their methods concerning the preparation of mouse organs lysates to facilitate reproducibly by other labs:

(i) Regarding the homogenisation step of the various organs, can the authors explain whether de-SUMOylation happens during this step, the reasoning for the absence of protease inhibitors during homogenization, and the precise composition of the lysis buffer? Was there a visible pellet after centrifugation and were there any SUMO2/3 conjugates in this pellet fraction? If so, a Western blot analysis of the pellet would be beneficial.

(ii) Can it be that the LysC digest from complex tissues does not release all proteins and expose all SUMO2/3 targets? A 1D gel followed by anti-SUMO2/3 Western blot of total tissues lysates, LysC digested before/after IP, as provided for HEK cells would be beneficial (Figure S2, A and B) and would nicely illustrate the depletion of SUMO2/3 peptides during the IP.

References:

- Barysch, S. V., C. Dittner, A. Flotho, J. Becker and F. Melchior (2014). "Identification and analysis of endogenous SUMO1 and SUMO2/3 targets in mammalian cells and tissues using monoclonal antibodies." *Nature protocols* 9(4): 896-909.
- Becker, J., S. V. Barysch, S. Karaca, C. Dittner, H. H. Hsiao, M. Berriel Diaz, S. Herzig, H. Urlaub and F. Melchior (2013). "Detecting endogenous SUMO targets in mammalian cells and tissues." *Nature structural & molecular biology* 20(4): 525-531.
- Krumova, P. and J. H. Weishaupt (2012). "Sumoylation in neurodegenerative diseases." *Cellular and molecular life sciences* : CMLS DOI: 10.1007/s00018-012-1158-3
- Sharma, K., S. Schmitt, C. G. Bergner, S. Tyanova, N. Kannaiyan, N. Manrique-Hoyos, K. Kongi, L. Cantuti, U. K. Hanisch, M. A. Phillips, M. J. Rossner, M. Mann and M. Simons (2015). "Cell type- and brain region-resolved mouse brain proteome." *Nat Neurosci* 18(12): 1819-1831.
- Tirard, M. and N. Brose (2016). "Systematic Localization and Identification of SUMOylation Substrates in Knock-In Mice Expressing Affinity-Tagged SUMO1." *Methods Mol Biol* 1475: 291-301.
- Tirard, M., H. H. Hsiao, M. Nikolov, H. Urlaub, F. Melchior and N. Brose (2012). "In vivo localization and identification of SUMOylated proteins in the brain of His6-HA-SUMO1 knock-in mice." *Proceedings of the National Academy of Sciences of the United States of America* 109(51): 21122-21127.
- Yang, W. and W. Paschen (2015). "SUMO proteomics to decipher the SUMO-modified proteome regulated by various diseases." *Proteomics* 15(5-6): 1181-1191.
- Yang, W., H. Sheng, J. W. Thompson, S. Zhao, L. Wang, P. Miao, X. Liu, M. A. Moseley and W. Paschen (2014). "Small Ubiquitin-Like Modifier 3-Modified Proteome Regulated by Brain Ischemia in Novel Small Ubiquitin-Like Modifier Transgenic Mice: Putative Protective Proteins/Pathways." *Stroke; a journal of cerebral circulation*. DOI: 10.1161/STROKEAHA.113.004315

Reviewer #2 (Remarks to the Author):

This study presents a new method to quantify SUMOylation in complex samples and carries out a large-scale comparison of SUMOylation patterns between human cell lines and mouse tissues with biological observations that are interesting to the community.

While the work appears to provide sequencing and quantification depth not achieved so far, the presented numbers are based on sometimes weak ground. The result section of the manuscript is not well written, making it very tricky to understand the authors' reasoning. By presenting too many often irrelevant numbers, the reader gets confused and gets to understand the overall

meaning of the results only after exhaustive evaluation of figures and methods. Manuscripts in a high-impact journal like Nature Communications have a higher level of writing than this manuscript.

Instead of throwing with huge amounts of numbers and interpretations based on often weak results, I strongly suggest that the authors shorten the manuscript and focus on their most relevant results.

- Abstract: The authors state "striking differences in SUMO metabolism" between cell culture, normal tissues and also between organ types. Which are the most interesting differences? The investigated cell types are very different and one therefore already expects them to be different in many ways, also including SUMOylation.

- The study solves common problems in SUMOylation MS by using serial digestion. As this method is not too commonly practised, I suggest to present the pros and cons of using serial digestions in a few sentences.

- The paragraph comparing number of identifications and total spectral intensity between endogenous and exogenous analyses are very difficult to read. The authors extensively state differences between pairs of conditions, and then summarize the paragraph by stating a "striking difference between endogenous and exogenous SUMO dynamics". The paragraph would read much better if the authors presented the differences in a summarized fashion. If I understand the numbers correctly, increase of exogenous SUMOylation to heat shock and MG132 treatment is stronger than for endogenous SUMOylation. Most of the results of the manuscript are presented in the same confusing way which makes it very difficult for the reader to follow.

- Median copy numbers: The comparison of protein-per-cell copy numbers is dubious at least. How did the authors estimate the number of cells? It is very difficult to identify the different studies in Fig. 1E as one needs to search for the corresponding literature references. Where does the "20-fold greater depth" come from? Why is a copy number of 100,000 versus 60,000 still strikingly different when taking into account the large variation in Fig. 1E?

- Motifs of SUMO sites: Table S1 contains a lot of information. Where do I find a table with the distribution of the different SUMOylation motifs? Figure 1F gives a total view of enriched residues. I cannot see the inverted [ED]xK motif there. It could easily be any other combination of the multitude of residues shown in the figure. I strongly suggest to carry out a motif enrichment analysis, such as by using motif-x. It is moreover shown that KxE-type sites are more abundantly modified with higher confidence identifications. Why should a modification be preferably decorating highly abundant proteins? I interpret not finding the KxE motif in low abundant peptides to be a result from mostly wrong identifications, thus showing that the sequencing depth is not as high as mentioned in the manuscript.

- Number of SUMOylation sites per protein: The paragraph in the Result section states that there are more SUMO sites after treatment. This can be observed from Fig. 1H as well. However, proteins found in all treatments together have only an average of 2 sites. The authors need to discuss this observation. Or does it come from falsely identified SUMO sites?

- catalogue of in vivo SUMO sites: "we could confirm the existence of most known SUMOylation sequence motifs within the mouse data (Fig. 2E)." The figures looks similar to 1F by showing similar distributions of residues but no motifs. How is the distribution not just the result of using the same antibody for the enrichment?

- "preferences of the SUMO enzymatic machinery varies across tissues". It is not shown that the observed differences do not just come of different protein abundances in the different tissues.

- Enrichment analysis of SUMO versus tissues: In order to show that observed enriched terms are

not due to different protein expression, Fig. 3D shows general differences in expression levels. This analysis is very shallow. The enrichment should be carried out on the proteins that were different between organs AND were corrected by protein expression changes. From what I understand, the authors did not carry out such a correction. This counts for all presented enrichment analyses. As the cell types are hugely different, the authors should additionally carefully choose the backgrounds.

- Comparison of human and mouse proteins: "We compared identified protein-coding genes". Isn't it "We compared identified proteins from ortholog genes"?
- SUMOylation in human and mouse: "Interestingly, some proteins only SUMOylated in response to stress in cell culture ...". Why is this interesting? How many are some proteins? Is it a significant proportion (e.g. compared to randomization)?
- Subcellular localization: Fig. 3B -> Fig. 4B
- Crosstalk: How much proline-directed phosphorylation is usually observed in the cell types? Show that the combination of SUMO and proline-directed phosphorylation is significant on basis of statistical tests. The same counts for the other observations of enriched phosphorylation sites. The paragraph contains too many numbers which do not provide important insights. The observation that Mdc1 was having most of the phosphorylation-proximal SUMOylation in testis could be from very high expression of this protein. This has not been investigated.
- Defining SUMO equilibrium: Which statistical tests were used to show that chain formation was "significantly" different?
- Discussion: The authors state that the observed fast evolution of SUMOylation targets may lead to higher (dys)regulation of SUMOylation in disease. How are regulation by disease and evolution connected here?
- Database search: By having multiple enzymes allowing many miscleavages and different PTMs, I assume that the search space is immense and most likely will have impact on the number of results obtained with an FDR of 1%. I suspect that this will lead to an unusual high number of false positives. I strongly suggest applying a lower FDR threshold and investigate how much the reproducibility between replicates improves, especially for the mouse tissues for which reproducibility was not shown.
- The method section states scores for localization and enrichment without providing their equation. The exact formulae need to be provided.
- Gene Set Enrichment Analysis is not an annotation category
- a p-value threshold of 2% in the enrichment analysis is unusually high and should therefore also written in the Result section.

Reviewer #3 (Remarks to the Author):

In the manuscript by Hendriks I et al. the authors describe an advanced proteomics method for identifying sumoylation sites from native endogenously expressed SUMO2/3. Using the SUMO2 8A2 monoclonal antibody initially generated by the Matunis lab (Zhang XD et al. 2008 Mol Cell) and characterized by Becker et al. 2013 NSMB, the authors refined a method involving peptide immunoaffinity enrichment and dual protease digestion to yield SUMO2/3 remnants with a 1 kDa

remnant (DVFQQQQTGG) that could be identified by mass spectrometry. One interesting aspect of this method is that it permits the quantification of free and conjugated SUMO2/3 in the context of each enriched sample, and it would have been nice to see carefully controlled experiments demonstrating how well this works quantitatively. Using their new protocol, the paper profiles the SUMO proteome of 8 different mouse organs in the basal state and of the HEK human cell line under a range stress conditions. The resulting dataset of endogenous Sumoylation sites reconfirm the preference of SUMO to modify the K-x-E consensus, lysines in disordered regions, and nuclear proteins. As the authors note, each of these findings is established in the SUMO proteomics field. The dataset itself is meaningful, and affords the authors an opportunity to describe what was seen and speculate about what it might mean in the context of cellular signaling. That said, the text is unnecessarily lengthy and could easily be trimmed by half without diminishing its value to the reader. An overarching concern stems from the extent to which this Sumoylation catalog was interpreted to make comparisons and draw conclusions about the function of the SUMO2/3 signaling (i.e. Figure 6). These concerns are further addressed in more detail below.

1. In Figure 1 the data compares this study to the recent exogenous K0-SUMO2 study published by the same group (K0-SUMO2 = His-SUMO2-K0-GG Hendricks I et al. 2017). The text makes several comparisons between endogenous and exogenous (overexpressed) SUMO. However, this is not an apples to apples comparison given that this study profiles endogenous SUMO2/3 sites in HEK cells while the exogenous K0-SUMO2 uses U2OS and HeLa cells. Even with similar treatments of heat stress or MG132, the differences in the proteome across these three cell lines cloud any interpretation or comparison between exogenous vs endogenous Sumoylation sites (Fig.1D). If the aim is to highlight discrepancies between endogenous and exogenous Sumoylation, wouldn't it have been better to profile endogenous SUMO2/3 sites of either U2OS or HeLa with heat and MG132 treatments and compare the data sets directly

2. The identification of sumoylation sites from peptide enrichment of endogenous SUMO requires a large ~1kDa remnant, which as a branched peptide could be subject to its own fragmentation to produce remnants. Can the authors show a couple 'representative' MS2 spectra showing assignment of ions emanating from the SUMO remnant? Are there characteristic diagnostic remnants that can serve as a reporter? Do the authors see any evidence suggesting that remnant fragmentation makes it difficult for search engines to properly assign fragment ions on the substrate sequence and assign of SUMO sites.

3. Have the authors performed experiments to show how efficiently AspN digests the extended SUMO2/3 remnants?

4. A major tenet of this paper is the importance of identifying endogenous SUMO target proteins and their sites from mouse tissues. Extending from this proteome to highlight possible functions of SUMO across the organs seems like quite a stretch, and one where the most meaningful discriminant is underlying baseline proteome of each tissue. Moreover, many SUMO sites could not be easily sequenced across organs leading to missing values between organs like the brain and heart. The paper attempts to overcome this issue by attempting MS1 precursor matching. Additional explanation is required for how this was done, and more importantly, how the resulting 'match between runs' hits were verified. It seems plausible that a sizeable fraction of those identified in this manner are false positives and it is not possible to assess how prevalent this is. At first principles, this approach is better suited for technical or biological replicates from a single biological system than for comparing different organs and tissues that have distinct proteomes.

5. If the authors removed the MS1 match between runs hits for the organs can and do they draw the same conclusions? (I.e. K-x-E consensus, etc).

6. A conclusion of the paper is that higher density of conjugated SUMO is present in a cell line than in mouse organs. This data stems from comparing the overall sample purity for sumoylated proteins after the SUMO peptide pull-down with the amount of background signal and show more

free SUMO2/3 in organs by almost twice as much as HEK cells. Can the authors rule out the possibility that this is due to experimental discrepancies. For example, if whole cell extract was immunoblotted with the anti-SUMO2/3 8A2 clone is more free SUMO observed in mouse organs than in HEK cells? Is this a uniform observation across all organs examined? Is it possible that the fraction of conjugated versus free SUMO observed is adversely affected by differences in lysate preparation protocols required for tissues versus cells or the intrinsic levels of desumoylating activity? It seems that there are a number of variables not accounted for here, so some confirmatory data would be reassuring.

7. The use of GO terms to classify the Sumoylated proteins (Fig.2G, 3C) is something often done in proteomics papers, but is arguably of little or no value to biologists working in this field. For example, Figure 3C references how SUMO sites in the brain “we observed enrichment for synapse, myelin sheath, neuron...”. Without some data elucidating how the SUMO proteome controls functions in specific tissues like the brain, the discussions surrounding this data do not meaningfully extend the field's understanding of SUMO function beyond the value of the data provided by the dataset itself.

8. The PCA analysis in Figure 3B shows reproducibility of the experiment, but otherwise doesn't add much. It seems expected that data sets cluster per organ based on differences in the baseline proteome.

9. Page 11 second paragraph, the figure is referenced should be 4B, not 3B, in the section “Subcellular localization”

10. Figure 4F and 4G are quite similar... is it necessary to convey the same point?

11. Figure legend in 3A could be improved by adding a legend for the heatmap.

12. Have the authors attempted to treat anti-8A2 enriched peptides with a SUMO isopeptidase like SENP2 to cleave the 1kDa remnant? Is it possible for the mass spec sequence the unmodified peptide that corresponds to the originally identified SUMO site? If so, this would validate the SUMO sites and potentially reveal additional peptides that were otherwise refractory to MS-identification with the remnant attached.

13. The sentence that reads “Reassuringly, this median copy number is on par with the second-largest exogenous SUMO study(ref28), and just short in depth of the largest exogenous SUMO study(ref2) which achieved a median copy number of 44,000 (Fig. 1E).” feels like an gratuitous self-reference and an unnecessary jab at another of the labs working in this field.

Point-by-Point response to reviewers

General remarks:

We would like to thank the reviewers for their critical reading of the manuscript, and we truly appreciate and acknowledge that a substantial amount of time has been spent going over our manuscript. We found the reviewer comments very helpful and constructive, and in the revised manuscript, we have addressed all comments raised by the reviewers as outlined in detail below. As a result, we are confident that the revised version of our manuscript has been greatly improved and is now overall stronger.

Reviewer #1 (Remarks to the Author):

The current study by Hendriks and colleagues present a novel proteomic resource of SUMO2/3 conjugated proteins found in a human-derived cell line and mouse tissues. The study describes the combination of an anti-SUMO2/3 immunoprecipitation with a mass spectrometric approach that enables enrichment and identification of endogenous SUMO2/3 targeted lysines in proteins from a human cell line and various mouse tissues. While such large proteomics resources were already available for multiple cell lines, here the authors went further and established a novel method that allows for the identification of SUMO2/3-targeted lysines in various mouse tissues, aiming to deepen our current understanding of the roles of SUMO2/3 conjugation *in vivo*.

Reply: We are happy that the reviewer recognizes the novelty of our method, and that the method was uniquely applied *in vivo* in order to gain insight into the function of SUMO2/3 in organs.

The amount of data generated in the Hendriks et al. study is enormous, and the impact for the field will be, without doubt, major. After careful characterization of their protocol using HEK cells, the authors applied their method to enrich and identify SUMO2/3 conjugates from 8 different mouse tissues and reached a resolution never achieved so far. They not only mapped endogenous SUMO2/3 acceptor sites in various mouse tissues, but they also made use of sophisticated bioinformatical tools to address major questions in the SUMO field.

They deciphered species and organ-specific SUMO2/3 proteomes, the sub-cellular preferences of SUMO2/3 conjugates, addressed cross-talk with phosphorylation, proteins structural preferences of the SUMO2/3-targeted sites, and evolutionary conservation among the SUMO2/3 proteins substrates. Interestingly, their approach also reveals the dynamics of endogenous SUMOylation by quantifying the equilibrium of free versus bound SUMO2/3, and by providing a description of the topology of SUMO2/3 chains in vivo and in vitro. In sum, this paper de-coded the SUMO2/3-ome in mice. As such, I highly recommend the study by Hendriks et al. to be published in Nature Communications. The present study is convincing and will become influential in the SUMO field and in other field of science.

Reply: We are grateful for the very kind and encouraging words from the reviewer, and the appreciation of the large amount of time and effort we invested towards generating an in-depth SUMO study and a proteomics resource of this magnitude.

The authors now face the challenges of making such an immense repertoire of information easily accessible to scientists of other disciplines. Further, the authors should highlight the usefulness of their method to other labs. In this context, one main point, and a few minor issues should be address by the authors.

Reply: Indeed, we realized during drafting of our manuscript that the amount of information was rather large, and often very technical, and we tried to streamline our manuscript to be accessible to a broader audience. We are grateful for the reviewer's suggestions on how to further improve our manuscript.

Major Point

The authors established the catalogue of SUMO2/3 substrates in 8 different mice organs and attempt to describe an organ-specific function of SUMO2/3 targets. To do so, they compare the SUMO2/3 targets found in the organ in questions to the targets found (and not found) in other organs. In the figure legend from Figure 3C, the authors say that "SUMO target proteins were considered enriched in an organ if significantly different from other organs". Consequently, in the brain, SUMOylated synaptic proteins are enriched as compare to other organs, and so are SUMOylated enzymes involved in metabolic processes in the liver, and SUMOylated proteins involved in muscle morphogenesis pathways in the heart. Unfortunately, this type of approach used to highlight tissue-specific enriched processes is counter-intuitive and strikes as circular. Since each organ differs

from the other, this list more likely reflects organ-specific proteome and not process-specific SUMOylated proteome.

For example, the brain is a very unique organ, with post-mitotic and highly specialized cells. The brain is indeed genuinely enriched in synapses as compared to all other organs of the body. Therefore synaptic proteins are (massively) enriched in brain-derived tissue, and all other organs are consequently “depleted” in synapses. It is therefore not clear what is the essence of a conclusion that SUMOylated synaptic proteins are enriched in the brain in comparison to other organs. The real question is what are the SUMOylated proteins enriched in neurons in comparison to the neuron proteome?

Enrichment analysis of SUMO2/3 pathways should be done by comparing the list of SUMO2/3 substrates found in a given organ to the proteome of that organ, to reflect enrichment pathways related to that specific organ, and not as compared to the presence of, or absence within the pool of SUMO2/3 targets found in the other organs. Such global organ-specific proteome analysis is available, at least, for the brain (Sharma, Schmitt et al. 2015).

Reply: The reviewer raises a fair point. We made a comparison of SUMO-enriched functions between different organs, and indeed in many cases this would highlight organ-specific functions because we specifically tested for organ-enriched proteins, using only SUMOylated proteins we identified as a background. After careful re-evaluation of our data processing, we observed that some self-reinforcement of GO terms may have occurred owing to organ-specific expression patterns. As the reviewer suggested, we have now re-performed the same analysis, but this time using relevant tissue background proteomes for enrichment analysis derived from the TISSUES database (Santos et al., 2015). Moreover, for this revised analysis we only considered proteins for each organ type harboring SUMO sites directly identified via MS/MS, hereby completely excluding MS1-level matching between runs (also in response to comments raised by Reviewer #3).

As expected, this re-analysis leads to many of the canonical SUMO functions enriched across all eight organs, as already entailed in our overall enrichment analysis. Thus, following the outcome of the re-analysis we redesigned figure 3C to only list terms uniquely enriched in the specific organs (i.e. not significantly enriched in any of the other organs), which in all cases included terms also significantly enriched over their respective background proteomes. For two organs, brain and kidney, this meant fewer organ-unique enriched terms, so instead we listed some globally SUMO-enriched terms (but not unique to these organs).

We additionally removed the combined lung and spleen comparison to the other six organs which was previously part of Table S8 and only mentioned in the discussion, as this approach was no longer compatible with the new improved analysis.

Further supporting the need for a re-analysis of the enrichment of SUMO2/3 targets in each organ, later in the manuscript the authors describe solely 21% of SUMO2/3 conjugation rate in the brain, with 78% of mature SUMO2/3 being unconjugated. Out of the 21% SUMO2/3 conjugates, more than half were to SUMO enzymes. In figure 4B, around 90% of the brain SUMOylated proteins are nuclear proteins and indeed, looking at Table S5 and S6, nuclear proteins are apparently the most abundant SUMO2/3 targets in the brain. These observations do not correlate with the claim that synaptic proteins are enriched SUMO2/3 targets in the brain, as apparent in Figure 6 and Figure 3C. The authors find by themselves these findings striking and stay precautionous about possible false-discovery of SUMOylated synaptic proteins. Still, such a strong argument could lead to an over-interpretation of the role of SUMOylation at synapses. As such, Figures 4B, 3C and the organ-specific function labeling in Figure 6 should either be removed and the argument of organ specificity dropped, or Figure 4B, 3C and 6 should be up-dated according to the novel comparison method.

Reply: As described above, we have revisited the term enrichment analysis for all organs, using relevant background proteomes to minimize self-reinforcement. For all organs, this means that SUMO is indeed a nuclear-enriched process, although the enrichment for brain is not as strong as it is for other organs. Whereas most other organs are actually depleted in cytoplasmic-annotated proteins, brain is not.

Moreover, after revisiting our term enrichment analysis, as outlined above, the synapse function was no longer significantly enriched for SUMOylated proteins we detected in brain, owing to testing it against the TISSUES brain proteome. Thus, the reviewer was absolutely correct in his or her comment, and we have updated figures 3C and 6 accordingly.

We would like to note that brain was an outlier from our previous analysis, as indeed brain features the highest percentage of unconjugated SUMO, and the highest relative fraction of SUMOylation occurring on the SUMO enzymes. Owing to this fact, the normal brain tissue we analyzed now demonstrates almost no uniquely SUMOylated pathways compared to all other organs, which is consistent with our other observations, and nonetheless remarkable. All other organs retained most of their organ-specific enriched SUMOylated terms, but are now both significantly enriched over their respective background proteomes, and are unique to that

specific organ, which according to our reasoning does make them worth mentioning – as technical validation and for biological insight.

Minor Points

- The scrolling between the 13 different Excel tables would have been easier if they were named according to their numbering in the manuscript.

Reply: We apologize for any erroneous numbering of the supplemental tables; we have now double checked the numbering of all supplemental items, and provided a descriptive list of all supplemental items. Moreover, the number of tables has dropped to 12, because the organ-specific GO analysis is now exclusively based on direct MS/MS evidence at the sites level.

- Not many studies previously described proteomics analysis from mice tissues. Along with the study from Becker et al. 2013 (NSMB), the authors should also mention previous work done in mice (Tirard, Hsiao et al. 2012, Yang, Sheng et al. 2014). The later two were the first in vivo SUMO proteomics and should be cited despite the absence of SUMO sites in their analysis and the use of genetically engineered animals.

Reply: These important studies have now been included in the introduction, and referenced accordingly.

- The evidence linking SUMOylation and neurological disorders such as PD and AD are still weak. As such, ischemia is the neurological disorder with the strongest link to altered SUMOylation. The authors should rather cite (Krumova and Weishaupt 2012, Yang and Paschen 2015).

Reply: We appreciate the reviewer's expertise on SUMOylation in the context of neurological diseases, and we have updated the introduction and references according to the reviewer's suggestion.

- The authors mention in their text that they used the antibody 8A2 from the hybridoma bank, but it says abcam in the materials and methods, from which company was the antibody bought?

Reply: The antibody used in this study is available from the hybridoma bank, and as such can be produced in-house by labs, which would be financially more attractive. Furthermore, the procedure for generating the antibody in-house has been described in detail (Barysch et al., 2014). However, because in-house production of antibodies can be time-consuming, we acquired the antibody from Abcam as a time-saving measure. We updated the main text and the methods section to clarify the origin of the used antibody, and we have now added a reference to Barysch et al.

- Over the entire manuscript, the authors compare humans versus mouse SUMO2/3-ome. However, HEK293 are perhaps human cells, but they are cancer cells with no evolutionary constrain and the distinction between human cancer cell line and human kind should be done more carefully in the manuscript. The differences found between the human and mouse SUMO2/3-ome might rather reflect differences between in vitro grown cancer cell lines and mouse organs in vivo.

Reply: The difference in SUMO architecture between *in vitro* grown cancer cells, and *in vivo* mouse organs, was exactly one of the major points we were trying to make. The comparisons made between these vastly different model systems mainly serve to outline some of the very large differences observed, and to highlight that the community should be aware that SUMOylation in cancer cell lines (a very common model system), may not necessarily reflect how SUMOylation would behave in normal tissues.

Similarly, the title and some part of data presentation are again over-selling, as it is a SUMO2/3 resource, and no information about SUMO1 is included. As such, the study does not enclose all SUMO paralogues and it should be referred specifically to SUMO2/3. So as for the species comparison, it would be better to tone down some part of the text and remain more factual, the density of the data is fantastic enough and do not require further embellishment.

Reply: Indeed, the method described in our manuscript cannot be directly used to enrich and study SUMO1. To emphasize this, we have included specific mentioning of SUMO2/3 in the abstract, while throughout the manuscript we have replaced numerous instances of “SUMO” with “SUMO2/3”. Hopefully, it is now much clearer to the readers that our method is entirely focused on the study of SUMO2/3.

- Figure 3b is cited instead of figure 4b in the subcellular localization part.

Reply: We have corrected the erroneous reference.

- Reproducibility of the IP protocol from mouse tissues:

Previous protocols for the enrichment and identification of SUMO substrates from mouse organs were described and are extremely useful resources (Barysch, Dittner et al. 2014, Tirard and Brose 2016). The authors should provide some more details about their methods concerning the preparation of mouse organs lysates to facilitate reproducibly by other labs:

(i) Regarding the homogenisation step of the various organs, can the authors explain whether de-SUMOylation happens during this step, the reasoning for the absence of protease inhibitors during homogenization, and the precise composition of the lysis buffer? Was there a visible pellet after centrifugation and were there any SUMO2/3 conjugates in this pellet fraction? If so, a Western blot analysis of the pellet would be beneficial.

Reply: We fully agree with the reviewer; one of our goals is to present the community with a ready-to-use method that can be reproduced with ease, and thus applied in any lab with access to contemporary MS-based proteomics.

To answer the reviewer question regarding sample preparation and homogenization: Using the same guanidine-based buffer we use for lysis of cell pellets, we performed bead-grinding of the deep-frozen organs directly in 6 M guanidine, buffered at pH 8.5 with 50 mM, in the absence of broad-range protease inhibitors. The reasons for this are primarily that (i) 6 M guanidine is highly chaotropic and sufficient to fully homogenize all cells and tissues, while simultaneously denaturing proteins and thus preventing SUMO protease activity (ii) SUMO proteases (and cysteine-proteases in general) are not inhibited at all by standard broad-range protease inhibitors (iii) furthering the notion that guanidine is capable of inhibiting SUMO proteases, guanidine is commonly used as an efficient agent to efficiently denature and thus inhibit RNAses (Chomczynski and Sacchi, 1987). We did add fresh chloroacetamide to the buffer just prior to homogenizing the organs, which is a cysteine-reactive chemical that inhibits ubiquitin- and SUMO-proteases. We did not add this for the lysis of cultured cells, as 6 M guanidine is harsh enough and no SUMO protease activity remains (as is apparent from the near-100% conjugation rate we observe in cell lines) – it was added to the organ homogenization as an extra safety measure. Note that we avoided the use of both iodoacetamide and NEM because whereas they are potently reactive towards cysteines, they can also induce MS-related artifacts.

The larger organs were reduced to smaller pieces by percussion while chilled at -80°C . After bead-grinding the organs, which generally completely homogenized the tissues within 10 seconds, and subsequent sonication and incubation with TCEP and additional fresh chloroacetamide, there were no discernable pellets visible after centrifugation. As organs comprise a complex mixture of proteins, lipids, etc., the main clarification of the homogenized lysate was achieved by filtering the lysate through 0.45 μm filters, as described in the methods section. We have clarified these points in more detail in the methods section of our manuscript.

(ii) Can it be that the LysC digest from complex tissues does not release all proteins and expose all SUMO2/3 targets? A 1D gel followed by anti-SUMO2/3 Western blot of total tissues lysates, LysC digested before/after IP, as provided for HEK cells would be beneficial (Figure S2, A and B) and would nicely illustrate the depletion of SUMO2/3 peptides during the IP.

Reply: There is a theoretical possibility that a Lys-C digest of tissue lysates may be less effective than a similar digestion of a cell lysate, but we have not observed this in practice. Especially considering that during sample preparation, we perform two rounds of Lys-C digest with considerable amounts of the enzyme to achieve proper digestion. Still, to be sure, and fully address the reviewers concern, we have extracted the missed peptide cleavages of the Lys-C enzyme from the proteomics data, in both cell lines and mouse tissues. From this we find that Lys-C performed very well in both cell line and tissue lysates. On average, we observed a Lys-C missed cleavage count of 1.042 across 258,081 SUMO site MS/MS spectra in human cells, versus an average missed cleavage count of 1.030 across 26,956 SUMO site MS/MS spectra from mouse tissues. Please note that the SUMO-modified lysine residue is by itself considered a missed cleavage by the data processing software (MaxQuant), so a proper missed cleavage count per peptide by Lys-C would correspond to 0.042 and 0.030 in cell lines and tissues, respectively. This demonstrates that Lys-C digestion is slightly more efficient in the tissue samples.

Nonetheless, as the reviewer suggested, we performed immunoblot analysis on representative organs pre- and post-IP, to demonstrate that SUMO-2/3 peptides are indeed also efficiently depleted from mouse organs (Figure S6). The immunoblot analysis also further demonstrates efficient Lys-C digestion of the mouse organs, with no large protein fragments visible on Ponceau-S, and the banding pattern on the 8A2 immunoblot closely matching between HEK cells and mouse organ. Finally, the immunoblot analysis supports the high density of SUMO2/3 in HEK cells compared to mouse organs, as observed from MS analysis.

References:

- Barysch, S. V., C. Dittner, A. Flotho, J. Becker and F. Melchior (2014). "Identification and analysis of endogenous SUMO1 and SUMO2/3 targets in mammalian cells and tissues using monoclonal antibodies." *Nature protocols* 9(4): 896-909.
- Becker, J., S. V. Barysch, S. Karaca, C. Dittner, H. H. Hsiao, M. Berriel Diaz, S. Herzig, H. Urlaub and F. Melchior (2013). "Detecting endogenous SUMO targets in mammalian cells and tissues." *Nature structural & molecular biology* 20(4): 525-531.
- Krumova, P. and J. H. Weishaupt (2012). "Sumoylation in neurodegenerative diseases." *Cellular and molecular life sciences* : CMLS DOI: 10.1007/s00018-012-1158-3
- Sharma, K., S. Schmitt, C. G. Bergner, S. Tyanova, N. Kannaiyan, N. Manrique-Hoyos, K. Kongi, L. Cantuti, U. K. Hanisch, M. A. Philips, M. J. Rossner, M. Mann and M. Simons (2015). "Cell type- and brain region-resolved mouse brain proteome." *Nat Neurosci* 18(12): 1819-1831.
- Tirard, M. and N. Brose (2016). "Systematic Localization and Identification of SUMOylation Substrates in Knock-In Mice Expressing Affinity-Tagged SUMO1." *Methods Mol Biol* 1475: 291-301.
- Tirard, M., H. H. Hsiao, M. Nikolov, H. Urlaub, F. Melchior and N. Brose (2012). "In vivo localization and identification of SUMOylated proteins in the brain of His6-HA-SUMO1 knock-in mice." *Proceedings of the National Academy of Sciences of the United States of America* 109(51): 21122-21127.
- Yang, W. and W. Paschen (2015). "SUMO proteomics to decipher the SUMO-modified proteome regulated by various diseases." *Proteomics* 15(5-6): 1181-1191.
- Yang, W., H. Sheng, J. W. Thompson, S. Zhao, L. Wang, P. Miao, X. Liu, M. A. Moseley and W. Paschen (2014). "Small Ubiquitin-Like Modifier 3-Modified Proteome Regulated by Brain Ischemia in Novel Small Ubiquitin-Like Modifier Transgenic Mice: Putative Protective Proteins/Pathways." *Stroke; a journal of cerebral circulation*. DOI: 10.1161/STROKEAHA.113.004315

Reviewer #2 (Remarks to the Author):

This study presents a new method to quantify SUMOylation in complex samples and carries out a large-scale comparison of SUMOylation patterns between human cell lines and mouse tissues with biological observations that are interesting to the community.

Reply: We are happy to read that the reviewer finds our biological observations interesting.

While the work appears to provide sequencing and quantification depth not achieved so far, the presented numbers are based on sometimes weak ground. The result section of the manuscript is not well written, making it very tricky to understand the authors' reasoning. By presenting too many often irrelevant numbers, the reader gets confused and gets to understand the overall meaning of the results only after exhaustive evaluation of figures and methods. Manuscripts in a high-impact journal like Nature Communications have a higher level of writing than this manuscript.

Reply: We are sorry to read that the reviewer is unsatisfied with our style of writing. We went through many rounds of internal manuscript revision and received input from multiple scientists with considerable experience in scientific writing. Still, the high amount of input on our manuscript, and the resulting shuffling around of the text during the final drafting, may have resulted in sentences that were either unclear or conclusions that were out of place. We also understand that the manuscript has a strong technical nature, as the development of a proteomics method to study a difficult PTM such as SUMOylation is not straight-forward. We have made further efforts to streamline the manuscript and make it more accessible to a broader audience. Still, we would like to emphasize that the main aim of the manuscript is to report the technical development of a new MS-based approach for studying endogenous SUMOylation.

Instead of throwing with huge amounts of numbers and interpretations based on often weak results, I strongly suggest that the authors shorten the manuscript and focus on their most relevant results.

- Abstract: The authors state "striking differences in SUMO metabolism" between cell culture, normal tissues and also between organ types. Which are the most interesting differences? The investigated cell types are very different and one therefore already expects them to be different in many ways, also including SUMOylation.

Reply: Our intent here is to highlight the considerable difference between cultured cell lines in general, and the organs we investigated, and secondly between the different organ types. The most interesting differences are highlighted throughout the main display figures, and among others include the overall distribution of SUMO targets and the equilibrium between free and conjugated SUMO. We agree that it is indeed partially expected that SUMO would be regulated differently between organs, as this was our initial working hypothesis, but up until this point this would have been mere speculation from a systems-wide proteomics point of view. We provide evidence that SUMO is indeed regulated differently between cell lines and organ types.

- The study solves common problems in SUMOylation MS by using serial digestion. As this method is not too commonly practised, I suggest to present the pros and cons of using serial digestions in a few sentences.

Reply: Based on our knowledge, serial digestion is commonly applied in the general proteomics field. Lys-C digestion of samples is often followed by subsequent digestion with trypsin. In the SUMO proteomics field, many of the existing exogenous strategies also involve serial digestions, for example the K0 method, which involves Lys-C digestion, purification, and trypsin digestion (Hendriks et al., 2014). Otherwise, exogenous SUMO proteomics strategies involve multiple differential purification steps. For example, the NQTGG method features a His-tag purification, trypsin digestion, and then an NQTGG-antibody purification (Lamoliatte et al., 2014). The T90K method features a His-tag purification, Lys-C digestion, di-glycine purification, and then optionally also Glu-C digestion (Tammsalu et al., 2014). We describe our experimental rationale and setup in detail in the manuscript. Nonetheless, we have added two sentences regarding serial digestions as applied to SUMO proteomics in the Results section of the manuscript.

- The paragraph comparing number of identifications and total spectral intensity between endogenous and exogenous analyses are very difficult to read. The authors extensively state differences between pairs of conditions, and then summarize the paragraph by stating a "striking difference between endogenous and exogenous SUMO dynamics". The paragraph would read much better if the authors presented the differences in a summarized fashion. If I understand the numbers correctly, increase of exogenous SUMOylation to heat shock and MG132 treatment is stronger than for endogenous SUMOylation. Most of the results of the manuscript are presented in the same confusing way which makes it very difficult for the reader to follow.

Reply: We agree with the reviewer that this paragraph was partially confusing, and we have now revised this paragraph entirely, also according to suggestions by Reviewer #3. We believe that the paragraph is now easier to understand.

- Median copy numbers: The comparison of protein-per-cell copy numbers is dubious at least. How did the authors estimate the number of cells?

Reply: We did not perform estimation of the number of cells in this specific case – the normalized per cell IBAQ values were directly derived from a comprehensive and recent deep-proteome study (Bekker-Jensen et al., 2017), and we utilized their IBAQ values to approximate protein expression levels within cultured human cell lines. The IBAQ algorithm is one of the most efficient tools available for label-free quantification of protein across a wide range of concentrations (Krey et al., 2014).

It is very difficult to identify the different studies in Fig. 1E as one needs to search for the corresponding literature references.

Reply: We have now added the first author names and publication years into the graph for all references.

Where does the "20-fold greater depth" come from?

Reply: The 20-fold greater depth was based on the median IBAQ value of the proteins we detect, versus that of the least sensitive endogenous screen included in the comparison. We have adjusted this to 3-fold greater depth, as this is the decrease in IBAQ median we observe compared to the most sensitive currently published endogenous screen.

Why is a copy number of 100,000 versus 60,000 still strikingly different when taking into account the large variation in Fig. 1E?

Reply: The displayed values in figure 1E correspond to standard box plot visualization – including the median and the 1st and 3rd quantile IBAQ values. These values are based on the indicated number of proteins over the datasets, and the box plot primarily serves to demonstrate that the comparison is based on normally

distributed data. While the copy numbers between individual proteins can indeed vary greatly, the dataset averages are in many cases significantly different, including the 100,000 versus 60,000 for SUMOylated proteins versus ubiquitylated proteins. We have now included statistical tests for significance between SUMO targets detected in our study and all other datasets in the figure panel, to highlight both the significance and the directionality of any differences.

- Motifs of SUMO sites: Table S1 contains a lot of information. Where do I find a table with the distribution of the different SUMOylation motifs?

Reply: At the reviewer's request, we have now performed Motif-X analysis (with default 10^{-6} significance setting) to validate different SUMOylation motifs, and added the complete analysis as a tab into the SUMO sites tables (Table S1 for human motifs, Table S5 for mouse motifs). The KxE motif is the only one to have a separate column in the primary tab of table S1 – we initially chose to not make columns for other known motifs because these would further increase the load of information in table S1.

Figure 1F gives a total view of enriched residues. I cannot see the inverted [ED]xK motif there. It could easily be any other combination of the multitude of residues shown in the figure. I strongly suggest to carry out a motif enrichment analysis, such as by using motif-x.

Reply: We agree that the combined iceLogo motif contains a lot of information, and although the [ED]xK motif is visible, it is indeed not as striking as the main KxE motif. However, we believe it serves as a nice overview figure, and addition of sub-selection motifs would require a considerable amount of figure space. However, as mentioned above, we have now isolated motifs using Motif-X (with default FDR setting of 0.000001), and added the visualizations as additional supplementary figures, Figure S4 for human motifs and Figure S7 for mouse motifs. Motif-X analysis confirms the existence of the [ED]xK motifs, although they are not as abundantly enriched as any of the forward KxE-type motifs.

It is moreover shown that KxE-type sites are more abundantly modified with higher confidence identifications. Why should a modification be preferably decorating highly abundant proteins? I interpret not finding the KxE motif in low abundant peptides to be a result from mostly wrong identifications, thus showing that the sequencing depth is not as high as mentioned in the manuscript.

Reply: We apologize if we misunderstand this point, but we assume this comment is based on figure 1G, where we demonstrate that higher-scoring and more abundant SUMO sites are more commonly residing in KxE-motifs. These abundance values are not based on proteins, but on the peptides themselves. Further, the fact that KxE modifications are more abundant (and thus lead to easier-to-identify and higher scoring spectra) has been published before and is completely in line with expectations (Hendriks et al., 2014; Hendriks et al., 2017). The sequencing depth, as described in figure 1E, is based on the IBAQ values of the proteins detected, and is not related to figure 1G, nor is it related to how many sites were actually detected in these proteins or in which motif those were in.

- Number of SUMOylation sites per protein: The paragraph in the Result section states that there are more SUMO sites after treatment. This can be observed from Fig. 1H as well. However, proteins found in all treatments together have only an average of 2 sites. The authors need to discuss this observation. Or does it come from falsely identified SUMO sites?

Reply: As we described in the manuscript: “Our endogenous analysis identified ~3 SUMOylation sites per protein under standard growth conditions and in response to MG132, which increased to ~4 sites per protein in response to heat shock (Fig. 1H-I)”. Sites-per-protein conclusions should be drawn from the total number of sites and proteins across the entire condition, and not any of the exclusive overlap sections of a Venn diagram. The reason that the number of sites here is lower is because many of the sites identified are unique to the treatment, whereas the proteins targeted by SUMO tend to remain the same. This has been described before for SUMO proteomics (Hendriks et al., 2014), and we have now made this clearer in the manuscript text when describing the Venn diagrams.

Further, similar to general proteomics, when sequencing depth goes up, more peptides will be identified whereas the number of proteins will not increase as much. The reason for this is that an increase in peptide identification (i.e. sequencing depth) often results in increased sequence coverage of already identified proteins. The exact same logic applies to mapping SUMO sites – the more are identified, the less likely they will identify a new protein, meaning that while a SUMO site can be unique to a stress, the protein may not be, leading to an increased relative participation of SUMO target proteins in the exclusive central section of Venn diagrams. Beyond the fact that Venn diagrams are unsuitable for judging the quality of proteomics data, we can ensure that our data were filtered according to the highest standards in the proteomics field, and

furthermore manually filtered and validated in order to reduce false-positive identification to the absolute minimum.

- catalogue of in vivo SUMO sites: "we could confirm the existence of most known SUMOylation sequence motifs within the mouse data (Fig. 2E)." The figures looks similar to 1F by showing similar distributions of residues but no motifs. How is the distribution not just the result of using the same antibody for the enrichment?

Reply: Indeed, the overall enrichment motif for SUMOylation in mouse organs is similar to what we observed in human cell lines, which confirms one of our major points – the core targeting preferences of the SUMO enzymatic machinery are similar across all systems we tested. It does indeed make sense that our enrichment procedure similarly enriches SUMO2/3-modified peptides from any type of vertebrate sample, and this global applicability of our method is exactly what we were aiming for. However, since the antibody targets an epitope that is located far away (~4 kDa) from the potential target peptide, there should be no sequence bias towards the to-be-enriched peptides, so if there would be any biological differences, our method would highlight these.

There is a slight difference in the inverted consensus motif for mouse organs, where we did not observe the D at the -2 position to be significantly enriched. However, after performing Motif-X analysis as the reviewer suggested, we found that both DxK and ExK are enriched in mouse organs when the lysine is followed by a proline, i.e. [ED]xKP. The Motif-X analyses and motifs have been added to supplementary items (Table S1 and Figure S4 for human motifs, Table S5 and Figure S7 for mouse motifs) to more elaborately illustrate these differences, and we have commented on this in the manuscript text.

- "preferences of the SUMO enzymatic machinery varies across tissues". It is not shown that the observed differences do not just come of different protein abundances in the different tissues.

Reply: This point is similar to points raised by Reviewers #1 and #3, and we have now updated the term enrichment analysis to compensate for the relevant background proteomes of tissues, and updated text and figures accordingly. Unless otherwise indicated, the highlighted terms in Figure 3C are now both enriched compared to the relevant background proteome, and unique compared to all other organs we included in our analysis.

- Enrichment analysis of SUMO versus tissues: In order to show that observed enriched terms are not due to different protein expression, Fig. 3D shows general differences in expression levels. This analysis is very shallow. The enrichment should be carried out on the proteins that were different between organs AND were corrected by protein expression changes. From what I understand, the authors did not carry out such a correction. This counts for all presented enrichment analyses. As the cell types are hugely different, the authors should additionally carefully choose the backgrounds.

Reply: The referee has a very valid point, and as described above, we have in the revised manuscript updated our term enrichment analysis. In this new analysis, the enrichment is performed against the background proteomes as well as compared to other organs. Moreover, the revised analysis is now based on direct SUMO site MS/MS evidence only, and no longer includes MS1 matching between runs (in response to a point raised by Reviewer #3).

- Comparison of human and mouse proteins: "We compared identified protein-coding genes". Isn't it "We compared identified proteins from ortholog genes"?

Reply: We have adapted the term "protein-coding genes" as this is widely used in the mass spectrometry literature (Bekker-Jensen et al., 2017). The terminology would indeed be similar to "proteins from ortholog genes". We have altered the structure of the particular sentence in order to avoid this confusion. Regardless of nomenclature, we previously chose to use "protein-coding genes" to group SUMO target proteins in order to remove duplication of protein isoforms with the same gene from our data, as that would falsely inflate identification numbers.

- SUMOylation in human and mouse: "Interestingly, some proteins only SUMOylated in response to stress in cell culture ...". Why is this interesting? How many are some proteins? Is it a significant proportion (e.g. compared to randomization)?

Reply: We believe this is interesting because it suggests that these stress-like conditions may exist at baseline within mouse tissues. In total, 112 proteins followed this trend; however, we believe it is not necessarily the number of proteins which would be most relevant, but rather the significant differences of these proteins

between any associated functional terms. The enriched biological functions we listed are all significantly enriched compared to randomly expected, as listed in Supplementary Table S10, tab "STRESS_BOTH". We added header rows to the different tabs in this table to better explain the analyses. Moreover, we have updated the manuscript text to indicate the number of proteins and added a short sentence to describe why we believe this is interesting.

- Subcellular localization: Fig. 3B -> Fig. 4B

Reply: We have corrected the erroneous reference.

- Crosstalk: How much proline-directed phosphorylation is usually observed in the cell types? Show that the combination of SUMO and proline-directed phosphorylation is significant on basis of statistical tests. The same counts for the other observations of enriched phosphorylation sites. The paragraph contains too many numbers which do not provide important insights. The observation that Mdc1 was having most of the phosphorylation-proximal SUMOylation in testis could be from very high expression of this protein. This has not been investigated.

Reply: As the reviewer requested, we performed an in-depth analysis to assess the overall extent of proline-directed phosphorylation. For this, we used the database derived from PhosphoSitePlus. In humans, proline-directed phosphorylation occurs at a rate of 34.7%, corresponding to 12,898 SP motifs extracted from a total of 37,168 serine phosphorylation sites described in at least 3 studies (Hornbeck et al., 2012). Since we observed a 70% occurrence for proline-directed phosphorylation across 526 SUMO-phosphorylation co-modified peptides, this finding is statistically highly significant with a p-value of 2.5E-23 using Fisher Exact testing. In mouse, proline-directed phosphorylation occurs at a rate of 37.5%, with 7,174 SP motifs extracted from 19,114 serine phosphorylation sites described in at least 3 studies (Hornbeck et al., 2012). Since we observed a 73% occurrence for proline-directed phosphorylation across 49 SUMO-phosphorylation co-modified peptides in mouse tissue, this finding is also significant with a p-value of 0.0022 using Fisher Exact testing. We have added these statistical tests to the manuscript.

The enrichment for the actual phosphorylation sites relative to SUMOylation sites was statistically confirmed with tests and significance values as indicated in the figure legends (Figures 4C and 4D).

With regard to Mdc1, the reviewer is correct. Mdc1 is indeed a protein that is more highly expressed in mouse testis compared to other mouse organs (Santos et al., 2015), and this could explain why we observe a notable amount of this SUMO-phospho co-modification event in testis. However, the fact that ~90% of SUMO-proximal phosphorylation occurs on Mdc1 in testis is nonetheless intriguing, considering Mdc1 only represents a fraction of the total protein pool in mice. Furthermore, Nop58 is a protein that canonically harbors phosphorylation-dependent SUMOylation (Westman and Lamond, 2011), and is frequently found as a major target of phospho-SUMO crosstalk in SUMO proteomics screens. However, Nop58 is not targeted as much as Mdc1 in testis, whereas Nop58 was targeted considerably more in all other organs we analyzed. Additionally, Nop58 is expressed at a relatively higher level than Mdc1 is in testis (Santos et al., 2015). We have added these observations to the manuscript.

- Defining SUMO equilibrium: Which statistical tests were used to show that chain formation was "significantly" different?

Reply: As outlined in the legend of figure 5C, we used two-tailed Student's t-testing to test for significance, with ** $P < 0.001$, * $P < 0.05$.

- Discussion: The authors state that the observed fast evolution of SUMOylation targets may lead to higher (dys)regulation of SUMOylation in disease. How are regulation by disease and evolution connected here?

Reply: With SUMO preferentially targeting poorly conserved residues, and mutagenesis of proteins being a common denominator in disease (Reva et al., 2011), there would be an increased chance that SUMOylation could target residues mutated into a lysine. This, in turn, could affect various biochemical processes related to disease regulation and progression, especially since the ubiquitin and SUMO systems have previously been linked to disease progression in general and aggregate formation in neurological disease (Flotho and Melchior, 2013; Kahl et al., 2018).

- Database search: By having multiple enzymes allowing many miscleavages and different PTMs, I assume that the search space is immense and most likely will have impact on the number of results obtained with an FDR of 1%.

Reply: Indeed, the search space is considerably larger when allowing more missed cleavages and various PTMs, which in terms of MaxQuant FDR means that the filtering automatically becomes more stringent to ensure a very low false discovery (Cox et al., 2011; Cox and Mann, 2008). As a result, the increased search space actually resulted in less peptides being identified. A high missed cleavage setting is necessary because the Asp-N protease has partial activity towards glutamic acid, and often skips stretches of glutamic acids. To ensure comprehensive and unbiased identification of all MS/MS spectra, we iteratively determined the missed cleavage setting by finding a setting where the maximum number of missed cleavages encompassed less than 0.1% of all non-modified peptides. Moreover, high missed cleavage counts in combination with the MaxQuant FDR have been used before in SUMO proteomics (Tammsalu et al., 2015), and even non-specific searches have previously been used with low-specificity enzymes, which generate an even more immense search space (Lumpkin et al., 2017). Furthermore, MaxQuant applies FDR filtering at three different levels, whereas other search engines do not. For example, all peptide identifications are filtered at 1% FDR, followed by a site-specific filtering of SUMOylation sites, and in the end all proteins are filtered at similar FDR values.

When searching our data we filtered for a minimum delta score of 20, which is more stringent compared to standard SUMO proteomics analyses published in the literature where the default MaxQuant minimum delta score of 6 was used (Hendriks et al., 2014; Lamoliatte et al., 2017; Tammsalu et al., 2014). Filtering for a minimum delta score of 20 means that the score difference between the best peptide-spectrum-match (PSM) and the second-best PSM should be over 20, and considering that the scoring follows the standard formula $S = -10 \cdot \log(P)$ (Cox et al., 2011), this essentially means that at the spectral level we apply an additional 99% confidence for inclusion of the highest scoring PSM as compared to the second-highest scoring PSM. Further, in our data processing we allow several unique diagnostic ions related to the AspN-digested SUMO mass remnant in the MaxQuant searches (described in detail in the methods section). We have added several fully annotated MS/MS spectra in the revised manuscript to demonstrate this. Collectively, this significantly increases the identification confidence of our SUMO sites, and altogether our data thus only contains highly confident identifications.

I suspect that this will lead to an unusual high number of false positives. I strongly suggest applying a lower FDR threshold and investigate how much the reproducibility between replicates improves, especially for the mouse tissues for which reproducibility was not shown.

Reply: The 1% FDR setting is well-established in the proteomics field and is widely used and accepted. The MaxQuant FDR is highly efficient, and works very consistently despite alterations to search space, and tends to work better when large volumes of data are processed together (Cox et al., 2011; Cox and Mann, 2008). We agree with the reviewer that we should have included reproducibility values for the mouse experiments during our initial submission, and we have now included an overview of the Pearson correlation between the same-organ replicates (Figure S8A). In general, we observed good correlation between organ replicates, with an average Pearson correlation of 0.82 between the 5 liver replicates, compared to the average Pearson correlation of 0.86 for all cell culture experiments. Lower correlation is to be expected when doing mouse experiments because the different animals are biologically not liable to correlate nearly as well as cells grown in culture.

Out of curiosity, we did investigate the mouse experiment raw data for Pearson correlation when filtering at the overly stringent FDR of 0.1%. The Pearson correlation between same-organ replicates averaged 0.6897, and increased by only 0.0023 when applying the more stringent FDR. The differences observed between biological replicates, i.e. different animals, were considerably larger than this (up to 0.2 points) – further validating that our data is properly validated from a mass spectrometric and bioinformatics points of view.

- The method section states scores for localization and enrichment without providing their equation. The exact formulae need to be provided.

Reply: The scores for localization are automatically written by MaxQuant into the text output, and have inserted additional references to the MaxQuant paper where necessary. Overall, the score cut-off used is identical to accepting localization at a probability of >80%, which is slightly more stringent than class I high confidence phosphorylation localization, as established in the literature (Olsen et al., 2006).

The enrichment score relates to the term enrichment analysis, and is derived from output values using Fisher Exact testing with multiple-hypotheses correction, using the Perseus software. We have now provided the exact formula we used for ranking the enriched terms in our methods, although it should be noted that every single term was significant prior to this enrichment score ranking – we simply did this to make visualization of the data more intuitive to the reader.

- Gene Set Enrichment Analysis is not an annotation category

Reply: We apologize for the nomenclature used, as we “annotate” the proteins with certain terms, whether this is Gene Ontology, or Pfam, or keywords, or Gene Set Enrichment Analysis. It simply means we appended this information, as contained within the Perseus software, to the relevant proteins within one matrix so that we could perform strict FDR-controlled Fisher Exact testing in one single computational run. We have rephrased the corresponding sentence in the Methods for clarity.

- a p-value threshold of 2% in the enrichment analysis is unusually high and should therefore also written in the Result section.

Reply: The 2% p-value is post multiple-hypotheses correction, and in almost all cases the p-values are 100 times lower prior to this correction. The setting of 2% (post-FDR) is a Perseus default and commonly applied in the field. We have added this to the Result section as per the reviewer’s request.

Reviewer #3 (Remarks to the Author):

In the manuscript by Hendriks I et al. the authors describe an advanced proteomics method for identifying sumoylation sites from native endogenously expressed SUMO2/3. Using the SUMO2 8A2 monoclonal antibody initially generated by the Matunis lab (Zhang XD et al. 2008 Mol Cell) and characterized by Becker et al. 2013 NSMB, the authors refined a method involving peptide immunoaffinity enrichment and dual protease digestion to yield SUMO2/3 remnants with a 1 kDa remnant (DVFQQQTGG) that could be identified by mass spectrometry. One interesting aspect of this method is that it permits the quantification of free and conjugated SUMO2/3 in the context of each enriched sample, and it would have been nice to see carefully controlled experiments demonstrating how well this works quantitatively. Using their new protocol, the paper profiles the SUMO proteome of 8 different mouse organs in the basal state and of the HEK human cell line under a range stress conditions.

Reply: We are grateful for the reviewer's kind words, and we acknowledge that – time-permitting – we would also have liked to more closely investigate and the quantification of free and conjugated SUMO using our method. However, the main focus of our manuscript was to develop a new methodology able to identify native and endogenous SUMO2/3. The ability to quantify the equilibrium only became apparent later during the method development as we converged on the ideal set of proteolytic enzymes to use. We designed the method for mass spectrometry-based proteomics, and within this context we do believe that the quintuplicate equilibrium analysis across eight different mouse organs demonstrates a considerable quantitative reproducibility. We have added further confirmation of our observations to the revised manuscript as outlined below.

The resulting dataset of endogenous Sumoylation sites reconfirm the preference of SUMO to modify the K-x-E consensus, lysines in disordered regions, and nuclear proteins. As the authors note, each of these findings is established in the SUMO proteomics field. The dataset itself is meaningful, and affords the authors an opportunity to describe what was seen and speculate about what it might mean in the context of cellular signaling. That said, the text is unnecessarily lengthy and could easily be trimmed by half without diminishing its value to the reader. An overarching concern stems from the extent to which this Sumoylation catalog was interpreted to make comparisons and draw conclusions about the function of the SUMO2/3 signaling (i.e. Figure 6). These concerns are further addressed in more detail below.

Reply: We appreciate the reviewer's strong support of our manuscript, and that he or she finds our dataset very meaningful, and confirms that we re-confirm many system-wide SUMOylation phenomena at the endogenous and native level. Many of these observations were previously indeed only established through exogenous SUMO proteomics. We agree that our manuscript is a bit on the lengthy side, and contains a lot of data. Thus, we are grateful for the reviewer's comments and suggestions on how to improve our manuscript.

1. In Figure 1 the data compares this study to the recent exogenous K0-SUMO2 study published by the same group (K0-SUMO2 = His-SUMO2-K0-GG Hendricks I et al. 2017). The text makes several comparisons between endogenous and exogenous (overexpressed) SUMO. However, this is not an apples to apples comparison given that this study profiles endogenous SUMO2/3 sites in HEK cells while the exogenous K0-SUMO2 uses U2OS and HeLa cells. Even with similar treatments of heat stress or MG132, the differences in the proteome across these three cell lines could any interpretation or comparison between exogenous vs endogenous Sumoylation sites (Fig.1D). If the aim is to highlight discrepancies between endogenous and exogenous Sumoylation, wouldn't it have been better to profile endogenous SUMO2/3 sites of either U2OS or HeLa with heat and MG132 treatments and compare the data sets directly

Reply: Our initial rationale for quantitatively investigating HEK cells was to test out our final optimized method. Whereas we initially developed the method using HeLa and U2OS cells, we reasoned a third cell line would be a better way to test the global applicability of our strategy, before we ventured to mouse organs. Thus, it was not our initial aim to directly compare endogenous and exogenous SUMOylation, but we rather chose to do this at a global level to demonstrate that many of the identified SUMO substrates and sites, along with the core targeting preferences of the SUMO enzymatic machinery, were not notably different between endogenous and exogenous approaches.

The primary difference we did observe was an overall much lower number of sites identified in response to MG132 in HEK cells, compared to large numbers identified with the K0-SUMO method in HeLa and U2OS cells. The reviewer correctly points out that this could potentially be owing to a more global effect where HEK cells may not respond to MG132 as abruptly as HeLa and U2OS do, although we believe this is not the case. We have updated the comparison and the corresponding text, and instead shifted the MG132 comparison to the largest exogenous SUMO proteomics study where MG132 was used on HEK cells (Lamoliatte et al., 2017). Figure 1D has been updated so that the MG132 is compared between endogenous and exogenous

methods in HEK cells, and we have clearly indicated that the other comparison for control condition and heat shock is between different cell lines. Additionally, we have toned down any strong claims we previously made based on comparisons that are not using exactly the same cell line and stress type.

2. The identification of sumoylation sites from peptide enrichment of endogenous SUMO requires a large ~1kDa remnant, which as a branched peptide could be subject to its own fragmentation to produce remnants. Can the authors show a couple 'representative' MS2 spectra showing assignment of ions emanating from the SUMO remnant? Are there characteristic diagnostic remnants that can serve as a reporter?

Reply: Indeed, the reviewer is correct, and the 1 kDa Asp-N remnant is subject to a unique fragmentation pattern. We have now included several fully annotated MS/MS spectra which demonstrate the unique diagnostic ions resulting from fragmentation of the SUMO remnant, as Supplementary Data Set 1, and have commented on these diagnostic ions in the manuscript text. The 10 most-abundant diagnostic ions were included in the MaxQuant searches, and these are fully described in the methods section, so that other labs may similarly use these ions for confident identification of SUMOylation without having to perform additional technical optimization.

Do the authors see any evidence suggesting that remnant fragmentation makes it difficult for search engines to properly assign fragment ions on the substrate sequence and assign of SUMO sites.

Reply: While in general remnant fragments resulting from branched peptide fractionation can convolute HCD MS/MS spectra, the Asp-N 1 kDa remnant is small enough that modern high-resolution mass spectrometers should have no problem separating ions originating from the SUMO remnant and ions originating from the substrate peptide. The top 10 most-abundant fragments we listed in the manuscript cover the majority of the spectral peak intensity resulting from the SUMO mass remnant. In fact, this correct set of fragment ions makes it easier for search engines to identify spectra, and moreover these fragments serve as an additional layer of confirmation that the peptide was actually modified by SUMO-2/3. To achieve this, beyond the usual stringent FDR applied by MaxQuant and the demand for a very high precursor mass accuracy, we also discarded any spectra that did not contain any of the mass remnant fragment ions.

3. Have the authors performed experiments to show how efficiently AspN digests the extended SUMO2/3 remnants?

Reply: Yes, we have investigated all the different protein digest strategies in quite some detail. For example, in our initial technical tests we investigated how well the various enzymes cleaved SUMO-2/3 attached to a variety of target substrates. From these tests, we found that Asp-N is >99% efficient in releasing the 1 kDa mass remnant. This is why we chose Asp-N over Glu-C, because we found Glu-C to cleave the SUMO C-terminus into a multitude of mass remnants. All these findings are included in the manuscript and outlined in our Supplementary Text.

To provide further evidence, we investigated our HEK data and considered only free mature SUMO-2/3 C-terminal peptides. Here, we find that 99.7% of the total mass spectrometric signal intensities is derived from the fully-cleaved DVFQQQTGG peptide, while only 0.3% of intensity originating from partially-cleaved DEDTIDVFQQQTGG and DTIDVFQQQTGG. This further highlights that Asp-N is very efficient in performing the carboxyl-terminal cleavage.

Finally, while Asp-N is poor at cleaving glutamic acid residues, it is remarkably efficient at cleaving aspartic acid residues. To confirm this, we analyzed our data for the overall efficiency of Asp-N in cleaving SUMO-modified peptides. We found that across 258,081 evidences corresponding to MS/MS-identified SUMO-modified peptides, the average missed cleavage rate of Asp-N towards aspartic acid residues was only 0.137 per peptide, with ~89% of all peptides containing zero missed aspartic acid cleavages. As a reference, we typically observe a missed cleavage rate of 0.35 in trypsin digests of HeLa which we use for mass spectrometric quality control runs.

4. A major tenet of this paper is the importance of identifying endogenous SUMO target proteins and their sites from mouse tissues. Extending from this proteome to highlight possible functions of SUMO across the organs seems like quite a stretch, and one where the most meaningful discriminant is underlying baseline proteome of each tissue. Moreover, many SUMO sites could not be easily sequenced across organs leading to missing values between organs like the brain and heart. The paper attempts to overcome this issue by attempting MS1 precursor matching. Additional explanation is required for how this was done, and more importantly, how the resulting 'match between runs' hits were verified. It seems plausible that a sizeable fraction of those identified in this manner are false positives and it is not possible to assess how prevalent this is. At first principles, this

approach is better suited for technical or biological replicates from a single biological system than for comparing different organs and tissues that have distinct proteomes.

Reply: We understand the reviewer's concerns. We initially performed matching between runs in order to increase identification density between same-organ replicates, and to a degree, allow the identification of substoichiometric SUMO sites in other organs. The matching between runs procedure was done using the standard MaxQuant algorithm (Cox et al., 2014), and was only used in the primary data analysis and the assessment and visualization of replicate reproducibility.

For most secondary analyses, we already performed these analyses exclusively on MS/MS-identified sites and substrates, during our initial submission. We apologize if this was not clear in the manuscript. These analyses included the KxE consensus motif (figure 2F), SUMO substrate subcellular localization (figure 4), SUMO-phospho crosstalk (figure 4), structural predictions (figure 4), evolutionary conservation analysis (figure 4), free vs. conjugated SUMO (figure 5), SUMO chain formation (figure 5). However, matching between runs data was used for the term enrichment analysis outlined in figure 3. As also mentioned in our response to Reviewer #1 and #2, we have now fully revised the term enrichment analysis, such that it now respects the background proteomes of the organ types when determining whether a function is enriched for SUMOylation. Moreover, this analysis is now exclusively based on direct MS/MS data only. This means that we've only used matching between runs for the primary data tables and visualization, where it serves to provide a picture that is as complete as possible. We believe this application of matching remains relevant because otherwise we would accept that absence of evidence equals evidence of absence. Matching between runs also provides another layer of quality control that is important to make a fair assessment of method reproducibility. Moreover, in the instances where matching between runs might lead to a false peptide identification, such misidentification rarely occurs in more than one replicate, and since we demand presence in 4 out of 5 replicates for quantification of the mouse data these falsely identified peptides will be removed prior to data analysis. Besides, at the protein-level, the contribution of multiple sites to the same protein further diminishes any aberrant effects.

During our initial data analysis, we did evaluate the effect of matching between runs on our advanced data analyses, and we found that in general inclusion of the matching between runs data did not affect the outcomes by more than 10%, and in no cases altered the conclusions or the significance of the data.

Nonetheless, we sought to acknowledge the reviewer's sensible request for verification of matching between runs accuracy. To this end, we performed a detailed investigation of the intensity assignments for

MS/MS and matching identifications within all the mouse organ SUMO data. Overall, only 16.7% of the total assigned SUMO signal was derived from matching between runs, with the large majority of all experimental evidence derived from direct MS/MS identifications. Upon closer inspection of all matched peptides, we found that 79.5% of these occurred to the same organ type, and 96.5% matched to the same chromatographic fraction. The 3.5% of matches corresponding to 'wrong' fractions could potentially be false-positive, which would correspond to 0.6% of the total experimental intensity. This still leaves FDR below 1%, and fraction-mismatched hits are not strictly wrong as peptides frequently appear in neighboring chromatographic fractions. We have added this validation into the Methods and Supplementary Information.

5. If the authors removed the MS1 match between runs hits for the organs can and do they draw the same conclusions? (I.e. K-x-E consensus, etc).

Reply: As outlined above, matching between runs was not used for many of the secondary analyses already included in our initial submission; hence, the described KxE consensus across organs was entirely based on direct MS/MS data. For all analyses we tested with and without matching between runs during our initial manuscript drafting, we did not observe any significant differences.

6. A conclusion of the paper is that higher density of conjugated SUMO is present in a cell line than in mouse organs. This data stems from comparing the overall sample purity for sumoylated proteins after the SUMO peptide pull-down with the amount of background signal and show more free SUMO2/3 in organs by almost twice as much as HEK cells. Can the authors rule out the possibility that this is due to experimental discrepancies. For example, if whole cell extract was immunoblotted with the anti-SUMO2/3 8A2 clone is more free SUMO observed in mouse organs than in HEK cells? Is this a uniform observation across all organs examined? Is it possible that the fraction of conjugated versus free SUMO observed is adversely affected by differences in lysate preparation protocols required for tissues versus cells or the intrinsic levels of desumoylating activity? It seems that there are a number of variables not accounted for here, so some confirmatory data would be reassuring.

Reply: We also found it remarkable to observe that a much higher degree of free SUMO was observed in mouse organs, compared to cultured cells. Indeed, we observed consistent amounts of background signal relative to the total amount of starting protein material, regardless of whether the purifications were

performed from cell lines or mouse organs. At the same time, we did purify less SUMO from mouse organs, indicative of a lower density of SUMOylation, and at the same time found more of this SUMO to be free.

The lysis procedure of the organs was in general synonymous to the procedure used for the cells. Cells were harvested in ice-cold PBS, going through an additional wash, just prior to their lysis with room temperature guanidine. This lysis generally takes around 30 seconds to fully dissolve the cell pellet. Despite this relatively lengthy handling procedure, we have observed no deconjugation of SUMO in these cells, suggesting that at ice-cold temperatures and while the cells are intact, SUMO proteases are either not active or otherwise compartmentalized away from SUMO conjugates. The organs were removed from the mice immediately after euthanizing them by cervical dislocation, and immediately snap frozen in liquid nitrogen, essentially as described previously (Tirard and Brose, 2016). This leaves virtually no time for any biological responses to occur. Organs were kept at -80°C until immediately prior to lysis, which occurred by vigorous bead-grinding in the presence of the same guanidine lysis buffer as used for cells. Additionally, we added fresh chloroacetamide prior to lysis to further counteract any potential SUMO protease activity. The bead-grinding procedure generally resulted in a homogenous liquid after just a few seconds, but we regardless used 20 second cycles. Thus, the homogenization of the organs occurred at least as fast as the lysis of cultured cells.

Moreover, whereas all mice were sacrificed on the same day, the processing of the organs was done in batches. All organs from mouse-1 were processed first, all organs from mouse-2 and mouse-3 two months later, and all organs from mouse-4 and mouse-5 two months after that. The order in which the different organs were homogenized was randomized. Important to note is that we did not, for example, process all brains together, or all lungs together, as that could indeed introduce an experimental bias. Despite the sample processing staggered apart by several months, and with organs originating from different animals, we still observed consistent ratios of free and conjugated SUMO at $n=5$ between same organs, yet distinct ratios between different types of organs. If uncontrolled amounts of deconjugation were to occur, we would not have acquired results with the same high consistency.

However, we did seek to fulfill the reviewer's request, by performing immunoblot analysis on whole HEK lysates and whole organ lysates and directly comparing them. One 'caveat' of our MS method is that lysis happens in guanidine buffer, which renders it innately incompatible with immunoblot analysis because guanidine cannot co-exist in solution with the SDS or LDS required for immunoblot analysis. Thus, we obtained additional mice of the same strain, gender, and age, and sacrificed them and acquired their organs as described above. As we could not use guanidine for lysis, we instead opted to lyse the mouse organs in high SDS buffer, essentially as described previously for endogenous SUMO proteomics (Barysch et al., 2014). Moreover, we

supplemented the lysis buffer with broad-range protease inhibitors and fresh NEM to a final concentration of 20 mM, further suppressing any deSUMOylation activity that could occur during lysis and homogenization. To achieve a fair comparison between organs and HEK cells, we harvested the HEK cells and pelleted them in the homogenizer tubes, prior to snap freezing the pellets with liquid nitrogen and from that point subjecting the HEK cell pellets to exactly the same procedure as the mouse organs. Within these carefully controlled conditions, any free SUMO observed in the samples will have originated from the biological state of the samples.

Next, we analyzed these lysates using immunoblotting, using the same 8A2 anti-SUMO antibody as we used throughout our study. Reassuringly, the immunoblot analysis (Figure S11) supports our mass spectrometry findings. First, we observed a much higher density of SUMO2/3 in HEK cells, with the SUMO2/3 signal for HEK dominating the picture even though we loaded 10 times less total protein on the gel (as supported by the Ponceau-S stain). Second, virtually all SUMO2/3 appeared as conjugated in HEK cells, whereas considerable pools of free SUMO2/3 were observed across all organs. Considering the experimental consistency between five replicates resulting from guanidine lysis and MS analysis, and two additional replicates resulting from SDS lysis and immunoblot analysis, we believe this firmly proves that deSUMOylation did not occur during sample preparation. We have updated the manuscript text to include the immunoblot data.

Still, despite immunoblot analysis globally supporting our MS observations, our described MS method provides several analytical advantages which are not easily be obtained through immunoblot analysis. For example, from a single MS analysis we concomitantly obtain information on free versus conjugated SUMO2/3, which SUMO enzymes are auto-modified, SUMO-chain topology, the exact proteins and lysine residues being modified by SUMO2/3, and quantification of immature and mature SUMO levels. Moreover, for some organs with very low SUMO density the sensitivity of immunoblot analysis is lacking, e.g. for muscle and heart. Further, depending on the electrophoretic methods used, the anti-SUMO2/3 antibody used, and the organ type analyzed, immunoblot could generate non-specific or non-linear signals, thereby prohibiting accurate quantification. Collectively, we believe that our MS strategy presents a superior tool for analysis of the SUMO architecture in any vertebrate model system.

7. The use of GO terms to classify the Sumoylated proteins (Fig.2G, 3C) is something often done in proteomics papers, but is arguably of little or no value to biologists working in this field. For example, Figure 3C references

how SUMO sites in the brain “we observed enrichment for synapse, myelin sheath, neuron...”. Without some data elucidating how the SUMO proteome controls functions in specific tissues like the brain, the discussions surrounding this data do not meaningfully extend the field’s understanding of SUMO function beyond the value of the data provided by the dataset itself.

Reply: As also noted in reply to Reviewers #1 and #2, we have now revisited the GO term analysis. It now only displays SUMO-enriched terms with the relevant background proteomes, unique across the eight organs we tested, and is no longer based on matching between runs data. We agree that simply listing off the terms from the figure in the main text is redundant, and we have now shortened this text. The overall term enrichment analysis is still an important part of the manuscript, as it quickly summarizes the enriched functions without the reader having to dig through the supplementary tables.

8. The PCA analysis in Figure 3B shows reproducibility of the experiment, but otherwise doesn’t add much. It seems expected that data sets cluster per organ based on differences in the baseline proteome.

Reply: The PCA clusters the replicates by organ which is what we hoped for, and we were happy to see the same-organ replicates cluster successfully. From our own experience, there are many cases where technical or other unwanted variance can overwhelm the desired variation, and where the PCA does not end up looking this consistent. It does indeed seem expected that replicates from the same organ types, cell lines, or treatment conditions, would cluster together so well in PCA, but it nonetheless demonstrates that the variance between the biological nature of the investigated samples was larger than undesired or unexplained variance. Moreover, the PCA does capture some co-variation in this specific case, for example between liver and kidney, and to a certain extent between lung and spleen. It also places brain far away from other organ types, and we find the overall SUMOylation in brain to be quite different from other organs. Overall, we believe that this PCA is both technically and biologically relevant enough to warrant its presence in a main display figure. Besides, Reviewer #2 specifically requested inclusion of more replicate analysis related to the mouse organ analysis, hence, we believe that the PCA analysis provides relevant information for the readers.

9. Page 11 second paragraph, the figure is referenced should be 4B, not 3B, in the section “Subcellular localization”

Reply: We have corrected the erroneous reference.

10. Figure 4F and 4G are quite similar... is it necessary to convey the same point?

Reply: Figure 4F focuses more on SUMO itself, and to keep it similar to Figure 4E it contains a global overlap to ubiquitin, which is based on all data extracted from the PhosphoSitePlus database. Figure 4G, on the other hand, focuses on a direct per-organ comparison of SUMO versus ubiquitin, and is based on a single proteomics study performed in a similar manner. We believe this allows a more fair comparison, and Figure 4G also demonstrates an even more marked preference for ubiquitin to target structured protein regions, especially in liver. We updated the manuscript to more elaborately describe Figure 4G.

11. Figure legend in 3A could be improved by adding a legend for the heatmap.

Reply: We have added a colored legend below the heatmap.

12. Have the authors attempted to treat anti-8A2 enriched peptides with a SUMO isopeptidase like SENP2 to cleave the 1kDa remnant? Is it possible for the mass spec sequence the unmodified peptide that corresponds to the originally identified SUMO site? If so, this would validate the SUMO sites and potentially reveal additional peptides that were otherwise refractory to MS-identification with the remnant attached.

Reply: This is a very interesting idea, and bears resemblance to previously published strategies (Andersen et al., 2011; Hendriks et al., 2015). We did not attempt this, and if an isopeptidase could remove the 1 kDa mass remnant, it would indeed be possible for the MS to detect the resulting unmodified peptide – and potentially peptides that would otherwise be resistant to identification with the mass remnant attached. As outlined in the PRISM paper (Hendriks et al., 2015), however, the lack of an actual mass remnant attached to the target peptide requires a different strategy for validation that the peptide was actually modified, and even with these controls in place there is a considerable false-discovery rate associated with these methods (Hendriks et al., 2015). Controls would be required with and without isopeptidase, and moreover a form of pre-labeling (e.g. SILAC) of the experimental peptides would be necessary to avoid any background peptides from contaminating the sample and then being identified as a false-positive. Such experimental alterations are not straightforward, and the optimizations required could take considerable periods of time.

We believe our current experimental strategy is highly optimized for its intended purpose. The high mass accuracy of the mass spectrometric analysis, in combination with stringent FDR and a suite of diagnostic ions originating from the unique SUMO-2/3 mass remnant, already provides a very high identification confidence. Further confirmation of the accuracy of our approach is provided by the considerable overlap with other SUMO proteomics strategies, and the faithful re-identification of many previously confirmed SUMO sites.

13. The sentence that reads “Reassuringly, this median copy number is on par with the second-largest exogenous SUMO study(ref28), and just short in depth of the largest exogenous SUMO study(ref2) which achieved a median copy number of 44,000 (Fig. 1E).” feels like an gratuitous self-reference and an unnecessary jab at another of the labs working in this field.

Reply: We apologize for any potential offense entailed within this sentence – we did not mean any. We have removed the sentence in question, and replaced it with a more moderate alternative.

Reference List

1. Andersen, J.L., Thompson, J.W., Lindblom, K.R., Johnson, E.S., Yang, C.S., Lilley, L.R., Freel, C.D., Moseley, M.A., and Kornbluth, S. (2011). A biotin switch-based proteomics approach identifies 14-3-3zeta as a target of Sirt1 in the metabolic regulation of caspase-2. *Mol. Cell* 43, 834-842.
2. Barysch, S.V., Dittner, C., Flotho, A., Becker, J., and Melchior, F. (2014). Identification and analysis of endogenous SUMO1 and SUMO2/3 targets in mammalian cells and tissues using monoclonal antibodies. *Nat. Protoc.* 9, 896-909.
3. Bekker-Jensen, D.B., Kelstrup, C.D., Batth, T.S., Larsen, S.C., Haldrup, C., Bramsen, J.B., Sorensen, K.D., Hoyer, S., Orntoft, T.F., Andersen, C.L., Nielsen, M.L., and Olsen, J.V. (2017). An Optimized Shotgun Strategy for the Rapid Generation of Comprehensive Human Proteomes. *Cell Syst.* 4, 587-599.
4. Chomczynski, P., and Sacchi, N. (1987). Single-step method of RNA isolation by acid guanidinium thiocyanate-phenol-chloroform extraction. *Anal. Biochem.* 162, 156-159.
5. Cox, J., Hein, M.Y., Luber, C.A., Paron, I., Nagaraj, N., and Mann, M. (2014). Accurate proteome-wide label-free quantification by delayed normalization and maximal peptide ratio extraction, termed MaxLFQ. *Mol. Cell Proteomics.* 13, 2513-2526.

6. Cox, J., and Mann, M. (2008). MaxQuant enables high peptide identification rates, individualized p.p.b.-range mass accuracies and proteome-wide protein quantification. *Nat. Biotechnol.* 26, 1367-1372.
7. Cox, J., Neuhauser, N., Michalski, A., Scheltema, R.A., Olsen, J.V., and Mann, M. (2011). Andromeda: a peptide search engine integrated into the MaxQuant environment. *J. Proteome. Res.* 10, 1794-1805.
8. Flotho, A., and Melchior, F. (2013). Sumoylation: a regulatory protein modification in health and disease. *Annu. Rev. Biochem.* 82, 357-385.
9. Hendriks, I.A., D'Souza, R.C., Chang, J.G., Mann, M., and Vertegaal, A.C. (2015). System-wide identification of wild-type SUMO-2 conjugation sites. *Nat. Commun.* 6, 7289.
10. Hendriks, I.A., D'Souza, R.C., Yang, B., Verlaan-de Vries, M., Mann, M., and Vertegaal, A.C. (2014). Uncovering global SUMOylation signaling networks in a site-specific manner. *Nat. Struct. Mol. Biol.* 21, 927-936.
11. Hendriks, I.A., Lyon, D., Young, C., Jensen, L.J., Vertegaal, A.C., and Nielsen, M.L. (2017). Site-specific mapping of the human SUMO proteome reveals co-modification with phosphorylation. *Nat. Struct. Mol. Biol.* 24, 325-336.
12. Hornbeck, P.V., Kornhauser, J.M., Tkachev, S., Zhang, B., Skrzypek, E., Murray, B., Latham, V., and Sullivan, M. (2012). PhosphoSitePlus: a comprehensive resource for investigating the structure and function of experimentally determined post-translational modifications in man and mouse. *Nucleic Acids Res.* 40, D261-D270.
13. Kahl, A., Blanco, I., Jackman, K., Baskar, J., Milaganur, M.H., Rodney-Sandy, R., Zhang, S., Iadecola, C., and Hochrainer, K. (2018). Cerebral ischemia induces the aggregation of proteins linked to neurodegenerative diseases. *Sci. Rep.* 8, 2701.
14. Krey, J.F., Wilmarth, P.A., Shin, J.B., Klimek, J., Sherman, N.E., Jeffery, E.D., Choi, D., David, L.L., and Barr-Gillespie, P.G. (2014). Accurate label-free protein quantitation with high- and low-resolution mass spectrometers. *J. Proteome. Res.* 13, 1034-1044.
15. Lamoliatte, F., Caron, D., Durette, C., Mahrouche, L., Maroui, M.A., Caron-Lizotte, O., Bonneil, E., Chelbi-Alix, M.K., and Thibault, P. (2014). Large-scale analysis of lysine SUMOylation by SUMO remnant immunoaffinity profiling. *Nat. Commun.* 5, 5409.
16. Lamoliatte, F., McManus, F.P., Maarifi, G., Chelbi-Alix, M.K., and Thibault, P. (2017). Uncovering the SUMOylation and ubiquitylation crosstalk in human cells using sequential peptide immunopurification. *Nat. Commun.* 8, 14109.
17. Lumpkin, R.J., Gu, H., Zhu, Y., Leonard, M., Ahmad, A.S., Clauser, K.R., Meyer, J.G., Bennett, E.J., and Komives, E.A. (2017). Site-specific identification and quantitation of endogenous SUMO modifications under native conditions. *Nat. Commun.* 8, 1171.

18. Olsen, J.V., Blagoev, B., Gnäd, F., Macek, B., Kumar, C., Mortensen, P., and Mann, M. (2006). Global, in vivo, and site-specific phosphorylation dynamics in signaling networks. *Cell* 127, 635-648.
19. Reva, B., Antipin, Y., and Sander, C. (2011). Predicting the functional impact of protein mutations: application to cancer genomics. *Nucleic Acids Res.* 39, e118.
20. Santos, A., Tsafou, K., Stolte, C., Pletscher-Frankild, S., O'Donoghue, S.I., and Jensen, L.J. (2015). Comprehensive comparison of large-scale tissue expression datasets. *PeerJ.* 3, e1054.
21. Tammsalu, T., Matic, I., Jaffray, E.G., Ibrahim, A.F., Tatham, M.H., and Hay, R.T. (2014). Proteome-Wide Identification of SUMO2 Modification Sites. *Sci. Signal.* 7, rs2.
22. Tammsalu, T., Matic, I., Jaffray, E.G., Ibrahim, A.F., Tatham, M.H., and Hay, R.T. (2015). Proteome-wide identification of SUMO modification sites by mass spectrometry. *Nat. Protoc.* 10, 1374-1388.
23. Tirard, M., and Brose, N. (2016). Systematic Localization and Identification of SUMOylation Substrates in Knock-In Mice Expressing Affinity-Tagged SUMO1. *Methods Mol. Biol.* 1475, 291-301.
24. Westman, B.J., and Lamond, A.I. (2011). A role for SUMOylation in snoRNP biogenesis revealed by quantitative proteomics. *Nucleus.* 2, 30-37.

Reviewers' comments:

Reviewer #1 (Remarks to the Author):

The authors have addressed all my points.

I recommend the study by Hendricks et al. to be published in nature Communications.

Reviewer #2 (Remarks to the Author):

The revised version of the manuscript has been improved. The result is now clearer but not mature and sufficiently comprehensible. Moreover, I suggest to rewrite and shorten the discussion section as it is more than 4 pages long without labeled paragraphs making it quite difficult to read.

In addition, the following points have not been answered sufficiently or do not support the presented results:

Differences in quantitative SUMOylation cannot be distinguished from changes of the proteins. The comparison with known expressions in the TISSUE database is not enough as individual protein changes in e.g. mice could perturb the overall picture. To my knowledge, it is now standard to adjust modified peptides by the protein amounts which is usually done by running the samples in parallel without any enrichment for the PTM in question. Maybe in this manuscript the main conclusions should be merely based on identifications and not quantifications?

Protein copy numbers and Fig. 1e: This part is not well explained and I am still not sure what the authors are trying to show. Did they take the protein expression values from the Bekker-Jensen study for the identified SUMOylated proteins in the respective studies and then show their distributions? If yes, this is not clearly written. If I remember correctly, the protein expression values in the Bekker-Jensen study were obtained from HeLa cells. The authors should mention that they assume that these protein expressions are representative for human cell lines in general.

It would be nice to see the actual distributions in Fig. 1e. This can be done using violin plots. Moreover, the statement (in the reply to the reviewers) that symmetric boxplots demonstrate normally distributed data is wrong. It demonstrates symmetry and that is it. Distribution plots like violin plots will give a better idea of the actual distributions.

The gene enrichment analysis is now more accurate as the authors changed the backgrounds to the organ proteomes. However, the observed changes of the SUMO sites can still result from changes of the protein as this work does not correct for protein abundances. Therefore, I do not see that the results yield detailed information about changes in SUMOylation.

Fig. S3b. Which numbers correspond to the color scale? Is it standard deviation or z scores?

Last sentence of "Mapping the endogenous SUMOylome ...": $R \sim 0.5$ does not correspond to "significant Pearson correlations". Given that rather different data sets were compared, I suggest to substitute "significant" by "reasonable" or alike.

Figure 3D: How were the values calculated? What is "correlation with protein abundance" of SUMO sites with proteins? What does the dashed line show? Such comparison with the TISSUE proteomes is not sufficient to pinpoint the observed quantitative changes to SUMOylation events. Low correlation between proteins and SUMOylations could also result from noisy measurements and/or different phenotypes and therefore are not worthy indicators. Given that the shown analysis in Fig. 3D would make sense, then the interpretation of SUMO changes in heart and

muscle would be without meaning as indirectly stated by the authors.

Gene Set Enrichment Analysis is still not an annotation term or category but a method to determine enriched terms in genes/proteins (such as "annotation of proteins"). I am not familiar with Perseus but there is something wrong in the description.

Reviewer #3 (Remarks to the Author):

The revised manuscript entitled "Endogenous, in vivo, and site-specific characterization of the SUMO architecture across species and organs" demonstrates a new methodology to identify substrates modified by native and endogenous SUMO2/3. The text shows that the technique generates data comparable to exogenous SUMOylation site mapping and can be applied to mouse organs. The dataset is a thoughtfully assembled resource, but from the standpoint of novelty, largely reconfirms what has been described in other similar SUMO proteomic studies. Even the concept of profiling the endogenous SUMO proteome has been the topic of multiple recent publications (i.e. Lumpkin et al 2017, Cai et al 2017), each pairing an optimized proteolytic step with a well matched immunoprecipitation reagent. The value of the study rests more on the depth and breadth of the SUMO2/3 proteome, rather than new biologies of SUMO itself. The text itself remains cumbersome even for readers with interest in proteomics technology. In that regard, this reviewer remains uncertain of its suitability for Nature Communications versus a more proteomics focused journal. A couple key points are below.

In addition to highlighting their new method, a key point of this paper is the importance of having profiled many Sumoylation events in tissues. Yet, this topic seems to be underdeveloped. Figures 1, 2, and 4 focus on comparing this method with exogenous SUMO site profiling, reaching the same general conclusions about the nature of Sumoylation. Only Figure 5 tries to tackle how this method could be used to address new and interesting functions of SUMO in tissues, such as with respect to poly-SUMO chains. The study doesn't go further in depth to investigate any potentially interesting Sumoylation events, focusing instead on numbers and superficial correlations.

A major concern originally, and still now, is the comparison between the organ proteome and the associated SUMO sites. Using the TISSUE database as a background proteome is an improvement, but still leaves plenty of room for overinterpretation. These concerns would be best addressed by follow up experimentation that shows the benefits of having identified in vivo Sumoylation events in a tissue specific manner.

A minor concern is that the figure (Supp Data Set 1) showing the representative mass spectra for the 1KDa remnant fragment ions could not be found in the upload.

Point-by-Point response to reviewers

General remarks:

We would like to thank all the Reviewers for their valuable remarks and criticism, and are happy to see that we addressed most of the concerns by all three Reviewers. In this revision, we have addressed remaining concerns raised by Reviewers #2 and #3, and focused on greatly shortening the main manuscript text, which remained quite lengthy. We believe that the manuscript is now concise and more accessible to a broad audience, and in a careful manner describes an optimized endogenous SUMO2/3 methodology, a rich Resource, and several novel biological insights.

Reviewer #1 (Remarks to the Author):

The authors have addressed all my points. I recommend the study by Hendricks et al. to be published in nature Communications.

Reply: We are happy to read that we addressed all the points raised by Reviewer #1, and would once again like to thank the Reviewer for the helpful and constructive criticism which allowed us to improve the quality of our manuscript.

Reviewer #2 (Remarks to the Author):

The revised version of the manuscript has been improved. The result is now clearer but not mature and sufficiently comprehensible. Moreover, I suggest to rewrite and shorten the discussion section as it is more than 4 pages long without labeled paragraphs making it quite difficult to read.

Reply: We are delighted to hear Reviewer #2 acknowledges the high quality of our revised manuscript, as we did our best to respond to all of the previous points raised by the referees. We agree that the manuscript remained on the lengthy side, and have now significantly reduced the length of the manuscript text and discussion in order to yield a more concise manuscript. Superfluous text has been moved to supplementary information. On a side note, we initially used labeled paragraphs in our discussion, but removed these during the revision as Nature Communications style does not use subheadings in the discussion.

In addition, the following points have not been answered sufficiently or do not support the presented results:

Differences in quantitative SUMOylation cannot be distinguished from changes of the proteins. The comparison with known expressions in the TISSUE database is not enough as individual protein changes in e.g. mice could perturb the overall picture. To my knowledge, it is now standard to adjust modified peptides by the protein amounts which is usually done by running the samples in parallel without any enrichment for the PTM in question. Maybe in this manuscript the main conclusions should be merely based on identifications and not quantifications?

Reply: We agree with the reviewer that reference proteomes from databases may not precisely correspond to the exact model system used. They are, however, the closest and most-reliable approximation we are aware of. In terms of adjustment of the PTM-enriched samples for total input levels, whereas this may be more commonly applied in the field of phosphoproteomics, this is not at all common for SUMO proteomics. To our knowledge (Hendriks and Vertegaal, 2016), only one SUMO proteomics screen has attempted to correct for total proteome input before (Schimmel et al., 2014), and here a larger part of the identified SUMOylated proteins (39%) were not detected at all in the total proteome, since SUMO tends to target proteins expressed at lower levels (Hendriks et al., 2017). Similarly, within our data we observed SUMOylation (by site MS/MS) of several proteins (9%) in mouse which were previously not found to be expressed from the TISSUES database, indicating a very good, although still not complete overlap. While an in-depth total proteome analysis of all organ types in question may contain interesting information, such experiments would be very arduous and would require an extraordinary amount of MS machine time. Moreover, such experiments would generate a very large additional amount of data that would further add to an already sizeable amount of data (as already emphasized by the reviewer). Besides, the availability of such tissue-specific proteomes would likely still not reach the comprehensiveness of the TISSUES database. Thus, we consider total proteome analysis beyond the scope of this manuscript.

Importantly, we would like to mention that we already changed the GO analyses to be qualitative rather than quantitative when we compare against the TISSUES database. To elaborate on this, we checked whether proteins were detected as SUMOylated by SUMO site MS/MS, and if so they were considered as SUMO target proteins when compared to the reference proteomes. This is standard procedure for virtually all published

SUMO proteomics studies, where frequently the entire theoretical proteome is used as a background. Instead, we utilized the more limited and organ-specific subsets of expressed proteins we now used. Whereas quantitative numbers are listed in the supplementary tables, we did not base any main conclusions on these. We have now stressed in the main manuscript text and the associated figure legend that GO analyses are qualitative (i.e. based on identifications) rather than quantitative. Still, we would like to emphasize that the results remain significant and relevant.

Protein copy numbers and Fig. 1e: This part is not well explained and I am still not sure what the authors are trying to show. Did they take the protein expression values from the Bekker-Jensen study for the identified SUMOylated proteins in the respective studies and then show their distributions? If yes, this is not clearly written. If I remember correctly, the protein expression values in the Bekker-Jensen study were obtained from HeLa cells. The authors should mention that they assume that these protein expressions are representative for human cell lines in general.

Reply: The reviewer is correct, and we took the protein expression values from the Bekker-Jensen study and made the assumption that these will largely correlate with other human cell lines. Whereas there is likely to be some variance in expression levels of proteins between the different cell lines compared, we believe that such variation is likely to average out when considering hundreds to thousands of proteins in the comparisons. The similarity in protein expression across various cancer cell lines was already shown in our referenced manuscript (Bekker-Jensen et al., 2017), where in-depth proteome analysis of five different cell lines overall revealed highly comparable expression levels across the vast majority of expressed proteins. We have now mentioned this in the manuscript text and figure legend.

It would be nice to see the actual distributions in Fig. 1e. This can be done using violin plots. Moreover, the statement (in the reply to the reviewers) that symmetric boxplots demonstrate normally distributed data is wrong. It demonstrates symmetry and that is it. Distribution plots like violin plots will give a better idea of the actual distributions.

Reply: The Reviewer is correct, and we cannot assume normally distributed data directly from symmetry. We have now included the 5th and 95th percentile in the data, and additionally overlaid histograms of the exact distributions of expression levels of detected SUMO target proteins in Figure 1E, as suggested by the reviewer.

The gene enrichment analysis is now more accurate as the authors changed the backgrounds to the organ proteomes. However, the observed changes of the SUMO sites can still result from changes of the protein as this work does not correct for protein abundances. Therefore, I do not see that the results yield detailed information about changes in SUMOylation.

Reply: As outlined above, the enrichment analyses were changed in the previous revision to be qualitative (i.e. based on identifications), meaning that we compared the presence or absence of SUMOylated proteins (by MS/MS SUMO site) associated with specific terms versus the background proteomes, with each term then evaluated using two-tailed Fisher's testing followed by Benjamini-Hochberg multiple-hypotheses correction to ensure p-values of <0.02 . This is standard procedure in the SUMO proteomics field.

Fig. S3b. Which numbers correspond to the color scale? Is it standard deviation or z scores?

Reply: The number represents standard deviation distance from the mean, as calculated by z-scoring. This is described in the methods section and in the figure legends.

Last sentence of "Mapping the endogenous SUMOylome ...": $R \sim 0.5$ does not correspond to "significant Pearson correlations". Given that rather different data sets were compared, I suggest to substitute "significant" by "reasonable" or alike.

Reply: We respectfully disagree with the reviewer's notion. Considering the number of data points used for the presented analyses, a Pearson correlation of 0.56 or 0.59 is highly significant (Bewick et al., 2003). Still, we have removed "significant" from the sentence, as suggested by the reviewer.

Figure 3D: How were the values calculated? What is "correlation with protein abundance" of SUMO sites with proteins? What does the dashed line show? Such comparison with the TISSUE proteomes is not sufficient to pinpoint the observed quantitative changes to SUMOylation events. Low correlation between proteins and SUMOylations could also result from noisy measurements and/or different phenotypes and therefore are not worthy indicators. Given that the shown analysis in Fig. 3D would make sense, then the interpretation of SUMO changes in heart and muscle would be without meaning as indirectly stated by the authors.

Reply: The values displayed in figure 3D represent ratios between the expression levels of SUMO target proteins detected within the organ, as compared to the expression levels of those exact same proteins across the other 7 organs. To this end, we used the 1st quantile, mean, median, and 3rd quantile expression values to calculate ratios, with these values derived from the TISSUES database (Mouse GeneAtlas V3). Thus, this analysis merely serves to show how much higher expressed proteins are in the organ where we find them to be SUMOylated (in a qualitative manner), as compared to the other organs. The dotted line is situated at a ratio of 1, which would indicate no difference in expression of these SUMO target proteins in other organs, which we have now added to the legend. The “correlation with protein abundance” refers to those organs, specifically brain, lung, spleen, and testis, where the ratio is not much larger than 1, which suggests that in these organs SUMOylation targets proteins that are not expressed at a higher level compared to other organs.

Indeed, some of the proteins we detect to be SUMOylated in heart and muscle are much higher expressed in those organs as compared to other organs. We believe this is fair to report, as abundance bias in detecting organ- or sample-enriched proteins is to be expected and also serves as technical validation of our method. Further biological conclusions, such as GO analyses or analysis of the SUMO equilibrium, are contained within the specific organ types and tested against the relevant backgrounds, and are thus not affected by expression bias.

Gene Set Enrichment Analysis is still not an annotation term or category but a method to determine enriched terms in genes/proteins (such as "annotation of proteins"). I am not familiar with Perseus but there is something wrong in the description.

Reply: We apologize for the confusion, and hope that the following explanation of the Perseus workflow can clarify the situation. In the context of Perseus, and using GSEA, all proteins in the reference proteome are labeled with the identifier name(s) of the Gene Set(s) they are part of. This is on a purely qualitative presence-or-absence basis. Next, proteins that we detect as SUMO targets are considered as the foreground cluster, and then tested versus the background as outlined above, considering all terms associated with the proteins, which would in this case be their participation in the gene set. If a significant (multiple-hypotheses corrected) number of proteins from a Gene Set is found to be SUMOylated (i.e. in the foreground), then we consider the Gene Set to be (qualitatively) targeted by SUMOylation more-so than randomly expected.

Reviewer #3 (Remarks to the Author):

The revised manuscript entitled “Endogenous, in vivo, and site-specific characterization of the SUMO architecture across species and organs” demonstrates a new methodology to identify substrates modified by native and endogenous SUMO2/3. The text shows that the technique generates data comparable to exogenous SUMOylation site mapping and can be applied to mouse organs. The dataset is a thoughtfully assembled resource, but from the standpoint of novelty, largely reconfirms what has been described in other similar SUMO proteomic studies. Even the concept of profiling the endogenous SUMO proteome has been the topic of multiple recent publications (i.e. Lumpkin et al 2017, Cai et al 2017), each pairing an optimized proteolytic step with a well matched immunoprecipitation reagent.

Reply: We thank the reviewer for appreciating our thoughtfully assembled resource, as one of the central goals of our manuscript is to serve as a strong Resource for the research community. With regard to the studies published by Lumpkin et al. and Cai et al., we were previously aware of these studies and in the first submission already included an extensive comparison to those methods, highlighting why we believe our method entails several analytical advantages. Please see the Supplementary Note included in our original submission for details. Mainly, our method is truly specific for SUMO-2/3 and thus fills a unique niche, and is more sensitive than the other methods published so far. Secondly, our described method allows for quantitative assessment of the SUMO equilibrium across cells and tissue, which is not at all feasible with any contemporary methodology. In the first revision, we additionally validated our quantitative observations using WB, as per request from the reviewer, which collectively confirms the validity of our analysis and the novel insights it provides.

We would like to note that developing an MS proteomics methodology for studying a specific PTM is in our opinion not as simple as merely matching an optimized proteolytic step with an IP reagent, and in this case was a laborious and costly process that involved numerous rounds of careful optimization. The main outcomes of these optimization steps are included and described in the supplementary data provided with the revised manuscript. The oldest MS RAW data files included with our manuscript date back to June 2016, demonstrating that while we may not be the first to publish on endogenous SUMO methodology, we have worked on this

method for quite some time without having other studies to serve as inspiration. Considering the strengths of our method compared to others, and the associated compendium of *in vivo* SUMO sites, we believe our manuscript will be of tremendous value and interest to the community.

The value of the study rests more on the depth and breadth of the SUMO2/3 proteome, rather than new biologies of SUMO itself. The text itself remains cumbersome even for readers with interest in proteomics technology. In that regard, this reviewer remains uncertain of its suitability for Nature Communications versus a more proteomics focused journal. A couple key points are below.

Reply: We appreciate the reviewer's acknowledgement of the depth and breadth of our Resource, which sets it apart from the previously published papers. Moreover, even with the primary purpose of our manuscript being a reliable and rich Resource, we additionally present biological novelties that can currently only be uniquely assessed with our method, as also mentioned by the Reviewer in the next paragraph. Previously, methodology-centric SUMO proteomics papers (Hendriks et al., 2015; Lumpkin et al., 2017) were also published in Nature Communications, both of which arguably presented less biological data than we now do. Furthering this, we would like to emphasize that similar resources describing tissue-wide analysis of other PTMs, including phosphorylation (Huttlin et al., 2010) and glycosylation (Zielinska et al., 2010), have all been published in very high impact journals, and obtained a high number of citations (657 and 459 citations, respectively, according to Scopus). Thus, we do not believe that Nature Communications would be an unsuitable journal.

Finally, we agree that the manuscript remained cumbersome after re-submission, and we have now further shortened the manuscript text and discussion, and moved superfluous text into supplementary information.

In addition to highlighting their new method, a key point of this paper is the importance of having profiled many Sumoylation events in tissues. Yet, this topic seems to be underdeveloped. Figures 1, 2, and 4 focus on comparing this method with exogenous SUMO site profiling, reaching the same general conclusions about the nature of Sumoylation. Only Figure 5 tries to tackle how this method could be used to address new and interesting functions of SUMO in tissues, such as with respect to poly-SUMO chains. The study doesn't go further in depth to investigate any potentially interesting Sumoylation events, focusing instead on numbers and superficial correlations.

Reply: We believe validation of our method and data, and benchmarking our results versus previously published screens, is a key aspect of ensuring that our method is technically sound and the Resource will not disappoint the readership. Moreover, the associated data sets allow readers to quickly compare SUMO sites and SUMO target proteins across multiple screens and methods. Thus, figures 1, 2, and 4 serve a valuable purpose. Moreover, figure 2 for the first time confirms that basic targeting properties of SUMOylation also exist across multiple types of mouse organs. Although the outcome of these experiments may come as no surprise to the Reviewer, these core properties of SUMOylation were not feasible to investigate until now at the endogenous and *in vivo* level, as the methodology for conducting such analyses was simply not available. Figure 3 further expands on this by highlighting terms that are uniquely SUMOylated on a per-organ basis. Figure 4H-I entail an evolutionary conservation analysis which has not previously been performed for SUMO. As acknowledged by the Reviewer, Figure 5 demonstrates new SUMO biology, which uniquely can be assessed at the *in vivo* proteomics level using our methodology only. From all the data we additionally distill more insights, such as the overall distinct pattern of SUMOylation in mouse brain, and the enriched SUMOylation in mouse testis. Finally, we believe there is great value in the in-depth comparisons between multiple SUMO proteomics studies,

We agree with the Reviewer that the technical confirmations entailed within figures 1, 2, and 4, may not necessarily be of the greatest interest to a wider audience. Thus, we have focused the shortening of the manuscript text associated with these figures, leading to a more concise and less cumbersome manuscript.

A major concern originally, and still now, is the comparison between the organ proteome and the associated SUMO sites. Using the TISSUE database as a background proteome is an improvement, but still leaves plenty of room for overinterpretation. These concerns would be best addressed by follow up experimentation that shows the benefits of having identified *in vivo* Sumoylation events in a tissue specific manner.

Reply: Using the TISSUES database as reference profiles for the specific organs is, to our knowledge, the fairest way of statistically evaluating our data in a qualitative manner. As also described above in reply to Reviewer #2, profiling in-depth reference proteomes for all mouse organs is not feasible, and also not necessarily better, as there would be considerably more missing proteins in the background proteome.

Previously, we included re-analysis and more supporting figures to address specific concerns raised by the Reviewer. As mentioned by the Reviewer, and in part owing to the previous revision expanding the story further, our manuscript is already very extensive. Our Resource contains a wealth of data on the endogenous SUMOylome across eight organs, and this catalogue of *in vivo* SUMOylation events will be far more valuable in the hands of the community compared to just our own. Respectfully, we also do not believe our manuscript would benefit from further general follow-up experimentation, as we fear that this would unnecessarily complicate our manuscript.

A minor concern is that the figure (Supp Data Set 1) showing the representative mass spectra for the 1KDa remnant fragment ions could not be found in the upload.

Reply: We apologize for the missing Supplementary Data Set 1, and have now ensured that all files are properly uploaded.

Reference List

1. Bekker-Jensen, D.B., Kelstrup, C.D., Batth, T.S., Larsen, S.C., Haldrup, C., Bramsen, J.B., Sorensen, K.D., Hoyer, S., Orntoft, T.F., Andersen, C.L., Nielsen, M.L., and Olsen, J.V. (2017). An Optimized Shotgun Strategy for the Rapid Generation of Comprehensive Human Proteomes. *Cell Syst.* 4, 587-599.
2. Bewick, V., Cheek, L., and Ball, J. (2003). Statistics review 7: Correlation and regression. *Crit Care* 7, 451-459.
3. Hendriks, I.A., D'Souza, R.C., Chang, J.G., Mann, M., and Vertegaal, A.C. (2015). System-wide identification of wild-type SUMO-2 conjugation sites. *Nat. Commun.* 6, 7289.
4. Hendriks, I.A., Lyon, D., Young, C., Jensen, L.J., Vertegaal, A.C., and Nielsen, M.L. (2017). Site-specific mapping of the human SUMO proteome reveals co-modification with phosphorylation. *Nat. Struct. Mol. Biol.* 24, 325-336.
5. Hendriks, I.A., and Vertegaal, A.C. (2016). A comprehensive compilation of SUMO proteomics. *Nat. Rev. Mol. Cell Biol.*
6. Huttlin, E.L., Jedrychowski, M.P., Elias, J.E., Goswami, T., Rad, R., Beausoleil, S.A., Villen, J., Haas, W., Sowa, M.E., and Gygi, S.P. (2010). A tissue-specific atlas of mouse protein phosphorylation and expression. *Cell* 143, 1174-1189.

7. Lumpkin, R.J., Gu, H., Zhu, Y., Leonard, M., Ahmad, A.S., Clauser, K.R., Meyer, J.G., Bennett, E.J., and Komives, E.A. (2017). Site-specific identification and quantitation of endogenous SUMO modifications under native conditions. *Nat. Commun.* 8, 1171.
8. Schimmel, J., Eifler, K., Sigurðsson, J.O., Cuijpers, S.A., Hendriks, I.A., Verlaan-de Vries, M., Kelstrup, C.D., Francavilla, C., Medema, R.H., Olsen, J.V., and Vertegaal, A.C. (2014). Uncovering SUMOylation Dynamics during Cell-Cycle Progression Reveals FoxM1 as a Key Mitotic SUMO Target Protein. *Mol. Cell* 53, 1053-1066.
9. Zielinska, D.F., Gnad, F., Wisniewski, J.R., and Mann, M. (2010). Precision mapping of an in vivo N-glycoproteome reveals rigid topological and sequence constraints. *Cell* 141, 897-907.